



# Experimental diagenesis: Insights into aragonite to calcite transformation of *Arctica islandica* shells by hydrothermal treatment

Laura A. Casella[1*], Erika Griesshaber[1], Xiaofei Yin[1], Andreas Ziegler[2], Vasileios Mavromatis[3,4], Dirk Müller[1], Ann-Christine Ritter[5], Dorothee Hippler[3], Elizabeth M. Harper[6], Martin Dietzel[3], Adrian Immenhauser[5], Bernd R. Schöne[7], Lucia Angiolini[8] and Wolfgang W. Schmahl[1]

[1]Department of Earth and Environmental Sciences and GeoBioCenter, Ludwig-Maximilians-University Munich, Munich, 80333, Germany
[2]Central Facility for Electron Microscopy, University of Ulm, Ulm, 89081, Germany
[3]Institute of Applied Geosciences, Graz University of Technology, Graz, 8010, Austria
[4]Géosciences Environnement Toulouse (GET), CNRS, Toulouse, 31400, France
[5]Institute for Geology, Mineralogy and Geophysics, Ruhr-University Bochum, Bochum, 44801, Germany
[6]Department of Earth Sciences, University of Cambridge, Cambridge, CB2 3EQ, U. K.
[7]Institute of Geosciences, University of Mainz, Mainz, 55128, Germany
[8]Dipartimento di Scienze della Terra "A. Desio", Università degli Studi di Milano, Milano 20133, Italy

*Corresponding author:   Laura Antonella Casella
                                Ludwig-Maximilians-University Munich
                                Department of Earth and Environmental Sciences
                                Theresienstr. 41
                                80333 Munich, Germany
                                Tel.: +49 89 2180-4354
                                eMail: Laura.Casella@lrz.uni-muenchen.de




**Abstract.** Biomineralised hard parts form the most important physical fossil record of past environmental conditions. However, living organisms are not in thermodynamic equilibrium with their environment and create local chemical compartments within their bodies where physiologic processes such as biomineralisation take place. Generating their mineralized hard parts most marine invertebrates thus produce metastable aragonite rather than the stable polymorph of $CaCO_3$, calcite. After death of the organism, the physiological conditions which were present during biomineralisation are not sustained any further and the system moves toward inorganic equilibrium with the surrounding inorganic geological system. Thus, during diagenesis the original biogenic structure of aragonitic tissue disappears and is replaced by inorganic structural features.

In order to understand the diagenetic replacement of biogenic aragonite to non-biogenic calcite, we subjected *Arctica islandica* mollusc shells to hydrothermal alteration experiments. Experimental conditions were between 100 °C and 175 °C with reaction durations between one and 84 days, and alteration fluids simulating meteoric and burial waters, respectively. Detailed microstructural and geochemical data were collected for samples altered at 100 °C (and at 0.1 MPa pressure) for 28 days and for samples altered at 175 °C (and at 0.9 MPa pressure) for 7 and 84 days, respectively. During hydrothermal alteration at 100 °C for 28 days, most but not all of the biopolymer matrix was destroyed, while shell aragonite and its characteristic microstructure was largely preserved. In all experiments below 175 °C there are no signs of a replacement reaction of shell aragonite to calcite in X-ray diffraction bulk analysis. At 175 °C the replacement reaction started after a dormant time of 4 days, and the original shell microstructure was almost completely overprinted by the aragonite to calcite replacement reaction after 10 days. Newly formed calcite nucleated at locations which were in contact with the fluid, at the shell surface, in the open pore system, and along growth lines. In the experiments with fluids simulating meteoric water, calcite crystals reached sizes up to 200 micrometres, while in the experiments with Mg-containing fluids the calcite crystals reached sizes up to one mm after 7 days of alteration. Aragonite is metastable at all applied conditions. A small bulk thermodynamic driving force exists for the transition to calcite, which is augmented by stresses induced by organic matrix and interface energies related to the nanoparticulate architecture of the biogenic aragonite. We attribute the sluggish replacement reaction to the inhibition of calcite nucleation in the temperature window from ca. 50°C to ca. 170°C, or, additionally, to the presence of magnesium. Correspondingly, in $Mg^{2+}$-bearing solutions the newly formed calcite crystals are larger than in $Mg^{2+}$-free solutions. Overall, the aragonite-calcite transition occurs via an interface-coupled dissolution-reprecipitation mechanism, which preserves morphologies down to the sub-micrometre scale and induces porosity in the newly formed phase. The absence of aragonite replacement by calcite at temperatures lower than 175°C contributes to explain why aragonitic or bimineralic shells and skeletons have a good potential of preservation and a complete fossil record.

**Key words.** Biominerals, hydrothermal alteration experiments, bivalves, aragonite, calcite, EBSD, EPMA element maps



**1 Introduction**

The skeletons of marine calcifiers are considered high resolution archives of proxies to understand the evolution of the Earth system. They are widespread in the fossil record and are sensitive to changes in seawater composition – which they record with a limited vital effect (e.g. Brand et al., 2003; Parkinson et al., 2005; Schöne & Surge, 2012; Brocas et al., 2013). However, diagenetic alteration of fossil biogenic carbonates is a significant obstacle in understanding past climate dynamics (Grossmann et al., 1993; Richardson et al., 2001; Immenhauser et al., 2005; Korte et al., 2005). Despite more than a century of research on carbonate diagenesis, many of the controlling processes are still only understood in a qualitative manner (Brand and Veizer, 1980, 1981; Swart, 2015). One of the main problems is that diagenetically altered carbonates occur as the product of a complex alteration pathway with an unknown number of intermediate steps and controlling factors (Immenhauser et al., 2015; Swart, 2015; Ullmann and Korte, 2015). Motivated by the lack of quantitative data on rates and products of marine, meteoric, and burial diagenesis, we performed laboratory-based alteration experiments with *Arctica islandica* shells with the aim to obtain time series data sets. The bivalve *A. islandica* has been studied in several scientific disciplines, i.e. biology (Morton, 2011; Oeschger and Storey, 1993; Taylor, 1976; Strahl et al., 2011), ecology (Beal and Kraus, 1989; Kilada et al., 2007; Lewis et al., 2001; Ridgway et al., 2012; Thórarinsdóttir and Einarsson, 1996), gerontology (Abele, 2002; Ridgway and Richardson, 2011; Strahl, 2007), pollution monitoring (Krause-Nehring et al., 2012; Palmer and Rand, 1977; Swaileh, 1996) and shellfisheries management (Adelaja et al., 1998; Harding et al., 2008; Thórarinsdóttir and Jacobson, 2005). *A. islandica* has also gained profound attention in paleoclimatology due to its long lifespan and its use as a high-resolution long-term archive (e. g. Schöne, 2004; Schöne, 2005a, 2005b; Wanamaker et al., 2008; Marchitto et al., 2000, Butler et al., 2009, Wanamaker et al., 2011, Karney et al., 2012 Schöne, 2013, Butler et al., 2013). On the long-term perspective, *A. islandica* plays an important role in palaeontology, not only as a Neogene palaeoecological and palaeoclimatic archive (e.g. Schöne, 2004; Schöne, 2005a, 2005b; Wanamaker et al., 2008; Marchitto et al., 2000, Butler et al., 2009, Wanamaker et al., 2011, Karney et al., 2012 Schöne, 2013, Butler et al., 2013, Crippa et al., 2016), but also as a biostratigraphic tool. Formerly considered a marker for the Pliocene-Pleistocene boundary (Raffi, 1986) in the Mediterranean region, its first appearance is now regarded as an indicator of the Gelasian-Calabrian (Early Pleistocene) boundary, around 1.7 Ma (Crippa & Raineri, 2015). The potential of this species for palaeontology is strictly dependent on its preservation, thus, the dynamics of diagenetic shell alteration.

At ambient conditions calcite is the stable and, thus, the least soluble polymorphic phase of $CaCO_3$ (Plummer & Mackenzie, 1974; Plummer & Busenberg, 1982, Sass et al., 1983, Walter & Morse, 1984; Bischoff et al., 1987, Redfern et al., 1989, Bischoff et al., 1993, Navrotsky, 2004; Morse & Lüttge, 2007; Gebauer at al., 2008, Gebauer & Cölfen, 2011, Radha & Navrotsky, 2013), while at higher pressures aragonite forms the stable Ca-carbonate polymorph (Redfern et al., 1989, Radha & Navrotsky, 2013). Accordingly, calcite crystallizes from aqueous solutions below ca. 50 °C (if no calcite-inhibitors are present). However, even in pure $Ca^{2+}/HCO_3^-$ solutions, at temperatures above ca. 50 °C metastable aragonite rather than calcite is obtained (Kitano et al. 1962; Taft, 1967, Ogino et al. 1987). There is no sharp tipping point but rather a gradual change of fraction of the precipitating phases (Ogino et al., 1987, Balthasar and Cusack, 2015). Further, inhibitors of calcite nucleation and/or growth decrease the temperature of this regime shift in precipitation even further, where in marine and diagenetic



environments the most important inorganic inhibitor is $Mg^{2+}$ (Kitano et al., 1972; Katz, 1973; Berner, 1975; Morse
et al., 1997; Choudens-Sanchez, 2009; Radha et al. 2010, Balthasar and Cusack, 2015; Sun et al., 2015).
The replacement reaction of aragonite to calcite in aqueous systems was investigated by Metzger & Barnard
(1968), Bischoff & Fyfe (1968), Bischoff (1969), Katz (1973), Kitano et al. (1972, Yoshioka et al. (1986), Oomori
et al. (1987), and more recently by Perdikouri et al. (2011, 2013). It was recognized by Fyfe & Bischoff (1965)
that the aragonite to calcite replacement reaction in aqueous environments occurs by dissolution and reprecipitation
reactions. Except for Metzger & Banard (1968) and Perdikouri et al. (2011, 2013), most authors used powdered
samples of natural or powdered synthetic aragonite For these powdered samples, they claim a rapid replacement
reaction of aragonite to calcite within hours or very few days at temperatures of ca. 100°C or above, depending on
temperature and the Mg-content of the solution.
Metzger & Banard (1968) and Perdikouri et al. (2011, 2013) investigated aragonite blocks or single crystals and
report that temperatures IN EXCESS of 160-170 °C are required to transform the aragonite to calcite within a
couple of days, whereas BELOW 160 °C aragonite remains present over many weeks.
The present study describes first experimental data of the replacement reaction of BIOGENIC aragonite to non-
biogenic calcite and investigates the kinetics of the replacement reaction of aragonite to calcite in shell material,
geochemistry, nano- and microstructure alteration, and crystallographic texture variation. During
biomineralisation living organisms create local micro-environments for physiological generate of their composite
hard tissues. After the death of the organism all tissues become altered by equilibration with the surrounding
environment - part of the complex set of processes called diagenesis. Thus, as diagenetic alteration proceeds, the
species-specific fingerprint of the biogenic structure disappears and is replaced by inorganic features. Despite the
fact that the evolutionary line of *A. islandica* dates back to the Jurassic (Casey, 1952) only a limited number of
studies have dealt with pre-Neogene *A. islandica* specimens due to the thermodynamically unstable nature of their
aragonitic shells. The aim of the present paper is to describe analysis-based detailed microstructural, geochemical,
phase, and texture data observed in the experimental simulation of diagenesis by hydrothermal treatment of modern
*A. islandica* shell samples. With this study we gain quantitative insight into processes that take place along
pathways from early marine porewater diagenesis to the pervasive recrystallisation under burial conditions. The
targets of the present study are the analysis of microstructural features, the preservation of the organic matrix in
the shell, and the kinetics of the replacement reaction of aragonite to calcite as investigated by X-ray diffraction,
SEM, and crystallographic microanalysis determined by Electron Backscatter Diffraction (EBSD).

## 2 Materials and Methods

### 2.1 Test materials

For this study, shells of *A. islandica* were collected from the recent shell middens of a fishing company in northern
Iceland and from Loch Etive waters in Scotland. On average, shells were between 8 and 10 cm in size and represent
adult specimens. Major morphological features of the shell of *Arctica islandica* are displayed in Fig. A1, see also
Schöne (2013).



**2.2 Methods applied**
**2.2.1 Organic matrix preparation by selective etching**
To image the organic matrix in modern reference shells and hydrothermally altered shell samples as well as the
mineral in reference, geologic, non-biological aragonite, shell or mineral pieces were mounted on 3 mm thick
cylindrical aluminium rods using super glue. The samples were first cut using a Leica Ultracut ultramicrotome
with glass knifes to obtain plane surfaces within the material. The cut pieces were then polished with a diamond
knife (Diatome) by stepwise removal of material in a series of 20 sections with successively decreasing thicknesses
(90 nm, 70 nm, 40 nm, 20 nm, 10 nm and 5 nm, each step was repeated 15 times) as reported in Fabritius et al.
(2005). The polished samples were etched for 180 seconds using 0.1 M HEPES (pH = 6.5) containing 2.5 %
glutaraldehyde as a fixation solution. The etching procedure was followed by dehydration in 100 % isopropanol 3
times for 10 seconds each, before the specimens were critical-point-dried in a BAL-TEC CPD 030 (Liechtenstein).
The dried samples were rotary coated with 3 nm platinum and imaged using a Hitachi S5200 Field Emission-
Secondary Electron Microscope (FE-SEM) at 4 kV.

**2.2.2 Hard tissue characterization methods**
For FE-SEM and Electron Backscatter Diffraction (EBSD) analyses 5 x 5 mm thick pieces were cut out of the
shell and embedded in epoxy resin. The surface of the embedded samples was subjected to several sequential
mechanical grinding and polishing steps down to a grain size of 1 µm. The final step was etch-polishing with
colloidal alumina (particle size ~ 0.06 µm) in a vibratory polisher. For EBSD analysis the samples were coated
with 4-6 nm of carbon, and for SEM visualisation and EPMA analyses with 15 nm, respectively. EBSD
measurements were carried out on JEOL JSM 6400 field emission SEM, equipped with a Nordlys EBSD detector.
The SEM was operated at 20 kV and measurements were indexed with the CHANNEL 5 HKL software (Schmidt
and Olesen, 1989; Randle and Engler, 2000). Information obtained from EBSD measurements is presented as band
contrast images, and as colour-coded crystal orientation maps with corresponding pole figures.
The EBSD band contrast the signal strength of the EBSD-Kikuchi diffraction pattern and is displayed as a grey-
scale component of EBSD scanning maps. The strength of the EBSD signal is high when a crystal is detected
(bright), while it is weak or absent when a polymer such as organic matter is scanned (dark/black).
Co-orientation statistics are derived from pole figures obtained by EBSD scans and are given by the MUD
(multiple of uniform (random) distribution) value. The MUD value is a measure of crystal co-orientation (texture
sharpness) in the scanned area. A high MUD values indicate a high crystal co-orientation (in this study calcite),
whereas low MUD values reflect a low to random co-orientation, respectively.
In order to trace the infiltration and percolation of fluids into and through the shells, pristine and hydrothermally
altered shell samples were scanned with EPMA (Goetz et al., 2014). Chemical data were obtained by using a
CAMECA SX100 EPMA system equipped with a $LaB_6$ cathode. An accelerating voltage of 15 keV at a current of
40 nA were used as operative settings. All elements were analysed with wavelength-dispersive X-ray
spectrometers. The Sr-K$\alpha$, Mg-K$\alpha$, and Na-K$\alpha$ were measured on a TAP crystal. Ca-K$\alpha$, and Ba-L$\alpha$ were measured
on a PET crystal, whereas K$\alpha$ emission lines of P, and Cl were measured on a LPET crystal. L$\alpha$ emission lines of
Mn, and Fe were detected with a LLIF crystal. A step size in the range of 1-2 µm with a dwell time of 150 ms was



chosen for the element mappings. Celestine (Sr), dolomite (Ca, Mg), ilmenite (Mn), apatite (P), albite (Na),
benitoite (Ba), vanadinite (Cl), and hematite (Fe), and were used as standard materials. Matrix correction was
carried out using the PAP procedure (Pouchou and Pichoir, 1984).


**2.2.3 Alteration experiments**
Hydrothermal alteration experiments mimicked burial diagenetic (and meteoric) alteration of recent *A. islandica*
under controlled laboratory conditions. Chemical and isotopic compositions of experimental fluids are given in
Table 1. All fluids for this were spiked with $^{18}$O-depleted oxygen in order to trace fluid-solid exchange reactions.

Pieces (2 x 1 cm) of recent *A. islandica* specimens were placed in a PTFE liner together with 25 mL of either the
meteoric (10 mM NaCl aqueous solution) or burial fluid (100 mM NaCl + 10 mM MgCl$_2$ aqueous solution) and
sealed with a PTFE lid. Each of the PTFE liners was placed in a stainless steel autoclave, sealed and kept in the
oven at temperatures between 100 °C and 175 °C for different periods of time between one day and 84 days (see
Table 2). Obviously, this temperature regime is far beyond natural meteoric diagenetic environments but are typical
for the burial realm. Nevertheless, elevated fluid temperatures were applied to meteoric experiments, too, as
reaction rates under surface conditions are too slow for experimental approaches. After the selected time period,
an autoclave was removed from the oven, cooled down to room temperature and then opened. The aqueous fluid
that had passed through a 0.2 μm cellulose acetate filter was subjected to further chemical and isotopic analyses.
Recovered solids were dried at 40 ºC overnight.


**2.2.4 Chemical analysis of experimental fluids**
Aqueous fluid concentrations were measured before and after the experiment by Inductive-Coupled Plasma
Optical Emission Spectrometry (ICP-OES) using a Perkin Elmer Optima ICP-OES 4300 at TU Graz (Austria)
with an uncertainty of ± 3%. The total fluid alkalinities were determined by a Schott TitroLine alpha plus titrator
with an uncertainty of ± 2%. The detection limit of this apparatus lies in the range of 5 x10$^{-6}$ M. The pH
measurements were performed with a Schott Blueline 28 combined electrode, calibrated with NIST standard
buffers at pH of 4.01, 7.00, and 10.00 with an uncertainty of ±0.07 units at room temperature.


**2.2.5 X-ray diffraction analysis**
X-ray diffraction analysis of pristine and hydrothermally treated samples was performed with Mo-Kα$_1$-radiation
in transmission geometry and with Cu-Kα$_1$-radiation in reflection geometry on a General Electric Inspection
Technologies XRD3003 X-ray diffractometer with an incident-beam Ge111 focussing monochromator and a
Meteor position-sensitive detector. The diffractograms were analysed by Rietveld analysis with the software
package FULLPROF (Rodriguez-Caravajal, 2001) using the aragonite structure data of Jarosch & Heger (1986)
and calcite structure data of Markgraf & Reeder (1985).



**3 Results**
**3.1 The shell ultrastructure of modern *Arctica islandica***
Figures 1 to 5 show characteristic ultrastructural features of the shell of modern *A. islandica*. Images of the pristine
shell are given in Figs. 1-3, while Figs. 4 and 5 present structural features of the hydrothermally altered shells. The
valve of *A. islandica* is layered, with various shell parts showing different internal structural features (Fig. 1). The
distribution patterns of porosity, pore sizes and the dimensions of basic aragonitic crystal units vary significantly
along the cross-section of the shell. The shell portion facing seawater, indicated with yellow stars in Figs. 1A and
1B, consists of aragonite crystal units in the 5 μm size range (Fig. 2A). This shell portion is highly porous (Fig.
1B) with pore diameters in the range of a few (< 5) micrometres. The inner shell portion, i.e., the part very close
to the soft tissue of the animal (indicated with white rectangles in Figs. 1A, 1C), is dense and is composed of very
small aragonite crystallites with sizes of less than 1 μm (Fig. 2B) and contains very few pores. The dimension of
pores in this shell region is in the 1 to 2 μm range. However, the innermost shell layer, the layer that is in contact
with the mantle tissue of the animal (white stars in Figs. 1A, 1C) contains large (up to 12 micrometre diameter)
and elongated pores that are oriented perpendicular to the rim of the shell (see white arrows in Fig. 1C). Growth
lines are clearly visible in the cross-section through the shell (white arrows in Fig. 1A) as thin layers characterised
by higher accumulations of organic material (this study and Richardson, 2001).
Figures 2, and 3 show, at increasing magnification, structural features of modern *A. islandica* shells that were made
visible by slight etching of the mineral and simultaneous chemical fixation of the organic matrix. Structural
characteristics of the reference material (non-biologic aragonite grown from solution), treated chemically in a
similar way as the biogenic aragonite samples, are shown in the appendix, in Fig. A2. Fig. 2A shows features that
are characteristic of the seawater-adjacent side of the shell, whereas Fig. 2B depicts internal characteristics of the
tissue-adjacent side of the shell (the region marked by white rectangles in Fig. 1).  Etching brings out the outlines
of the aragonite grains, revealing the fabric of the biopolymer matrix within the hard tissue and its interlinkage
with the mineral. The mineral units (crystals) in the outer shell layer are highly irregular in shape with dimensions
in the 1-5 μm range (Fig. 2A). In contrast, although the mineral units (crystals) in the dense layer of the shell also
have irregular morphologies, they are of significantly smaller dimensions, mainly in the 1-2 micrometre range and
below (Fig. 2B). The predominant fabric of the organic matrix in the shell of *A. islandica* is a network of
intracrystalline fibrils (Fig. 3, yellow arrows in Figs. 3A, 3B) that interconnect the mineral units across the grain
boundaries. However, organic membranes are occasionally also present and surround the mineral units (white
arrows in Fig. 3A). Like all other biological carbonate hard tissues, at the finest scale, the shell of *A. islandica* is
composed of nanoparticles that are a few tens of nanometres in diameter (white arrows in Fig. 3C). These are co-
aligned to form mesocrystals - here in the 1-5 μm size range.



**3.2 The ultra-, and microstructure of experimentally altered *A. islandica* shells**
In order to trace fluid infiltration into and their percolation through the shell we performed major and minor
element chemical analyses by EPMA. The distribution patterns of sodium, chlorine and strontium are shown as



characteristic examples (Figs. A3, A4, A5). Fluids enter the shell through pores and along growth lines, as
demonstrated by the perfect correspondence between increased Na, Cl contents and the outlines of annual growth
lines, indicated by elevated Sr contents (Fig. A3). These growth lines are readily detected by an increase in Sr
contents in pristine (Figs. A3A) as well as in hydrothermally altered shell samples (Figs. A4, A5, see also Shirai
et al. 2014). However, neither the temperature of hydrothermal alteration, nor the chemistry of the alteration fluid
has an influence on the amount of Sr present along growth lines. Relative to neighbouring shell increments, the Sr
content along the growth lines is always higher. Maximal concentrations (along annual growth lines) in pristine
and altered shells vary between 0.4 and 0.6 wt% Sr (Figs. A3, A4, A5).
FE-SEM images of Figs. 4 and 5 highlight the grain structure and remnants of the organic matrix in hydrothermally
altered *A. islandica* shells. In the case of the samples shown in Figs. 4 and 5, burial water was used as alteration
solution; the hydrothermal treatment conditions were 100 °C for 28 days (Figs. 4A, 4B, 5A, 5B) and 175 °C for 7
days (Figs. 4C, 4D, 5C, 5D), respectively. SEM images on the left hand side of Figs. 5 and 6 are taken from the
shell section adjacent to seawater, while SEM images on the right hand side of Figs. 4 and 5 are taken from the
dense layer of the shell very close to the soft tissue of the animal. Alteration at 100 °C for 28 days did not change
the internal ultrastructure of the shell significantly. The shape and size of the mineral units are retained and they
are still interconnected with a few organic fibres (Figs. 4A, 4B, 5B). However, at 175 °C for 7 days, the formerly
present network of biopolymer fibres and membranes has vanished completely (Figs. 4C, 4D, 5C, 5D). At higher
magnification a multitude of tiny holes (indicated with yellow arrows in Figs. 5C, 5D) become readily visible. In
the unaltered shell these holes were filled with the network of biopolymer fibrils interconnecting the mineral units
(e.g. Fig. 3B). The tiny holes in the mineral units start to become visible even in the samples altered at 100 °C
(yellow arrows in Fig. 5B). Although at 175 °C shell aragonite has transformed to large calcite crystals (see
following the description of results), etching still outlines a grain fabric on the size-scale of the former bioaragonite
crystal units (Figs. 4C, D). The newly formed fabric resembles that of a fine-grained inorganic ceramic material.
Aragonite crystal orientation patterns of modern *A. islandica* shells and those altered at 100 °C are presented in
Figs. 6, A6, A7 with EBSD grey-scale band contrast images (upper images of Figs. 6A, 6B, 6C, A6), EBSD colour-
coded orientation maps (lower images of Figs. 6A, 6B, 6C), and corresponding pole figures. Fig. 6E gives grain
area information deduced from the EBSD measurements that are shown in Figs. 6A to 6C. Alteration occurred at
100 °C, over a period of 28 days, and took place in meteoric (Fig. 6B) and burial fluid (Figs. 6C, A6), respectively.
The microstructure and texture of pristine *A. islandica* shell material is shown in Fig. 6A. The crystallographic co-
orientation in pristine and altered *A. islandica* shells is axial with the c-axes (setting a = 4.96 Å, b = 7.97 Å, c =
5.74 Å, space group *P*mcn) pointing approximately perpendicular to the growth lines. Co-orientation of the
aragonite crystallites in the shell portion adjacent to seawater, even in the modern *A. islandica*, is very low with
Multiple of Uniform Random Distribution (MUD) values of 12 (Fig. 6A) and 32 (Fig. A7A). Hydrothermal
treatment of *A. islandica* at 100 °C does not produce a significant change in aragonite co-orientation pattern,
texture, grain fabrics, and grain size distributions. The pristine and the hydrothermally treated shell materials
appear to be quite similar. The small changes in MUD values must be attributed to fact that it was impossible to
locate the EBSD scan fields on the different samples in exactly corresponding spots with respect to the outer shell
margin and to the patterns of growth lines. The shell is not uniformly textured. In particular, the slight preferred
crystallographic orientation of the a*-axes (the (100) plane normal) in Fig. 6C is a singular case, while c-axis



preferred orientation is otherwise dominant (Fig. 6C). Figs. 7, A6B, A6C show  microstructure and texture
characteristics deeper within the shell (Figs. 7A, A6, A7) and at the innermost margins next to the soft tissue of
the animal (Figs. 7C, 7D; alteration in meteoric fluid: Figs. 7A to 7D; alteration in burial fluid: Figs. 7E, 7F). In
the EBSD band contrast map of Fig. 7A we clearly see the change in microstructure from the distal (seawater
adjacent) shell layer with the larger aragonite crystals (yellow star in Fig. 7A) to the inward shell portion where
aragonite crystals become small to minute (white star in Figs. 7A, A6B, A6C). As the pole figures and MUD
values demonstrate, the axial c- and a-axes co-orientation increases gradually towards the inner shell layer (the
layer close to the soft tissue of the animal) where MUD values of almost 100 are reached (Figs. 7D, 7F, A6, A7).
Using X-ray diffraction (XRD) we obtained an overview of the kinetics of the *A. islandica* biogenic aragonite to
calcite transition under hydrothermal conditions up to 175 °C in artificial burial solution (Figs. 8A, 8B, A8). A
representative Rietveld-plot of the analysis of the XRD data obtained for the six days alteration is given in Fig.
A9. As Fig. A6 demonstrates, experiments below 175 °C show no signs of a replacement reaction of bioaragonite
to calcite in the XRD bulk measurements. At 175 °C in burial solution, calcite formation starts after a passive
period of about 4 days (Figs. 8A, 8B A8) and then proceeds rapidly. After 7 days only a few patches of aragonite
in the dense shell layer are not yet completely transformed to calcite (as seen in the EBSD investigations, unaltered
shell portions are indicated with white rectangles in Fig. 1A). After 8 days the transition to calcite is complete.

EBSD data clearly show that after a hydrothermal treatment at 175 °C, with either, meteoric or burial fluid, shell
aragonite is transformed to calcite (Figs. 9, 10 and 11). In the outer shell layer the replacement reaction to calcite
is complete with the development of large crystal grains, some reaching sizes of hundreds of micrometres (see
EBSD maps in Figs. 9 and 10). In contrast, dense shell regions devoid of pores still retain patches of the original
aragonitic microstructure and texture (coloured EBSD maps in Figs. 11A, 11B). The MUD values for the newly
formed calcite material are high (Figs. 9, 10), but this is related to the fact that within the range of the EBSD scan
just a small number of large, newly formed, individual crystals is encountered.  Figure 11 shows shell regions
where patches of aragonite have survived which contain first-formed calcite. Calcite nucleation sites are the
locations where the hydrothermal fluid has access to the shell: at its outer and inner surfaces (yellow stars in Fig.
11B) and at growth lines (yellow arrows in Fig. 11A). Fig. 11A demonstrates how calcite crystals form strings
along linear features, which correspond to growth lines in the pristine shell material.


**4 Discussion**
In sedimentary environments replacement reaction of metastable biogenic aragonite and high-Mg calcite to
inorganic low Mg-calcite occurs mainly through the following processes: (1) shell dissolution followed by
subsequent low Mg-calcite formation and/or (2) micro- to nanoscale recrystallization to low Mg-calcite with
preservation of some specific boundaries such as prisms, tablets and fibres in bivalves (Swart, 2015). The
replacement reaction from metastable carbonate phases to a stable low-Mg calcite phase is driven by the difference
in solubility (free energy) between aragonite and calcite (Plummer & Busenberg, 1982). Thus, as the replacement



reaction proceeds, the percolating diagenetic pore fluid is undersaturated with respect to aragonite but is saturated
with respect to calcite. However, the replacement reaction first requires a nucleation step, the formation of the first
calcite crystallites larger than the critical size (Morse & Lüttge, 2007), where the bulk energy gained by further
growth exceeds adverse energy contributions such as increasing interface energies and strain energies arising from
increasing the size of the calcite crystals.


**4.1 Characteristics of the grains obtained by reaction at 100 °C and 175 °C**
Our laboratory-based hydrothermal alteration experiments were carried out at 100 °C in both meteoric and burial
fluids. At 100 °C, both, the aragonite mineral as well as the characteristic biological microstructure, survive the
hydrothermal treatment up to at least 28 days. This is consistent with the findings of Ritter et al. (2016) who
analysed the light stable isotope signatures ($\delta^{13}C$, $\delta^{18}O$) of our hydrothermally treated samples. In the 100 °C
alteration experiments using isotope-doped hydrothermal fluids, Ritter et al. (2016) found that the carbon and
oxygen isotope ratios of the treated shells are within the same range as those measured in the pristine samples.
Furthermore, no obvious patterns emerge from the comparison of sub-samples exposed to seawater, meteoric, and
burial fluids. Most of the extensive literature on aragonite precipitation from aqueous solutions and aragonite-
calcite replacement reactions in aqueous environments, as reviewed in the introduction, makes clear that both
temperatures around the boiling point of water and the presence of $Mg^{2+}$ inhibit calcite nucleation. Thus, the
inhibition of calcite nucleation favours the growth of aragonite if the solution is supersaturated with respect to the
Ca-carbonate phases. If supersaturation is exceedingly high and rapidly generated, vaterite or even amorphous
calcium carbonate will precipitate and reduce the supersaturation below the levels required for aragonite or calcite
nucleation (Gebauer et al., 2008, 2012; Navrotsky, 2004; Radha et al., 2010). However, it is unlikely that these
levels of supersaturation are reached in our case, as aragonite is already present. We, thus, conclude that the
absence of aragonite to calcite replacement reaction in our 100 °C treatments is related to inhibition of calcite
nucleation (Sun et al., 2015), a mechanism that has rarely been rigorously explored.
At 175 °C the replacement reaction of biological aragonite to coarse-grained calcite occurs rapidly; it starts after
a dormant period of about 4 days and proceeds rapidly almost to completion after 3 more days (Figs. A3, 13A).
Calcite nucleation occurs (and replacement reaction proceeds) where the hydrothermal fluid is in contact with the
bio-aragonite: at the surfaces of the shell, in pores and along growth lines (Figs. 7, 8, 9).


**4.2 The time lag of aragonite to calcite replacement reaction at 175 °C**
The several-day dormant period followed by the rapid growth of calcite indicates that the nucleation of calcite is
inhibited, at least initially. Inhibition of calcite nucleation occurs by the presence of magnesium in the solution
and/or by temperatures between 70 °C and 160 °C even without Mg (Kitano et al., 1962; Taft, 1967; Kitano et al.,
1972; Katz, 1973; Berner, 1975; Morse et al., 1997; Choudens-Sanchez, 2009; Radha et al. 2010, Balthasar &
Cusack, 2015; Sun et al., 2015). A possible scenario explaining the dormant period could be that, despite inhibition,
there is still a small, but non-zero nucleation rate of calcite, which permits the formation of individual nuclei after
4 days, which then grow rapidly. Another, perhaps more likely scenario is the initial, rapid formation of a



passivation layer on the surface of the aragonite or on the surface of any calcite nuclei; the dormant period is then
the time that is needed to dissolve this passivation layer, at least in some places, where subsequently calcite nuclei
of critical size can form.
In meteoric solutions the grain size of the newly formed calcite reaches 200 micrometres (e.g. Figs. 9C) while in
the Mg-bearing burial solution newly formed calcite crystals reach sizes in the 1 mm range, in both, the 7- and 84-
day treatments (e.g. Figs. 10B, 10C). The large grains exceeding 200 micrometres suggest an Ostwald-ripening or
strain-driven grain growth. The latter is potentially due to the 8.44 % volume increase when the denser aragonite
transforms to calcite. The determination of *local misorientation* from an EBSD data set gives small orientation
changes and visualizes regions of deformation of the crystal structure, thus, the local misorientation parameter is
an indication of internal strain or stress in the material. Figure 12 depicts the distribution pattern of local
misorientation within five selected EBSD maps (Fig. 12B, 12E, 12H, 12K, 12N). Legends accompany all local
misorientation maps (Figs. 12C, 12F, 12I, 12L, 12O). A high degree of internal strain is indicated by blue colours,
while light green to yellow colours highlight shell portions where local misorientation is low. Grains in Fig. 12 are
defined by a *critical misorientation* value set at 5° (i.e. tilts smaller than 5° are counted as mosaic block
boundaries), thus, a tilt of five degrees has to be present for the distinction between two adjacent grains rather than
a single grain with small angle grain boundaries < 5° misorientation. For the better visualization of individual
grains we outlined these with white lines and show them in random colours (Figs. 12A, 12D, 12G, 12J, 12M). As
it is well visible from the white 'dots' in Figs. 12A, 12D, 12G, 12J, 12M, all calcite grains contain numerous small
calcite crystallites, a clear cause for the occurrence of internal strains. In all five investigated data sets local
misorientation goes up to 2 degrees, thus, neither alteration time, nor the chemical composition of the used
alteration solution exerts a major impact to the degree of strain accumulation within the EBSD maps (and grains).
Figure 13 shows in color-coded maps the pattern of strain/stress accumulation distribution in two large calcite
grains that grew at 175 °C in Mg-bearing burial solution. Corresponding legends are given below the grains. In
the case of the grain shown in Fig. 13A, obtained at an alteration that lasted for 7 days, (see also Figs. 12J and
12K, the white star in 12K indicates the location of the grain within the map) some portions of the crystal are
strain/stress-free (portions with the blue to dark green colours in the map shown in 13A). In contrast, the grain
marked by a yellow star in Figure 13C (and Figs. 12M and 12N), shows high stress/strain accumulations
everywhere within the grain and no strain/stress-free regions. We find that the local misorientations are mainly
curvilinear structures in the cross section (white arrows in Figs. 13A, 13B) and correspond to subgrain boundaries
within the newly formed calcite crystals. These boundaries do not appear to heal or to disappear with an increased
alteration time, an indication again of the little effect of alteration duration on the fabric and internal structure of
calcite grains crystallized from *Arctica islandica* shell bioaragonite.
To investigate grain growth patterns further, we took a statistical approach based on the EBSD measurements
shown in Figs. 9 and 10 (alterations experiments carried out for 7 and 84 days at 175°C in meteoric and burial
solution, respectively). Figs. 14A and 14B show the statistics of grain area (we define a grain by a critical
misorientation of 5 °) versus degree of *mean* misorientation within a grain. On the basis of our measurements and
observations, we do not see major evidence for a specific calcite grain growth phenomenon with an increase in
alteration time, as we expected for the 7 and 84 days alteration series. However, we find that experiments
conducted with the Mg-containing burial solution yield larger calcite crystals (black arrows in Fig. 14B) in



comparison to the size of the grains obtained from experiments carried out with meteoric water (Fig. 14A). Grains
obtained from alteration experiments with meteoric fluid show a significantly higher degree of mean
misorientation (up to 10 degrees, black arrows in Fig. 14A), compared to that in grains that grew in burial solution.
We attribute this to the nucleation rate: the crystals growing from each nucleus consume the aragonite educt (the
precursor, original aragonite) until they abutted each other. Thus, larger crystals in the experiment with burial
solution result from a smaller number of calcite nuclei. Again, this supports the idea that $Mg^{2+}$ inhibits calcite
nucleation.


### 4.3 The calcite to aragonite replacement reaction kinetics

Inorganic experiments on aragonite to calcite transition at 108 °C in hydrothermal conditions were reported by
Bischoff & Fyfe (1968) and by Bischoff (1969). These authors used fine-grained powders as educts (the precursor,
original material) and observed a comparatively rapid transition to calcite that was complete within 48 hours,
depending on the composition of the fluid. For example, larger $CO_2$ partial pressure (leading to lower pH and thus
larger solubility of the carbonates) accelerated, while the presence of Mg-ions retarded the process. This rapid
reaction kinetics as reported by Bischoff & Fyfe (1968) and by Bischoff (1969) is discrepant to our observations.
We do not see a replacement reaction of the biogenic aragonite to calcite at 100 °C even within 28 days.
Hydrothermal experiments by Metzger & Barnard (1968) and by Perdikouri and co-workers (2011, 2013),
however, who used aragonite single crystals in their experiments, report reaction kinetics which correspond to our
observations. They do not observe any evidence of the replacement reaction at 160 °C even within 1 month, but a
partial replacement of their aragonite crystals by calcite within 4 weeks at 180 °C. Setting kinetic control aside, it
appears that aragonite loses its metastability between 160 °C and 180 °C, just like the tipping point between 50 to
60 °C (Kitano et al. 1962, Balthasar & Cusack, 2015) above which aragonite precipitates from a solution rather
than the stable calcite phase. We observed that the fluids used (artificial meteoric and/or burial fluids) cause only
a minor difference in replacement reaction kinetics in our experiments, with the $MgCl_2$-bearing artificial burial
fluid reducing the nucleation rate of calcite, thus, leading to the observed significantly larger calcite crystals in the
recrystallised product. As compared to the work of Perdikouri et al. (2011, 2013) on aragonite single crystals,
shell-aragonite does not crack during the replacement of the aragonite by calcite. The reason for this difference
may be ascribed to the porosity of the bioaragonite which results from the loss of its organic component. Figs. 5C
- D and 6C - D illustrate that the (newly formed) calcite product reveals an internal structure that is very reminiscent
of the original bioaragonite/biopolymer composite. The structure arises as the solution penetrates along former
sites of organic matrix (former aragonite grain boundaries), such that the structural features obtained after
alteration still outline the former aragonite grains. Thus, limited grain size of the bioaragonite together with the
formerly biopolymer-filled spaces reduce any stresses that may be built up by the specific volume change of the
$CaCO_3$ during the replacement reaction. The replacement process preserves original morphological features.
Several studies (Putnis & Putnis, 2007, Xia et al., 2009, Putnis and Austrheim, 2010, Kasioptas et al., 2010, Pollok
et al., 2011) experimentally investigated mineral replacement reactions creating pseudomorphs, even reproducing
exquisite structures such as the cuttlebone of *Sepia officinalis*. These studies conclude that the essential factor in





producing pseudomorphs is the dissolution of the replaced parent material as the rate-limiting step, while the
precipitation of the product phase and the transport of solution to the interface must be comparatively fast. The
preservation of morphology - even as observed on the nano- to microscale - is ensured if nucleation and growth of
the product immediately take place at the surface of the replaced material when the interfacial fluid film between
the dissolving and the precipitating phase becomes supersaturated in the product after dissolution of the educt: an
interface-coupled dissolution-reprecipitation mechanism (Putnis & Putnis, 2007). If dissolution of the educt is fast
and precipitation of product is slow, more material is dissolved than precipitated, and the solutes can be transported
elsewhere. This would create not only an increased pore space which potentially collapses under pressure, but the
dissolved material would eventually precipitate elsewhere with its own characteristic (inorganic) morphology.
**4.4. A paleontological perspective of our laboratory-based hydrothermal alteration experiments**
The understanding of the diagenetic processes which control organism hard tissue preservation is very important
in palaeontology as it a prerequisite to taxonomic, taphonomic, palaeoecological, and biostratigraphic studies (e.g.
Tucker, 1990). Most organisms have hard tissues composed of calcium carbonate, and its metastable form,
aragonite, is one of the first biominerals produced at the Precambrian-Cambrian boundary (Runnegar & Bengtson,
1990), as well as one of the most widely used skeleton-forming mineral in the Phanerozoic record and today; in
fact, aragonitic shells/skeletons are characteristic of hyolithids, cnidarians, algae, and the widespread and
diversified molluscs.
Several studies (Cherns & Wright, 2000; Wright et al., 2003; Wright & Cherns, 2004; James et al., 2005) have
underscored that Phanerozoic marine faunas seem to be dominated by calcite-shelled taxa, the aragonitic or
bimineralic groups being lost during early diagenesis (in the soft sediment, before lithification), potentially causing
a serious taphonomic loss. Considering that most molluscs are aragonitic or bimineralic, this loss could be
particularly detrimental both for palaeoecological and biostratigraphic studies. However, it has been shown that
the mollusc fossil record is not so biased as expected (Harper, 1998; Cherns et al., 2008). This is due to high
frequency taphonomic processes (early lithification/hardgrounds, storm plasters, anoxic bottoms, high
sedimentation rates) that produce taphonomic windows allowing molluscs preservation (James et al. 2005; Cherns
et al., 2008).
In this perspective, the laboratory-based hydrothermal alteration experiments performed here offer very interesting
insights into the fate of the aragonitic or bimineralic hard tissues that escape early dissolution during shallow burial
and have the potential to enter the fossil record, a matter relatively neglected so far. In particular, the resistance of
biogenic aragonite to replacement by calcite up to temperature of 175 °C during hydrothermal alteration offers an
additional explanation for the preservation of aragonitic shells/skeletons, besides the taphonomic windows
envisaged by Cherns et al. (2008). The results of our experiments neatly explain the observation that the mollusc
fossil record is good and allows restoration of evolutionary patterns.



### 5 Conclusions


1. Aragonite crystallite size, porosity, and pore size varies across the cross-section of the valve of modern *Arctica*
*islandica*. While the outer shell layer is highly porous, with pore sizes in the range of a few micrometres, and
contains mineral units in the 1-5 μm size range, the inner shell layers (which are closer to the soft tissue of the
animal) are characterised by a dense shell structure with small (1 μm) mineral units and a very low porosity.
The innermost section of the shell is penetrated by elongated pores oriented perpendicular to the shell inner
surface. At annual growth lines Sr contents are always high, relative to shell increments between the growth
lines in both pristine and experimentally altered shell samples. The chemistry of the alteration fluid and the
duration of the alteration experiment do not exert a major effect on the concentration of Sr along the growth
lines.
2. During hydrothermal alteration at 100 °C for 28 days, most but not all of the biopolymer matrix is destroyed,
while shell aragonite and its microstructure are largely preserved.
3. During hydrothermal alteration at 175 °C for 7 days or more, the biopolymer shell fraction is destroyed, such
that pathways for fluid penetration are created. At this temperature shell aragonite is almost completely
transformed to calcite.
4. When meteoric solution is used for alteration, newly formed calcite crystal units reach sizes up to 200
micrometres, while alteration in burial solution induces the formation of calcite crystals that grow up to 1 mm
in 7 days. We attribute the latter, larger grains to the Mg-content of the burial solution, which inhibits calcite
nucleation. The formation of fewer nuclei leads to the growth of larger calcite crystals.
5. Geochemical results show that calcite nucleates and replacement reaction proceeds where the hydrothermal
fluid is in contact with the aragonite: at the two shell surfaces, in pores, and at growth lines, which are thin,
formerly organic-filled layers.
6. The replacement reaction of bioaragonite to calcite does not proceed at temperatures much lower than 175 °C.
At 175 °C we observe a dormant time of about 4 days during which no XRD-detectable calcite is formed. The
replacement reaction then proceeds within 2-3 days to virtually complete replacement reaction, with small
amounts of aragonite still prevailing in the dense, proximal layer of the shell.
7. Between two tipping points, one between 50 and 60 °C, the other between 160 and 180 °C, aragonite appears
to precipitate from supersaturated aqueous solutions rather than calcite, such that the hydrothermal treatments
of aragonite within this temperature bracket do not yield calcite.
8. The absence of aragonite replacement by calcite at temperatures lower than 175°C contributes to explain why
aragonitic or bimineralic shells and skeletons have a good potential of preservation and a complete fossil
record.






**6 Appendix A**

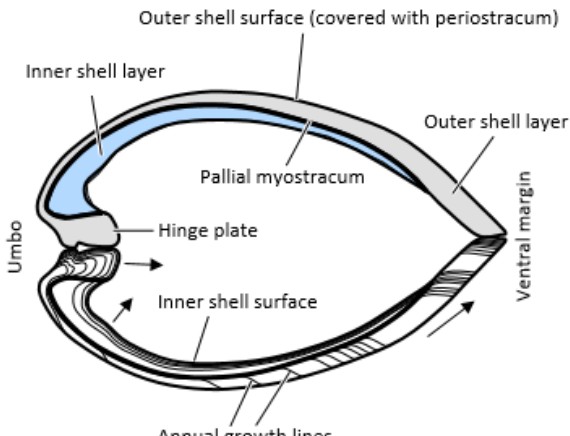


**Appendix Fig. A1.** Morphological characteristics of the shell of the bivalve *Arctica islandica*. A detailed
description is given in Schöne (2013).







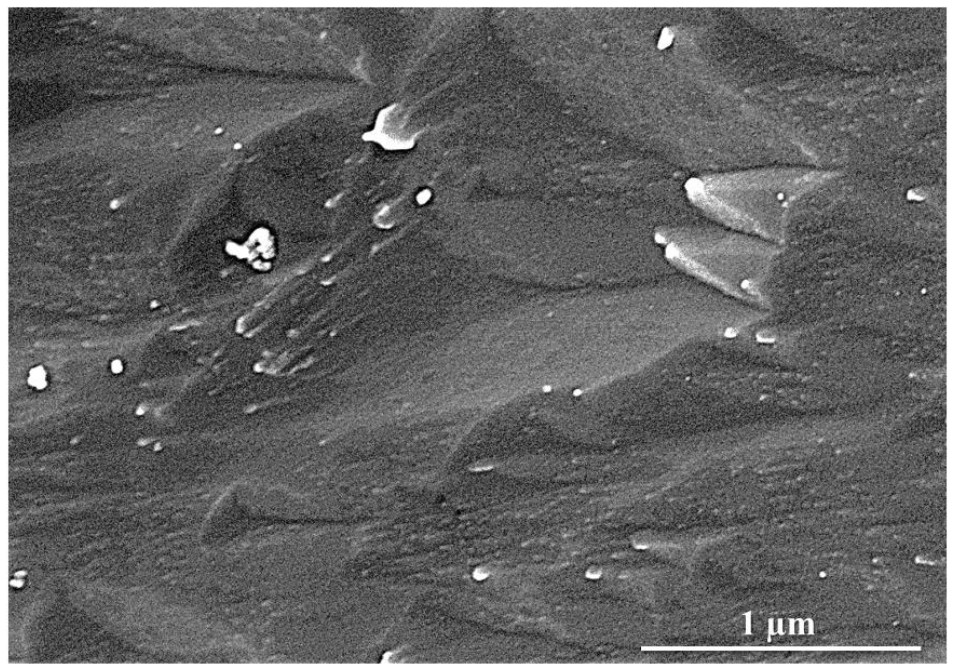


**Appendix Fig. A2.** FE-SEM image of microtome cut, polished, etched and critical-point-dried surface of non-

biologic aragonite grown from solution.





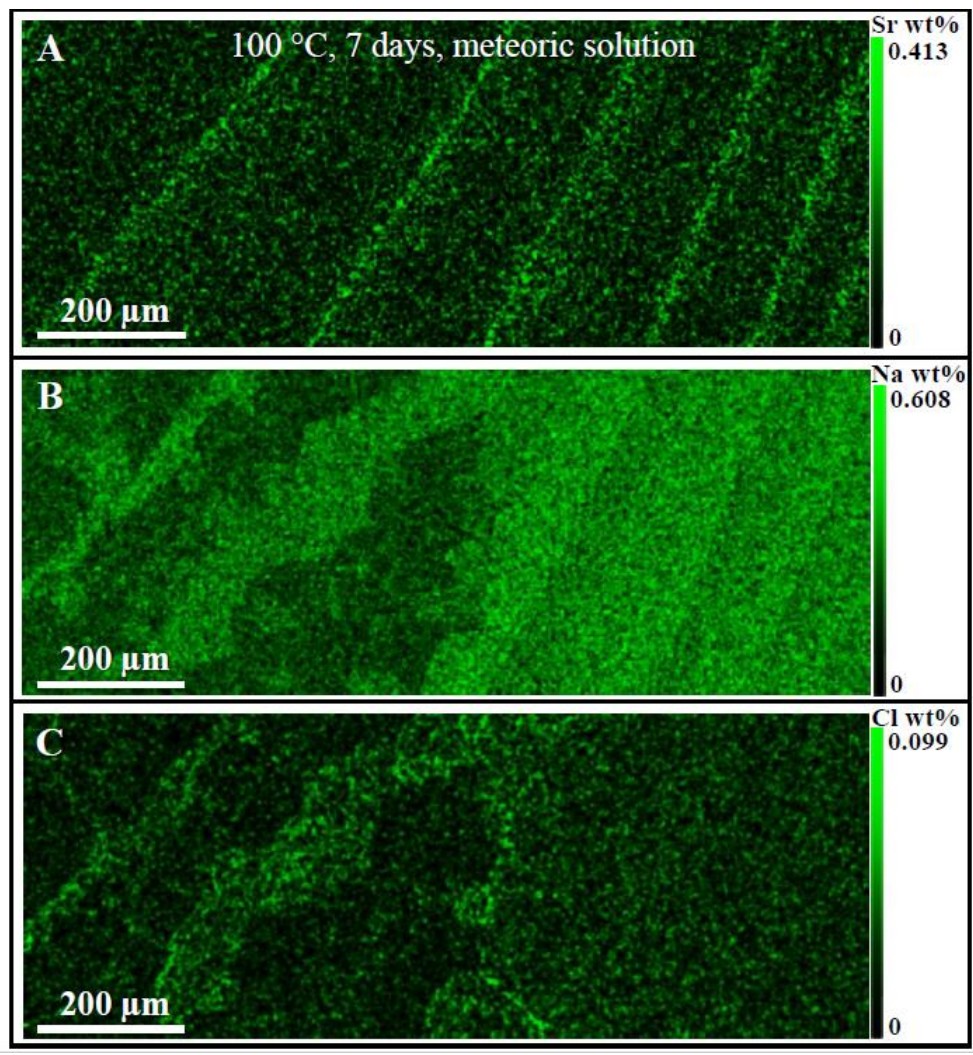


**Appendix Fig. A3.** Sr$^{2+}$, Na$^+$, and Cl$^-$ concentrations along annual growth lines in a hydrothermally altered shell portion of *Arctica islandica*. The alteration fluid is NaCl-rich, i. e., simulating meteoric waters. The degree of fluid infiltration into and through the shell is well traceable with Na+ and Cl- concentrations. Infiltration occurs, in addition through pores, along growth lines that act as conduits for fluid circulation.

548

549

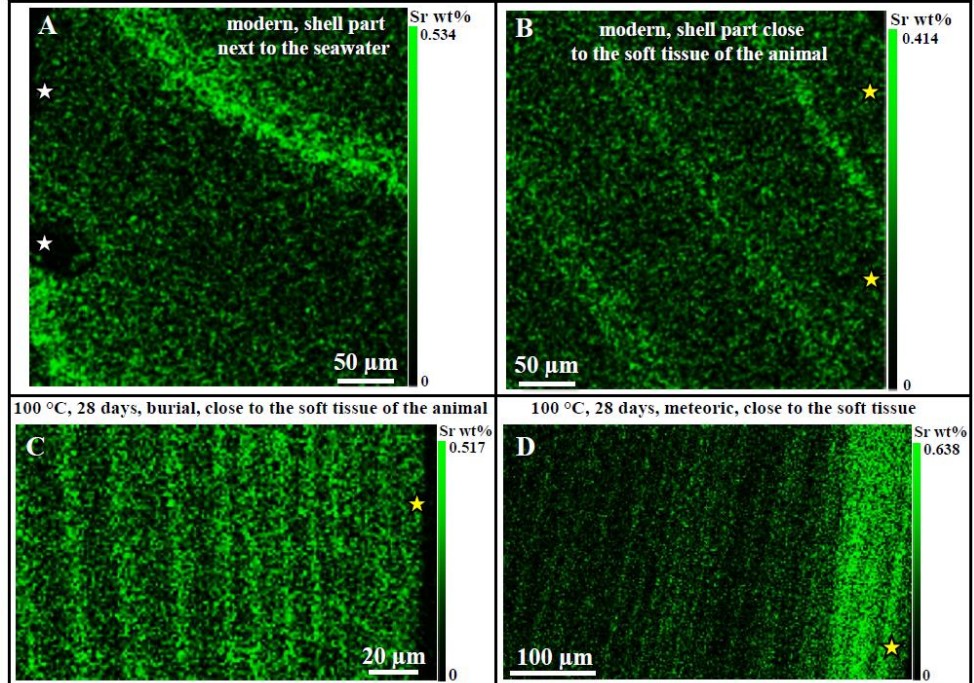

**Appendix Fig. A4.** $Sr^{2+}$ concentrations along annual growth lines in pristine (A, B) and hydrothermally altered (C, D) *Arctica islandica* shell portions. White stars indicate shell regions next or close to seawater, while yellow stars point to the shell parts that are next or close to the soft tissue of the animal. Fluids enter the shell at its two surfaces (see enrichment in $Sr^{2+}$ in Fig. 8D) and, especially along growth lines. Neither the degree of hydrothermal alteration, nor the chemistry of the alteration fluid changes significantly the $Sr^{2+}$ contents along the growth lines. Maximal values for both, pristine and altered samples, range between 0.4 and 0.6 wt% Sr.





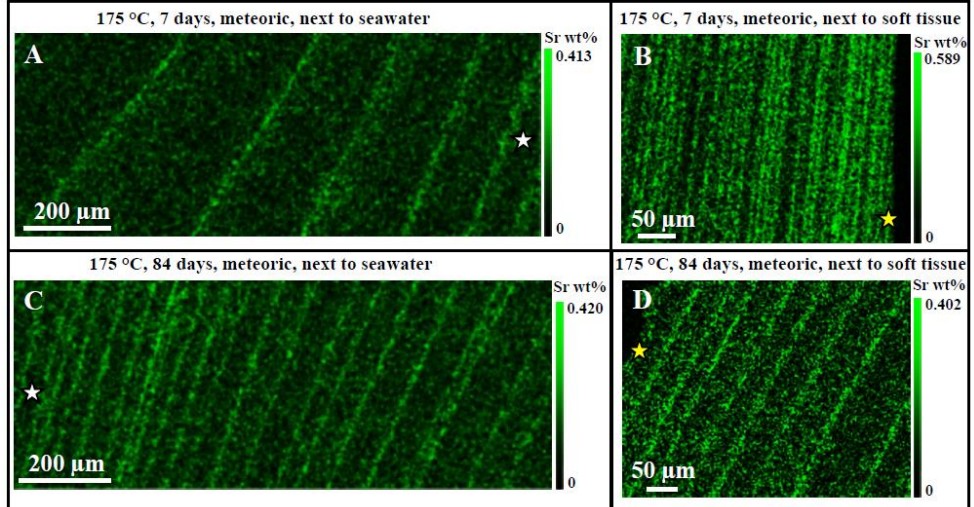

**Appendix Fig. A5.** $Sr^{2+}$ concentrations along annual growth lines in hydrothermally altered *Arctica islandica* shell portions. Alteration temperature was 175 °C; meteoric water was used as alteration fluid; the alteration experiments lasted for 7 and 84 days. $Sr^{2+}$ concentration scatters for both alteration times around 0.4 wt% $Sr^{2+}$ and is similar to the value measured in the pristine *Arctica islandica* reference samples (see Figs. 8A, 8B).





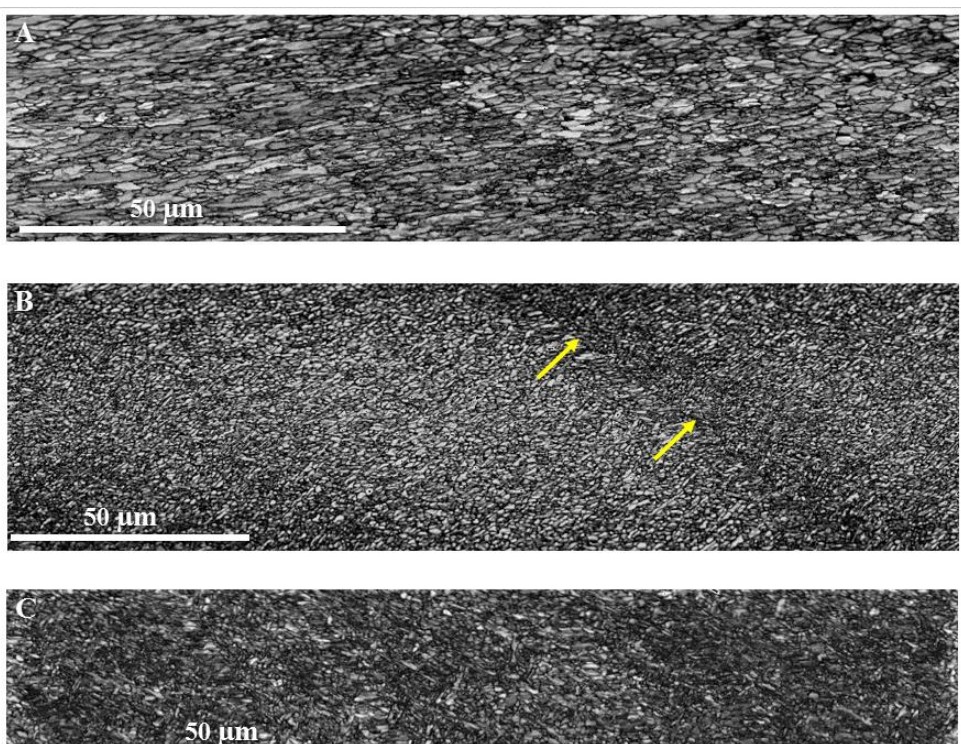

566

**Appendix Fig. A6.** EBSD band contrast images taken along a cross section from different parts of the shell of

pristine *Arctica islandica*. (A) Shell region next to seawater, (B) central shell section, (C) shell layer next to the

soft tissue of the animal. Well visible is the difference in crystallite size. In contrast to the shell layer close to

seawater (A), the innermost shell section, the section close to the soft tissue of the animal, is highly dense and

consists of minute aragonite crystals.

572





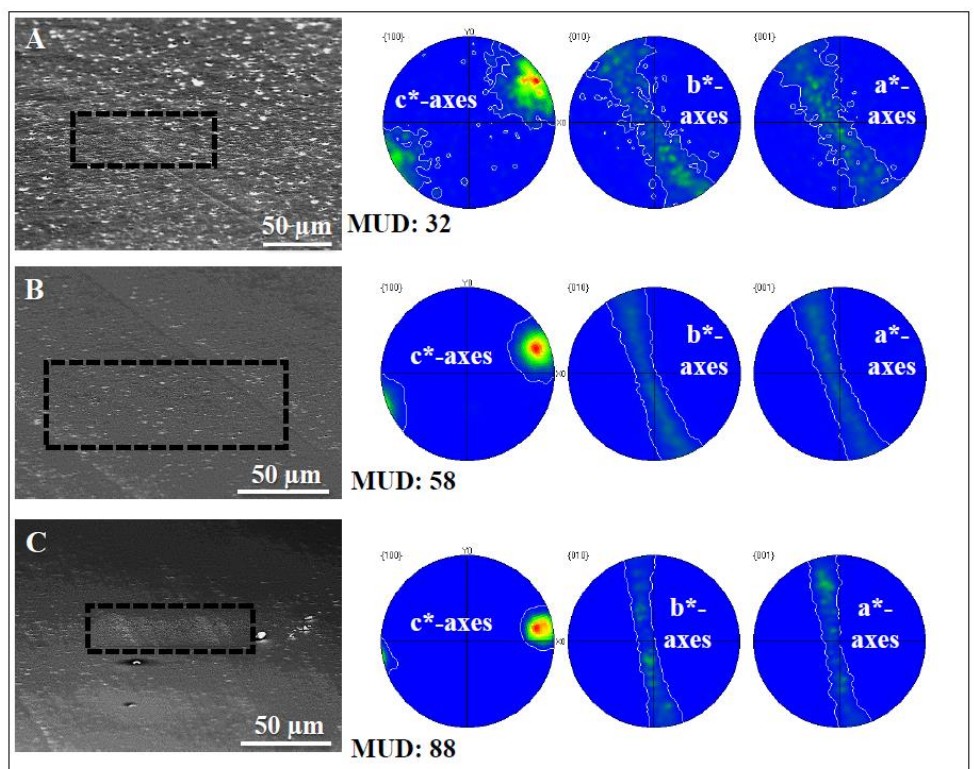

573

**Appendix Fig. A7.** Pole figures obtained from EBSD measurements shown in Figure A6. Measurements are done on pristine *Arctica islandica*. SEM images on the left hand side indicate the location of EBSD maps; (A) shell layer facing seawater, (B) central shell portion, (C) shell part close to the soft tissue of the animal. The pole figures and MUD values indicate clearly that aragonite co-orientaion increases significantly towards innermost shell sections.

579



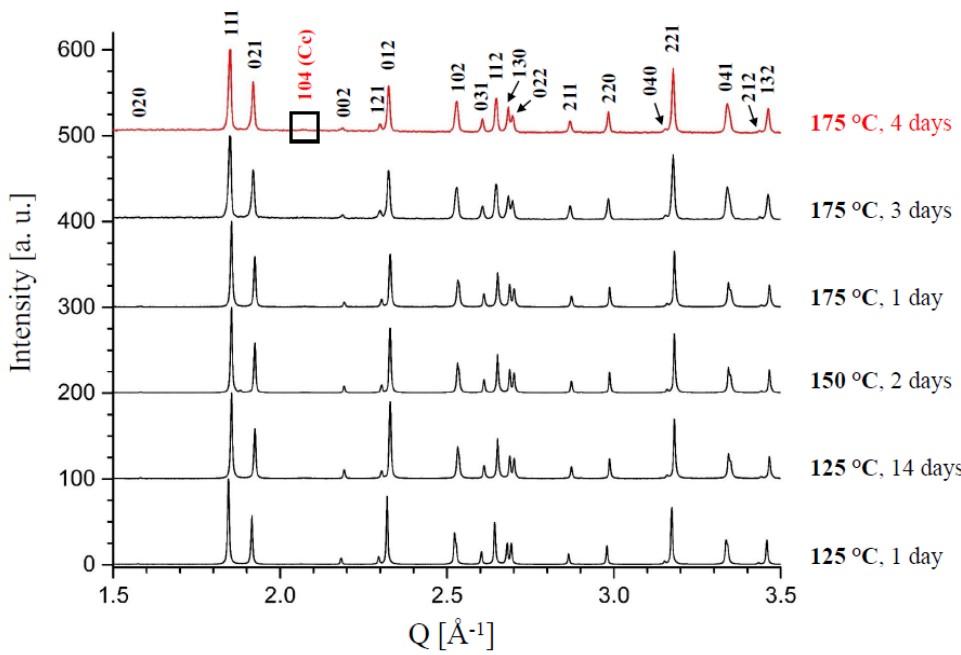

**Fig. A8.** XRD measurements of experimentally altered *Arctica islandica* samples subjected to alteration temperatures between 125 °C and 175 °C for various lengths of time (1, 2, 3, 4 and 14 days). Calcite formation starts at 175 °C and an alteration time of four days. Red Miller indices (Cc): calcite and black Miller indices: aragonite.



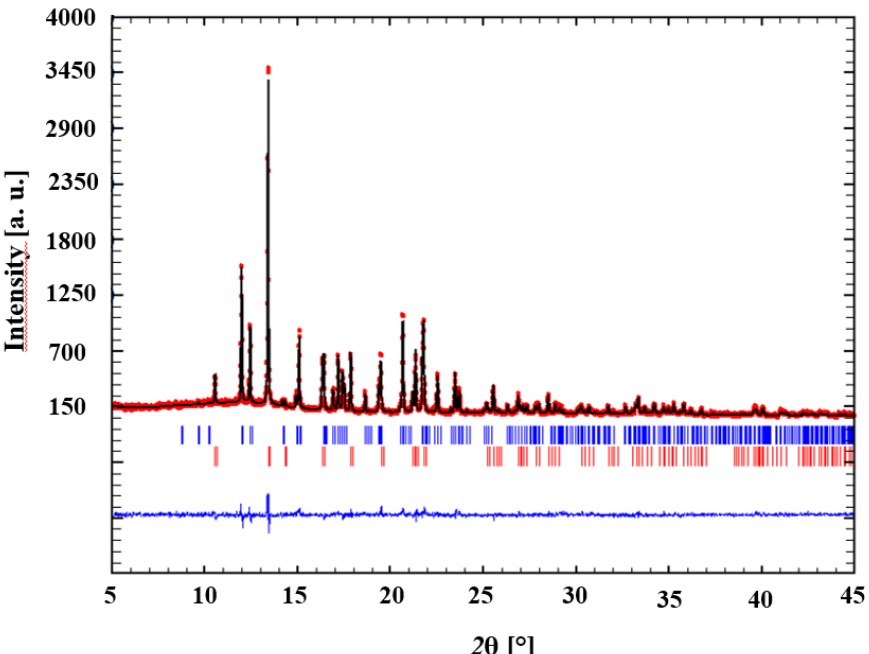

**Appendix Fig. A9.** A representative Rietveld plot for the product of the alteration experiment at 175 °C for 6 days in artificial burial solution.

## 7 Competing interests

The authors declare that they have no conflict of interest.

## 8 Acknowledgements

We sincerely thank Dr. F. Nindiyasari for her help with biochemical sample preparation, microtome cutting and microtome polishing and S. He for the preparation of samples for XRD measurements. We very much thank Prof. J. Pasteris, Dr. E. M. Harper, Prof. U. Brand, Prof. L. Fernández Díaz for their corrections and fruitful discussions. We thank the German Research Council (DFG) for financial support in the context of the collaborative research initiative CHARON (DFG Forschergruppe 1644, Grant Agreement Number SCHM 930/11-1).





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





**Figure captions**




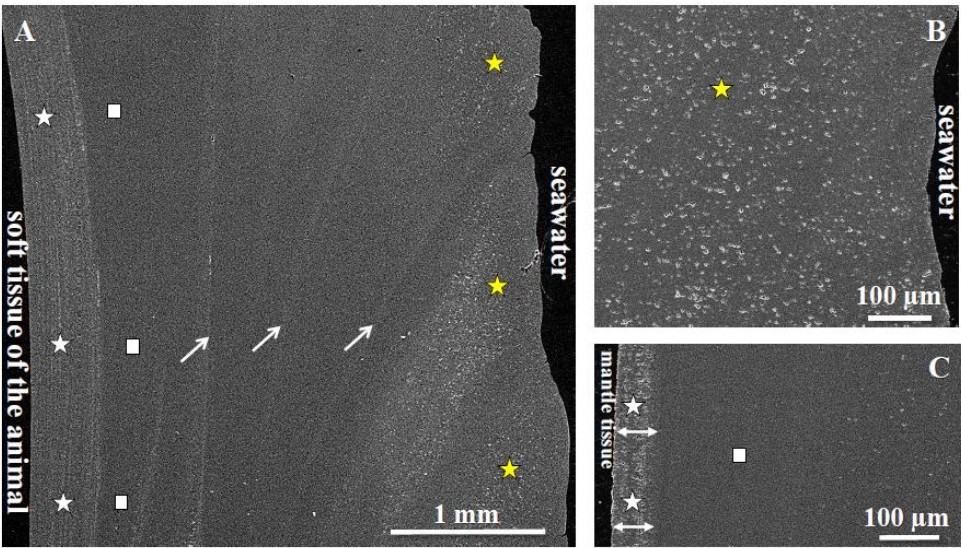


**Fig. 1.** SEM image showing ultrastructure characteristics of the shell of modern *Arctica islandica* (A), its high
porosity in shell layers facing seawater (yellow stars in A, B) and the dense shell portions (white stars in A, C)
close to the soft tissue of the animal. The innermost shell portions contain elongated pores (white star in C) with
the long axis of the pores oriented perpendicular to the inner surface of the shell (white arrows in C). Dense shell
parts are also present (white rectangles in A, C), in which pore density and size is very low and minute aragonite
crystals are closely packed. White arrows in A indicate the location of growth lines.





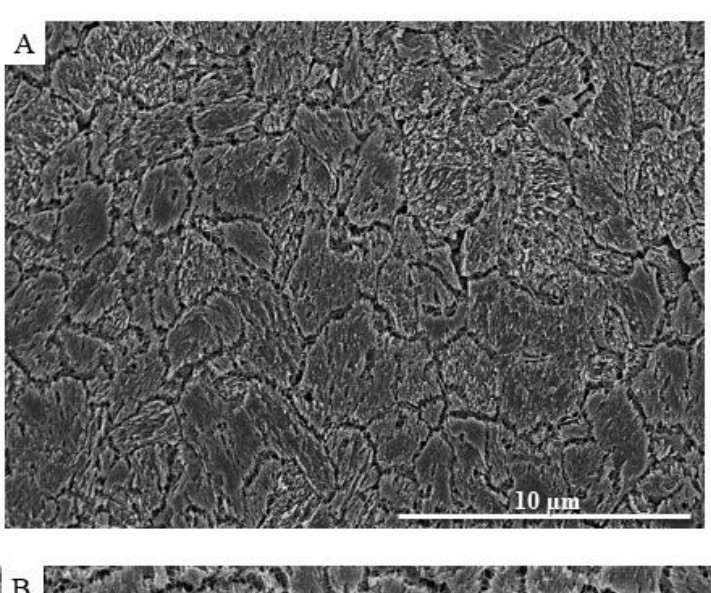

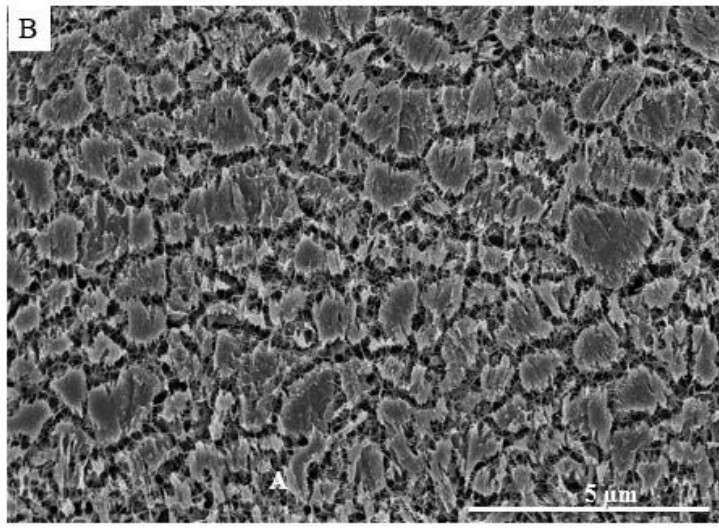


**Fig. 2.** FE-SEM micrograph of microtome cut, microtome polished, etched, and critical-point-dried surface of the
shell of modern *Arctica islandica*. (A) the shell portion next to seawater, (B) shell layer close to the soft tissue of
the animal. Etching occurred for 180 seconds and was applied to remove aragonite in order to visualise the spatial
distribution of (glutaraldehyde-stabilised) biopolymers within the shell. The portion of the shell facing seawater
consists of large and irregular mineral units, connected to each other and infiltrated by a network of organic fibrils.
In the layers next to soft tissue of the animal consists of significantly smaller mineral units. These are also
interconnected by organic fibrils.







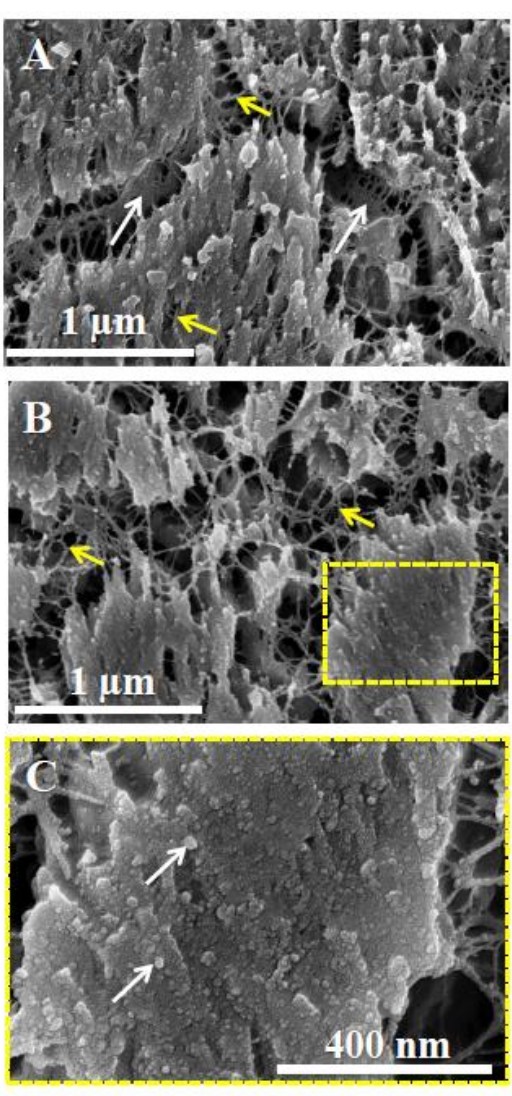


**Fig. 3.** FE-SEM micrographs of cut, microtome polished, etched, and critical-point-dried surfaces of of modern *Arctica islandica* next to seawater (A) and close to the soft tissue of the animal (B, C). Etching occurred for 180 seconds and removed aragonite in order to visualise the spatial distribution of (glutaraldehyde-stabilised) biopolymers within the shell. Readily visible is the nano-particulate consistency of the aragonitic hard tissue (white arrows in C) and the presence of biopolymer membranes (white arrows in A) and fibrils (yellow arrows in A, B) between the mineral units as well as between the aragonitic nanoparticles.






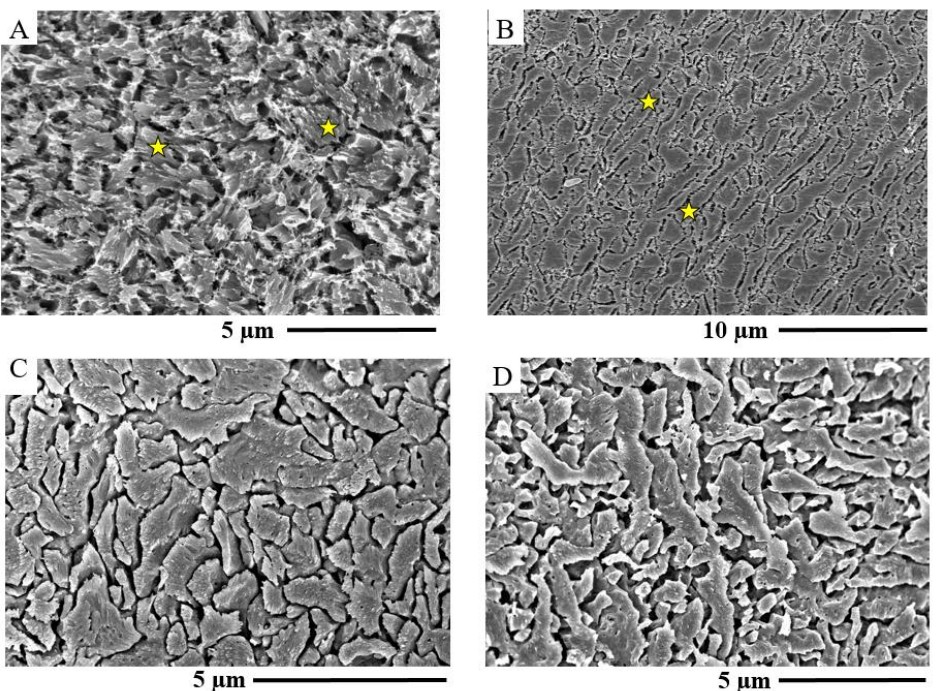



**Fig. 4.** SEM micrographs of cut, microtome polished, etched, and critical-point-dried surfaces of the of
experimentally altered *Arctica islandica* shell materials: (A, C) sweater-adjacent layer, and (B, D) shell layer close
to the soft tissue of the animal. Etching occurred for 180 seconds and was applied for visualisation of the spatial
distribution of (glutaraldehyde-stabilised) biopolymers within the shell. 10 mM NaCl + 10 mM $MgCl_2$ aqueous
solution (burial fluid) was used for alteration at 100 °C for 28 days (A, B) and at 175 °C for 7 days (C, D). Mineral
units are indicated by yellow stars in A and B.



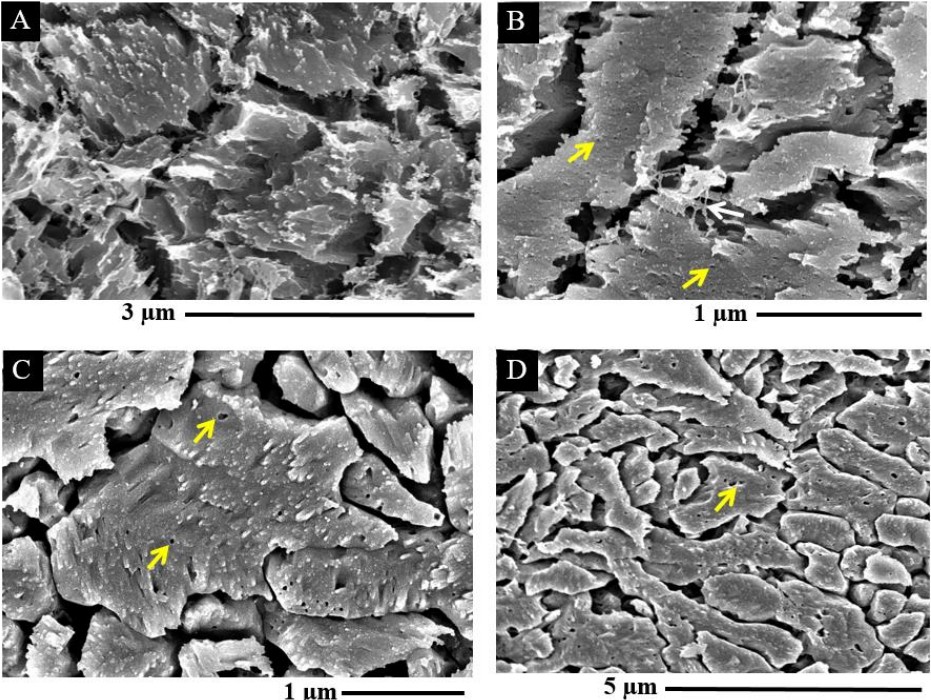


**Fig. 5.** FE-SEM micrographs that zoom into experimentally altered *Arctica islandica* shell material shown in Fig.
5. 10 mM NaCl + 10 mM MgCl$_2$ aqueous solution (burial fluid) was used for alteration at 100 °C for 28 days (A,
B) and at 175 °C for 7 days (C, D). Figs. A and C show material from the seawater-adjacent shell layers; B and D
depict material from shell layers at the soft tissue of the animal. For the material treated at 175 °C (C and D) the
biopolymers have decomposed and dissolved. Readily observable are minute round holes within the mineral units
(yellow arrows in B, C, D) that were filled in the pristine shell, prior to alteration, by biopolymer fibrils.





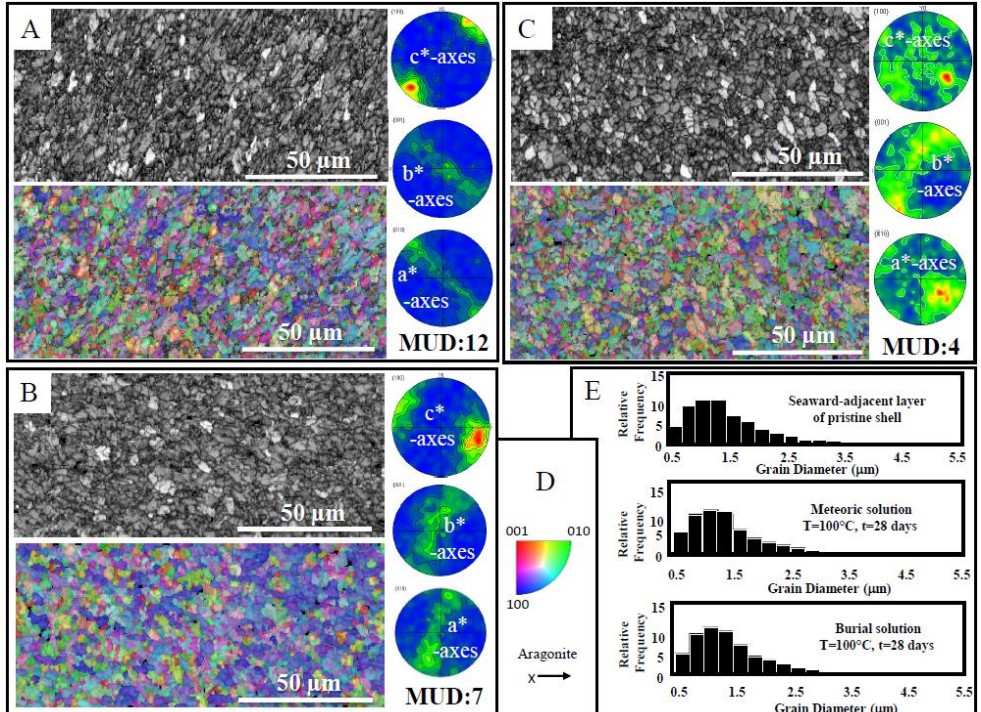

1025

**Fig. 6.** EBSD band contrast images (grey scale) and orientation maps (colored, color code given in D) with corresponding pole figures of pristine (A) and experimentally altered (B, C) *Arctica islandica* shell material. In the pole figure colour codes for pole density, with the maximum in red corresponding to the given MUD value for each set of pole figures, respectively. All EBSD measurements were taken at the seawater side of the shell. Alteration temperature was 100 °C and was applied for 28 days. The solutions used were artificial meteoric fluid in (B) and artificial burial fluid in (C). As the pole figures show, in comparison to the microstructure of pristine *Arctica islandica* (A), the crystal orientation pattern in the skeleton is not affected by treatment with the solutions. (E) Grain diameter statistics for pristine and experimentally altered *Arctica islandica* shell material obtained from the EBSD measurements that are shown in Figures A to C. There is no significant difference in grain size between pristine and hydrothermally altered *Arctica islandica* shells.



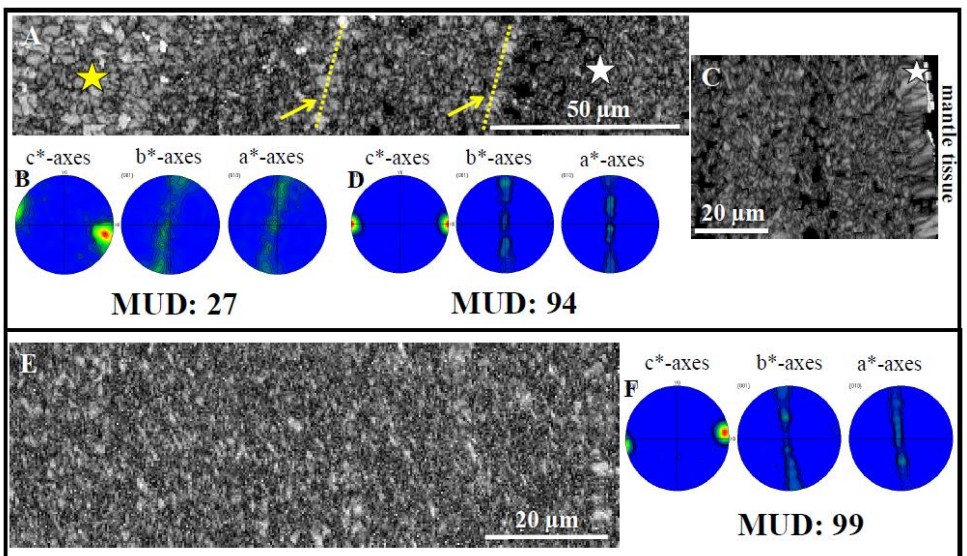

**Fig. 7.** EBSD band contrast images (grey scale) and corresponding pole figures of hydrothermally altered (100 °C for 28 days) *Arctica islandica* shell material with artificial meteoric fluid (A, B, C, D) and artificial burial fluid (E, F). In Fig. A the change in shell microstructure is visible from the seawater-adjacent shell layer that contains large aragonite crystals (yellow star in 7A) and many pores, to shell portions close to the soft tissue of the animal, which consist of densely packed small aragonite crystallites (white star in 7A). In C and E band contrast maps and pole figures are shown that were taken at the shell portion next to the soft tissue of the animal. As the pole figures and the high MUD values in D and F highlight, this part of the shell remains almost unaltered and the pristine *Arctica islandica* microstructure is kept. In (A) the two yellow arrows and the two dashed lines indicate the location of former growth lines where, in pristine shells, an increased amount of organic material is present. As the latter is destroyed during hydrothermal alteration space becomes available for infiltration of fluids.




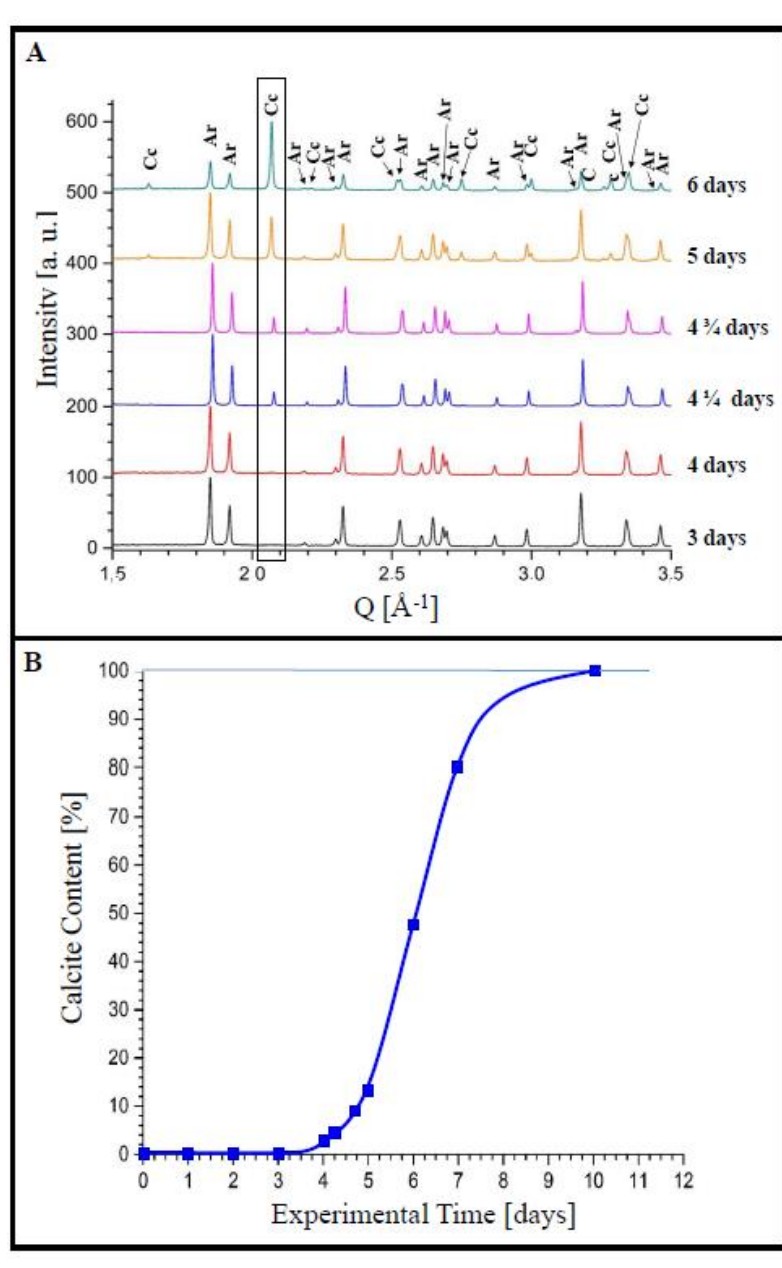


**Fig. 8.** (A) Selected x-ray diffractograms for three to six days of alteration of *Arctica islandica* shell material.
Alteration took place in artificial burial solution at 175 °C. (B) Newly formed calcite content relative to alteration
time (days) calculated from Rietveld analyses of the XRD measurements.




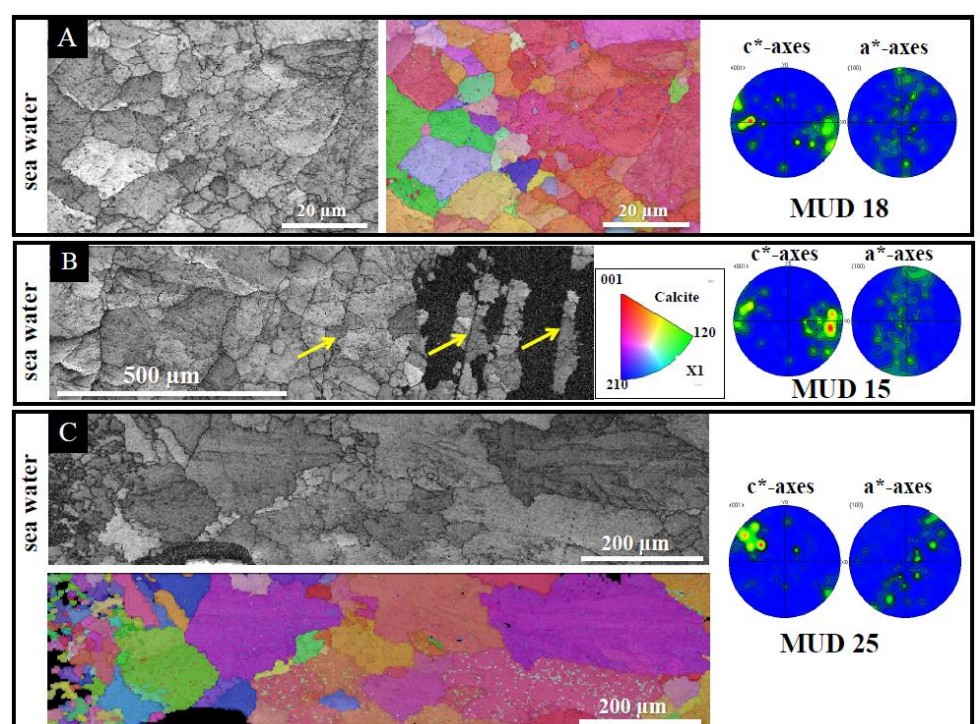


**Fig. 9.** EBSD band contrast maps, colour-coded orientation maps, and corresponding pole figures highlight the microstructure and texture of altered *Arctica islandica* shells at 175 °C in artificial meteoric solution. EBSD measurements shown in (A, B) were taken on shells that were subject to hydrothermal alteration for 7 days. Measurements shown in image C refer to shells where alteration lasted for 84 days. At 175 °C for both alteration times aragonite has transformed almost completely to calcite, and the shell microstructure is characterised by large and randomly oriented calcite crystals. The initial growth of calcite is visible at the location of former growth lines (yellow arrows in B).

1070





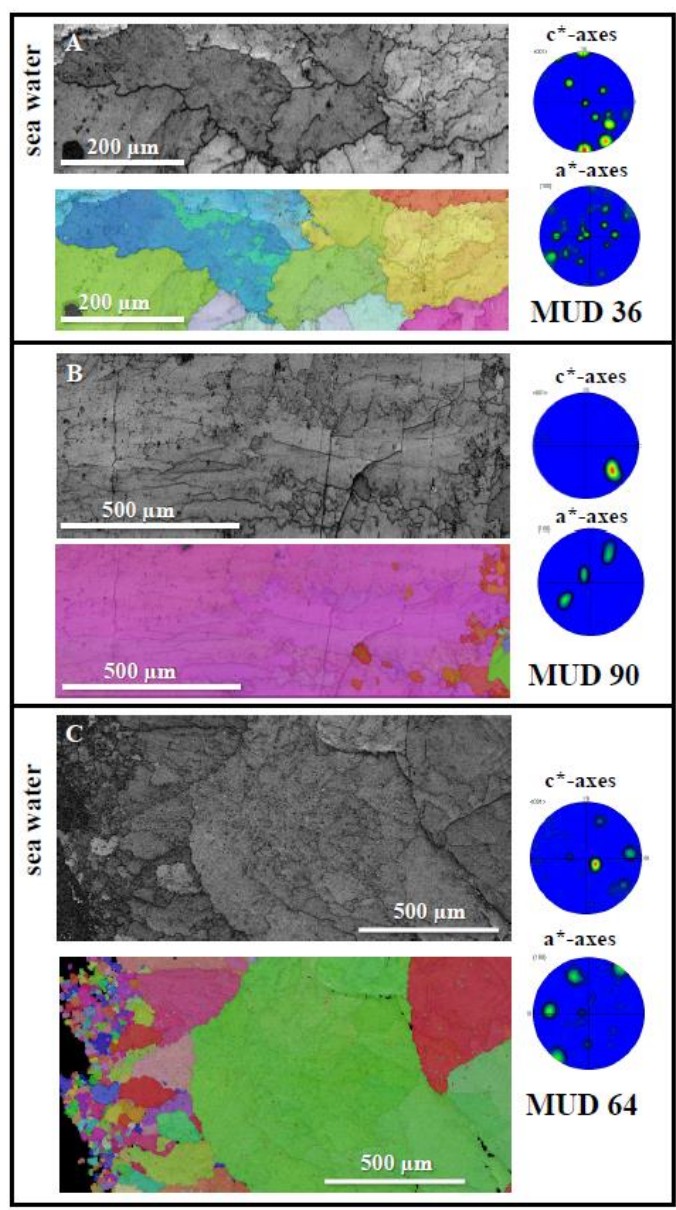

**Fig. 10.** EBSD band contrast maps and colour-coded orientation maps with corresponding pole figures for hydrothermally altered *Arctica islandica* shells at 175 °C in water simulating burial diagenesis. EBSD measurements shown in A and B were taken on shells that were subject to hydrothermal alteration for 7 days, while the measurement shown in C was performed on shells where alteration lasted for 84 days. At 175 °C for both alteration times most of the aragonite has transformed to calcite.





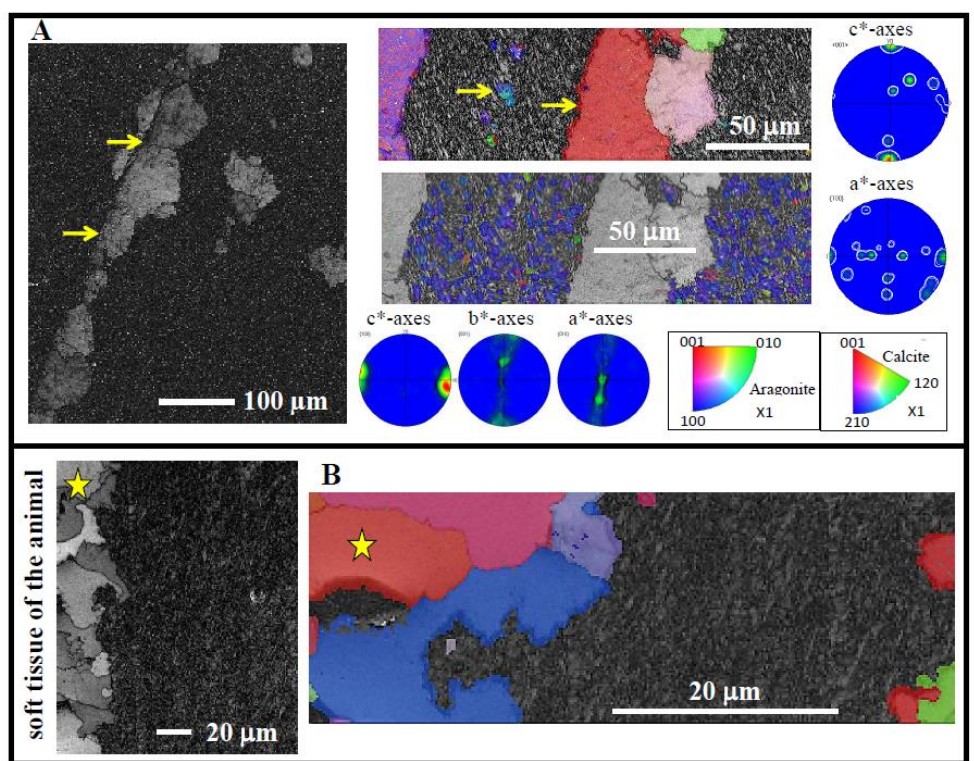

**Fig. 11.** EBSD band contrast (in grey), crystal orientation (colour-coded for orientation) maps, and corresponding pole figures of altered *Arctica islandica* shells at 175 °C in artificial meteoric (A) and burial (B) solution, respectively. Clearly visible is the initial formation of calcite at the location of former growth lines (yellow arrows in A) and the growth of large calcite crystals (yellow stars in B) that formed at the shell portion that is in direct contact with the alteration solution. Note that some pristine aragonite in the dense shell portion is still present.



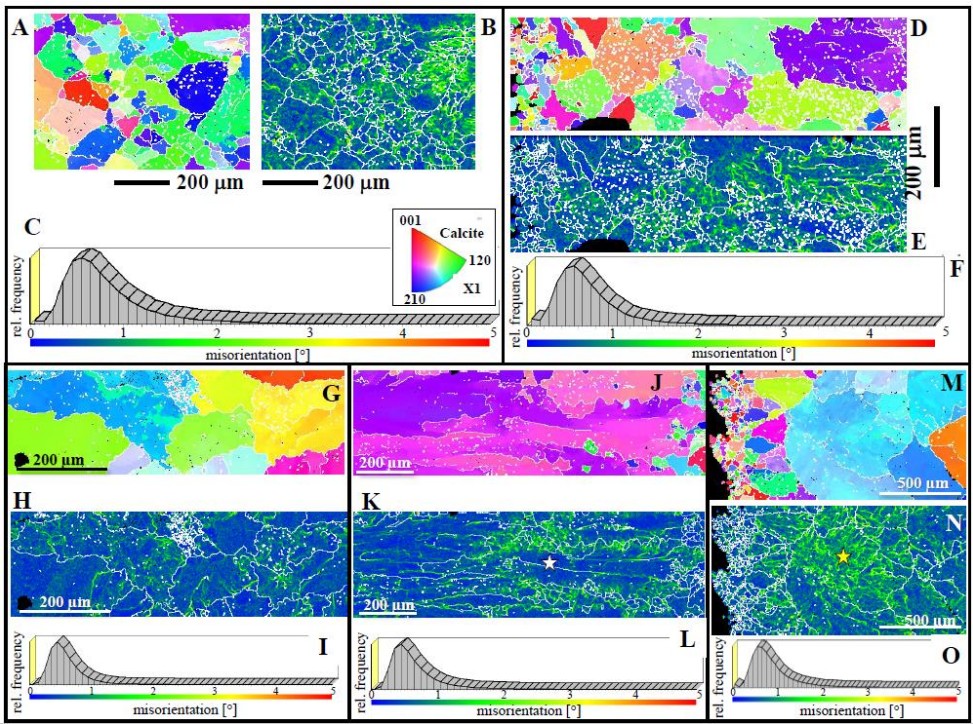

1086

1087

**Fig. 12.** Presentation of grain size (A, D, G, J, M), pattern of local misorientation distribution (indication of internal stress/strain) (B, E, H, K, N), and degree of local misorientation (C, F, I, L, O) for experimentally altered shells of *A. islandica* carried out in simulated meteoric solution at 175 °C for 7 (A to C) and for 84 days (D, E, F), and in burial solution at 175 °C for 7 (G to L) and 84 days (M to O), respectively. Grains are defined by using a critical misorientation of 5°. The outer perimeter of each grain is outlined with white lines and, for a better visualisation of individual grains, the grain map is shown in random colours (thus the colour-coding in A, D, G, J, M does not show orientation differences). At an alteration temperature of 175 °C, large calcite grains form. Local misorientation is about 2 degrees (see legends in C, F, I, L, O), irrespective of alteration duration and solution. The white star in K marks stress-free shell portions, while the yellow star in N indicates the location of an increased stress accumulation.





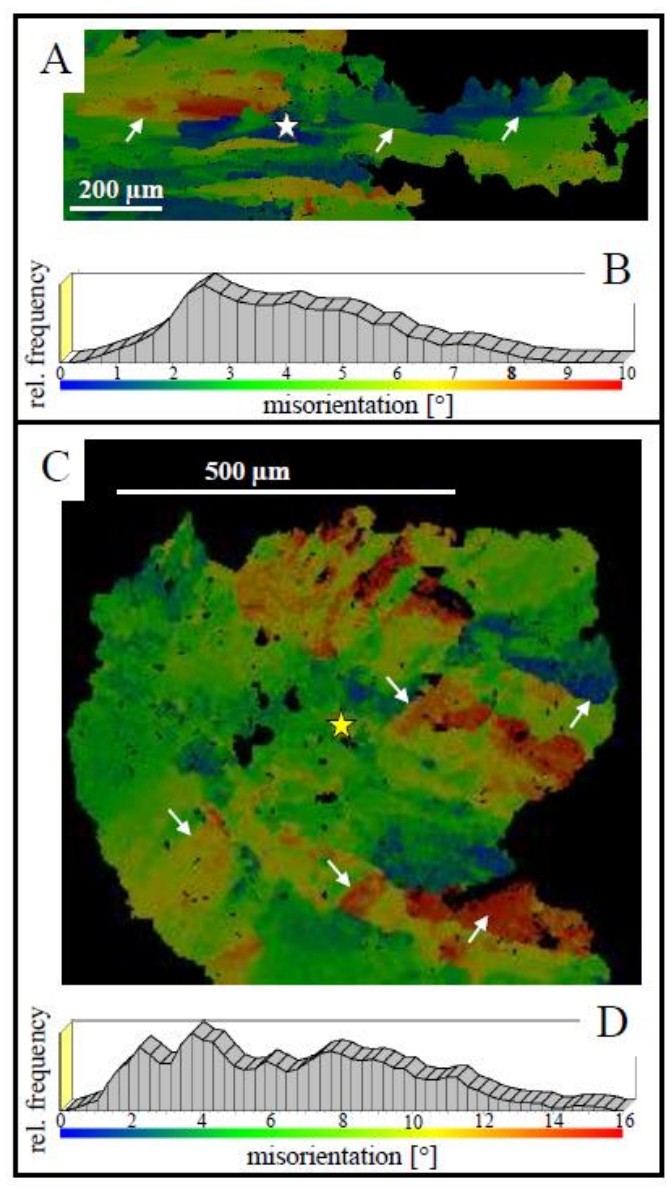


**Fig. 13.** Colour-coded visualisation (A, C) and degree of internal misorientation (B, D) within two large, mm-sized grains that grew in simulated burial solution at 175 °C for 7 (A) and 84 (C) days. The grain shown in A contains some stress-free portions within its centre (indicated by blue colours and the white star in A), while internal misorientation in the grain shown in C is highly increased and occurs everywhere within the grain (D). The yellow star in C points to the region where, in this grain, stress accumulation is highest.







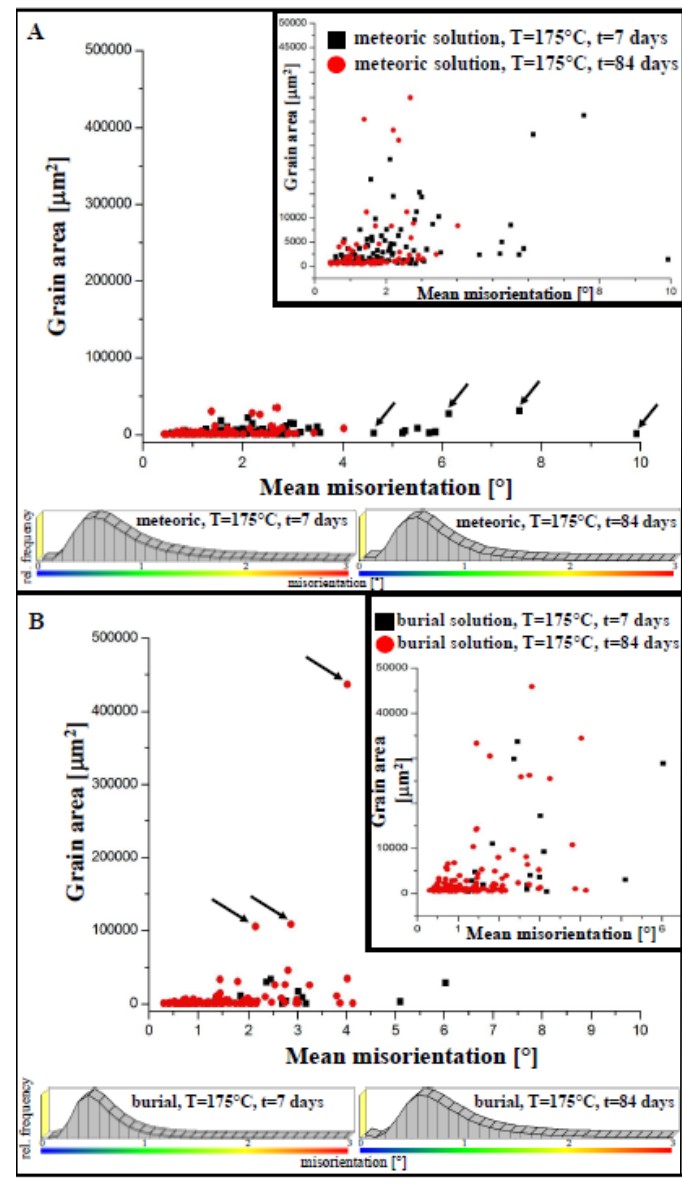

**Fig. 14.** Grain area versus mean misorientation within individual grains obtained for newly formed calcite at alteration of *Arctica islandica* aragonite in artificial meteoric (A) and in burial (B) solutions at 175 °C and for 7 and 84 days, respectively. The Mg-containing (burial) alteration fluid induces the formation of large calcite grains that show a low degree of misorientation within the grains (B), while with artificial meteoric solution, the solution that is devoid of Mg, significantly smaller grains are obtained. However, the latter occur with a high mean misorientation within the individual, newly formed grains.





**Tables**





**Table 1.** Detailed conditions used in hydrothermal alteration experiments of modern *Arctica islandica*.
Major and minor element chemical data of pristine *Arctica islandica* were obtained by EPMA using a
CAMECA SX100 system (Goetz et al., 2014) and amounts to: 0.11 wt% Sr, 38.06 wt% Ca, 0.05 wt% Mn,
0.54 wt% Na, 0.01 wt% P, 0.01 wt% Mg, 0.09 wt% Ba, 0.05 wt% Fe(III) and, 0.02 wt% Cl.

| Sample name | Fluid type | NaCl content [mM] | MgCl₂ content [mM] | Temperature [°C] | Experimental time | Alkalinity [mM] | pH | Mg-content of fluid after experiment [mg/L] |
|---|---|---|---|---|---|---|---|---|
| CHA-M-040 AI21 B2 | meteoric | 10 | - | 100 | 28 days | 1.69 | 7.91 | 3 |
| CHA-M-042 AI 23 B2 | meteoric | 10 | - | 175 | 7 days | 7.72 | - | 0 |
| CHA-M-046 AI27 B1 | meteoric | 10 | - | 175 | 84 days | 10.75 | 7.78 | 1 |
| CHA-M-043 AI24 B2 | burial | 100 | 10 | 100 | 28 days | 2.02 | 8.39 | 112 |
| CHA-M-041 AI22 B2 | burial | 100 | 10 | 175 | 7 days | 9.96 | - | 84 |
| CHA-M-046 AI 27 B2 | burial | 100 | 10 | 175 | 84 days | 6.99 | 7.51 | 165 |















**Table 2.** Crystal co-orientation (texture) strength expressed as multiple of uniform (random)
distribution (MUD) of modern and experimentally altered *Arctica islandica* shells. Ar: aragonite, Cc:
calcite.

| Sample name | Fluid type | Temperature [°C] | Experimental time | MUD value of the outermost shell part | MUD value of the central shell part | MUD value of the innermost shell part |
|---|---|---|---|---|---|---|
| modern reference | - | - | - | 12 Ar/32 Ar | 58 Ar | 88 Ar |
| altered specimen *CHA-M-040 AI21 B2* | meteoric | 100 | 28 days | 7 Ar | 27 Ar | 94 Ar |
| altered specimen *CHA-M-043 AI24 B2* | burial | 100 | 28 days | 4 Ar | - | 99 Ar |
| altered specimen *CHA-M-042 AI23 B2* | meteoric | 175 | 7 days | 18 Cc | 15 Cc | - |
| altered specimen *CHA-M-046 AI27 B1* | meteoric | 175 | 84 days | 25 Cc | 32 Cc | - |
| altered specimen *CHA-M-041 AI22 B2* | burial | 175 | 7 days | 36 Cc | 90 Cc | 80/81 Cc |
| altered specimen *CHA-M-046 AI27 B2* | burial | 175 | 84 days | 64 Cc | 62 Cc | - |





