# Peer review of "Experimental diagenesis: Insights into aragonite to calcite transformation of *Arctica islandica* shells by hydrothermal treatment"

_Biogeosciences, 2016_

## Referee Comment (RC1) · Anonymous Referee #1 · 4 Oct 2016

**Review to the manuscript bg-2016-355:**

**"Experimental diagenesis: Insights into aragonite to calcite transformation of Arctica islandica shells by hydrothermal treatment",** by L. A. Casella, E. Griesshaber, X. Yin, A. Ziegler, V. Mavromatis, D. Müller, A.-C. Ritter, D. Hippler, E. M. Harper, M. Dietzel, A. Immenhauser, B. R. Schöne, L. Angiolini and W. W. Schmahl
* * *
This is a very interesting manuscript on the effect of burial and diagenetic processes on the characteristics of the microscopic structure of *Artic Islandica* shells. Using with guarantees the fossil register as a source of information on the paleo-Earth's physicochemical conditions requires understanding how the record of these conditions in fossils may have been altered during diagenesis. This work represents an important step in this direction.

The topic of this manuscript fits the scope of Biogeosciences and may be of interest to a variety of geoscientists.

This work approaches the problem of fossils' diagenetic changes in an innovative experimental way. The experiments are well design and the samples are thoroughly characterized using EBSD to analyze changes in *Artic Islandica* shells' microstructure. This approach produces a sound set of results which is discussed taking into consideration both, the physicochemical characteristics of the $CaCO_3$-$H_2O$ system and most recent advancements in the understanding of mineral replacement phenomena.

This manuscript is well written and clearly organized. The methods section is very thorough. The results are presented in a clear way and illustrated by well selected images. The discussion is well organized and easy to follow. Finally, the conclusions of this work are relevant and well based on the experimental results. Furthermore, the reference list is extended and very complete. I only have a few suggestions (see list below) for the authors to consider. Most regard with typos and statements that, in opinion, could be better qualified.

1. Line 279: "... SEM images on the left hand side of Figs. 5 and 6 are taken from ..." Shouldn't it read Figs. 4 and 5?

2. Line 303: "... The small changes in MUD values must be attributed to **the** fact that it was impossible ..."

3. Lines 343-348: " Thus, as the replacement reaction proceeds, the percolating diagenetic pore fluid is undersaturated with respect to aragonite but is saturated with respect to calcite."  If the fluid were saturated with respect to calcite this phase would not nucleate in the first place and would not grow after its nucleation. A certain degree of supersaturation is required for the system to overcome the energy barriers associated to both, heterogeneous nucleation of calcite on aragonite and calcite growth. This could be better explained.

4. Lines 420-428: I basically agree with the authors explanation. However, I would have liked a discussion of the Mg content of the newly formed calcite. If this is magnesian calcite, is higher solubility compared to that of pure calcite would determine a smaller driving force for the transformation. In other words, both nucleation and growth would occur under lower supersaturation, which would further explain the smaller number of crystals and their larger sizes.

5. Line 472: " in palaeontology as it **is** a prerequisite to taxonomic, taphonomic...."

6. Line 489: " In particular, the resistance of biogenic aragonite to replacement by calcite up to temperature of 175 °C during hydrothermal alteration offers an additional explanation for the preservation of aragonitic shells/skeletons, besides the taphonomic windows envisaged by Cherns et al. (2008)." The authors conduct experiments at two temperatures, 100 and 175 ºC. In my opinion, the fact that at 100 ºC the aragonite-calcite transformation does not occur after 28 days does not qualify them to state that there is a resistance of biogenic aragonite to be replaced by calcite at temperatures below 175 ºC. The window temperature between 100 and 175 ºC is too large and 28 days is not such a large time, not for an experiment and more so when compared to geological times.

7. Conclusion 7. This is not a conclusion of this work. I understand that it must correspond to the paper by Balthasar and Cusack (2015), but it is not supported by results in this manuscript.

8. Conclusion 8. See comment 6.

9. Lines 1019-1020. "... shown in Fig. 5. 10 mM ...". I gues that it should read ""... shown in Fig. **4**. 10 mM ..."

10. Fig. 14 label. I miss a discussion of why the mean misorientation within the individual, newly formed grains in contact with burial solutions. I guess that this points to Mg incorporating into the newly formed calcite.

---

## Referee Comment (RC2) · A. Lüttge (Referee) · 27 Oct 2016

Review of the article „*Experimental diagenesis: Insights into aragonite to calcite transformation of Arctica islandica shells by hydrothermal treatment*" by Casella et al. – Contributing to an interesting discussion

Andreas Luttge, Cornelius Fischer & Rolf S. Arvidson
MARUM, Department of Earth Science (FB5), University of Bremen, Germany

The paper by Casella et al. tackles an interesting and important topic, i.e., the kinetics of the transition from biomineralized aragonite to inorganic calcite. The authors present an impressive data pool and some extensive observations from long-term laboratory experiments (up to 84 days!) at temperatures ranging from 100 to 175 °C, and two different Na-, and Mg-concentrations. These settings mimic burial and meteoric fluid conditions.

The main result of this study is that below 175 °C there are no signs of aragonite to calcite transformation. At 175 °C the authors found a period of 4 days where the systems seems "dormant". After this period the transformation reaction becomes detectable and runs to completion within a remarkable short time of just 6 days. It would be of great interest for future studies to elucidate the processes that occur during the 4 day dormant period. This seems to be a likely analog to cement hydration reactions, in which an induction period of low heat production is followed by an "acceleration period", during which significant nucleation and growth advances the overall extent of reaction.

The authors present SEM images of their starting material, the shells of modern *Arctica islandica*. This material is characterized by heterogeneous distribution of porosity pattern and pore sizes. There is an interesting structural difference between the inner side of the shell, the center section, and the outer side that has originally faced the sea water. It is certainly a complication that there is also organic material interspersed with the solid shell material.

Discussion: For the community, it is an interesting question as to whether (micro)organisms simply passively modify local environmental conditions, thus rendering metastable what would otherwise be thermodynamically unstable phases, or if, conversely, such organisms actively form unstable phases by pumping electrons or redox-sensitive molecules against gradients [1,2]. In any case, we would expect that the deviation from thermodynamic equilibrium would be relatively small. In other words, one would expect that energetically expensive reactions would also be "expensive" for the organism involved. Casella et al.'s findings seem to nicely support this hypothesis. It takes a significant increase in temperature to convert the aragonite into thermodynamically stable calcite. However, this insight does not answer the important question: why do organisms produce metastable aragonite and not the stable phase calcite?

Another interesting point is the potential occurrence of spatial patterns in the distribution of replacement reaction rates. The existence of a characteristic porosity distribution within the shell material is able to foster heterogeneous material fluxes. Thus, in future studies we expect direct mapping of the reaction rate in order to identify rate components that form the heterogeneous overall reaction rate. Such rate components provide important input values for the simulation of fluid-solid reactions [3].

Reference:
[1] A Luttge, L Zhang, KH Nealson (2005) Mineral surfaces and their implications for microbial attachment: results from Monte Carlo simulations and direct surface observations. American Journal of Science 305 (6-8), 766-790
[2] KJ Davis, KH Nealson, A Luttge (2007) Calcite and dolomite dissolution rates in the context of microbe–mineral surface interactions. Geobiology 5 (2), 191-205

[3] C Fischer & A Luttge (2016) Beyond the conventional understanding of water–rock reactivity. Earth and Planetary Science Letters, http://dx.doi.org/10.1016/j.epsl.2016.10.019

---

## Referee Comment (RC3) · Anonymous Referee #3 · 27 Oct 2016

Overall assessment The manuscript presents an interesting study of the simulation, of taphonomic processes with two different experimental fluids, called meteoric and burial. In both cases the shells of Arctica are subjected to increasing temperature. The authors show how the changes in mineralogy, from aragonite to calcite, progress across the growth planes and analyze the orientation, composition, size and distribution of the neoformed crystals, as well as the timing of the transformation process. The study is competent, and the techniques (mainly XRD, EPMA and EBSD) are adequate. The wealth of data coming from the crystallographic study (EBSD) is particularly remarkable. The results are very interesting and are highly significant with regards to the taphonomy of aragonitic shells (although not in a general sense, but in the case

in which transformation of aragonite to calcite progresses by a process of coupled dissolution-reprecipitation mechanism).

I will now deal with the specific aspects required by Biogeosciences Discussions (BGD): 1. Does the paper address relevant scientific questions within the scope of BG? YES, particularly taphonomy of shells. 2. Does the paper present novel concepts, ideas, tools, or data? YES, particularly, the focus is highly innovative. 3. Are substantial conclusions reached? YES, it adds substantial knowledge to the particular process studied. 4. Are the scientific methods and assumptions valid and clearly outlined? YES, in general, although I find that some conceptual aspects should be explained more in length (see detailed comments. 5. Are the results sufficient to support the interpretations and conclusions? YES, in general (but see detailed comments). 6. Is the description of experiments and calculations sufficiently complete and precise to allow their reproduction by fellow scientists (traceability of results)? YES, in general, although additional data on the composition of the fluids (Table 1) are needed. 7. Do the authors give proper credit to related work and clearly indicate their own new/original contribution? YES, in general (but see detailed comments). 8. Does the title clearly reflect the contents of the paper? YES. 9. Does the abstract provide a concise and complete summary? YES. 10. Is the overall presentation well-structured and clear? YES. 11. Is the language fluent and precise? YES. 12. Are mathematical formulae, symbols, abbreviations, and units correctly defined and used? Some technical abbreviations (EPMA, TAP, PET, LPET, LLIF…) should be defined for non-specialists. 13. Should any parts of the paper (text, formulae, figures, tables) be clarified, reduced, combined, or eliminated? NO. 14. Are the number and quality of references appropriate? YES. 15. Is the amount and quality of supplementary material appropriate? YES.

Detailed comments: 133. 'for physiological generate', this is an odd expression. 138. 'pre-Neogene A. islandica', certainly not this species. 157. 'mineral in reference', reference mineral? 159: 'knifes', knives, blades. 193. Remove 'and' 198-199. This initial

bit needs to be expanded further. More data on the composition of fluids and why their compositions have been chosen are necessary. 207. At least a reference is needed before the period. 237-238. This feature is not visible at the magnification of Fig. 1B. 240. Fig. 2B is heavily etched. Accordingly, any conclusion about the density of pores is doubtful. 260. Again, the nanogranules I Fig. 3C are etching and not original features. 260-261. 'These are co-aligned to form mesocrystals', probably true, but not demonstrated here. I advise deletion. 284-287. 'holes', do the authors refer to the membranes between mineral units? Please, be precise. 302-303. the coincidence in the orientations of the c-axes of aragonite and calcite, before and after alteration, is remarkable. I would like the authors to comment on this. Could there be some kind of epitaxial growth of calcite on the aragonite? 303. insert 'the', before 'fact'. 304-305. 'and to the patterns of growth lines', what is meant with this exactly? please reword. 305-306. There is something wrong here. The maximum for a* is more or less coincident with the maximum of c*, and does not conform with the orientations of b*. The authors should review their data to make sure that the a* pole figure is the correct one. 306. 'the', before 'c-axis' 329-330. The explanation based on the small number of crystals mapped is not enough. Again, as above, is there the possibility of epitaxial growth? 351. Subsection 4.1.- I do not think that the title (particularly 'Characteristics of the grains') actually conforms to the contents. 372. 'Figs. 7, 8, 9', I find that only Fig. 9b is relevant in the context. 394-395. May be I do not understand properly, but in the scale of misorientations, blue colors imply very low values, whereas green, yellow and red are increasing values. Isn't it the other way round? 400-401. 'numerous small 400 calcite crystallites, a clear cause for the occurrence of internal strains'. Something is inconsistent here. Crystallite boundaries are defined by misorientations values above 5°, but this is not real, just a convenience. At the positions of 'crystallites' (defined in this way) misorientations values just peak. Therefore, they are not real and cannot be the origin of misorientations. To demonstrate that they are real, other techniques (e.g., TEM) are necessary. 411. '13B', rather 13C? 412. The linear structures in 13C at consistent angles argue for crystalline structures ({104} faces of calcite?). 422-424.

This conclusion does not seem significant in the absence of a statistical analysis, and in view of the low number of data which deviate from the trend in 14B. 426-428. I cannot follow the statements contained in the two sentences. 431. Subsection 4.3.- In my opinion, the paragraph immediately above could be included within this subsection. 451. Fig. 6C-D is not relevant within the context. The images supporting this statement have to be referred unequivocally. 472. 'it a', insert 'is' 475. 'mineral', minerals. 485. 'molluscs, mollusc. Page 13, last paragraph. Somewhere around here, the authors should state that the change from aragonite to calcite does not always proceed in this way. For instance, there is also the possibility that total dissolution followed by precipitation by calcite (or other minerals, gypsum, pyrite) takes place. Page 14. Conclusions, point 7.- Please, state clearly that the recognition of the first tipping point ($50°$ to $60°$) is not derived from the present study. 598. Dr. E.M. Harper is also a coauthor; please remove. 654, 657, 666, 677. As far as I know, some of these Spanish names should bear accent. Please check. In 666, 'Gonzales' is actually González. 670, 775. Italicize species and genus names. Check other cases I could have missed. 718. Space between 'bivalve' and 'molluscs' 826. 'Zetterstrom' is Zetterström. 1023-1024. How do the authors know that the minute holes were filled with biopolymer fibrils. I do not think the statement can be drawn with the techniques used in this study. I will remove it.

---

## Referee Comment (RC4) · Anonymous Referee #4 · 30 Oct 2016

This study provides important new data and approaches to understand the transformation of biological aragonite to calcite, a problem that has significant impact on many aspects of the biogeosciences and still lacks a thorough understanding of the major underlying processes.

The paper is well written; figures are of excellent quality and support very well the presentation of data and discussion. The organization of the paper could be improved, and some recommendations to this end can be found in the detailed comments listed below, together with some editorial suggestions and question that could be addressed to improve the discussion of the extensive, well documented dataset.

First paragraph of 'Abstract' may be moved to 'Introduction'.

[Figure]

Line 52: I understand from the text that experimental conditions were at 100° and 175°C, and not 'between' these temperatures.

Line 58: '...in all experiments below 175°C...' This implies that the described observation occurs at, e.g., 174°C. Be precise.

Line 72: Again, this statement implies that 175°C is a critical threshold temperature, while it is simply the temperature at which many of the experiments were run.

Line 82: '...limited vital effect...' This is certainly an over-simplification, as 'vital effects' depend on the complexity of the biomineralisation process of the organism, the element or isotope system considered; and can be negligible or significant.

Line 91-106: Do we need to know that much about A. islandica in order to understand the significance of this paper, which is more concerned about general aspects of biological aragonite diagenesis? I suggest shortening this paragraph as it distracts from the main focus of the paper.

Line 124: ...natural or synthetic aragonite...

Lines 128-130: edit capitalized words

Lines 132-140: These statements should be at the beginning of the 'Introduction'.

Lines 199-200: No isotope data are reported in table 1.

Line 200: Something wrong with this sentence; as no isotope data are reported or discussed, sentence could be omitted.

Line 205: According to table 2, samples were experimentally altered during 7, 28, and 84 days at temperatures of 100 or 175°C. Be precise.

Line 232: It may be advisable to refer to inner and outer shell layers, consistent with the terminology in Fig. A1. 'Shell portion facing seawater' is somewhat misleading, as only the surface of the shell may be in contact with seawater, and is additionally protected

by the periostracum (see Fig. A1).

Lines 273-275: No explanation is offered for the high Sr-concentrations along growth lines throughout the text. Is Sr part of the organic matrix, or the mineral aragonite?

Line 288: I suggest reporting the EBSD results in a separate section with its own subheading.

Line 315: The observation of calcite formation is a very fundamental result that should be higher up in the 'Result' section, not at its end, hidden in the description of aragonite microfabrics.

Line 368: Is there evidence that supersaturation was reached in the 100°C experiments? The experimental fluids should be initially undersaturated, no aragonite dissolution is reported from the experiments, and the data in Table 1 are not sufficient to estimate saturation states.

Line 370: Fig. A3 shows alteration at 100°C, and not at 175°C.

Line 375: Only the first paragraph of this section is about the time lag of calcite replacement, while 4/5 of the section are about orientation of newly formed calcite crystals. The latter discussion is very important, and certainly deserves its own subheading.

Line 445: The pH of the solution and/or CO2 partial pressure must be considered important, as they control the solubility of the original phase. Experiments of the current study were run with fluids with relatively high pH values and presumably very low pCO2 (no data provided). This should be discussed when comparing the present result with older studies that observed more rapid reactions with low-pH solutions (lines 435-436).

Line 470-493: I disagree with many statements in this section. Aragonite dissolution is known from many near-surface environments and occurs essentially at surface temperatures, e.g. in the vadose environment. Factors other than temperature, e.g. pH of vadose fluids, anoxic diagenetic environments with sulfate reduction etc., have to be considered. Much has been written about a taphonomic bias related to shell mineralogy as well as about carbonate diagenesis, and these complex issues cannot be adequately addressed in such a short paragraph. It is not strictly relevant to the main topic addressed in this paper and I suggest omitting it.

Line 499: While it is correct that the inner shell layer is closer to the soft body of the bivalve, this is probably quite irrelevant for the observed differences in shell structure and porosity. Inner and outer shell layers are precipitated from the extrapallial fluid, i.e. at about the same distance from the tissue of the mantle margin where the shell is formed.

Table 1: Please clarify that all data (except Mg) are referring to fluids before the experiment. Did changes of any of the tabulated items occur during the experiment, or was this not analyzed?

'hydrothermal': rephrase to 'solution', 'diagenetic fluid', or 'experimental' fluid. 'Hydrothermal' is not synonymous with 'hot water'.

What about the importance of the organic matrix, its decay, and its role of providing pathways for fluids and in changing the chemistry of fluids? The time lag observed before nucleation of calcite crystals could be related to the time required for disintegration of the organic matrix, formation of local acidic conditions due to the addition of $CO_2$ from the inorganic breakdown of organic molecules, and the permeability created in the process.

---

## Author Comment (AC1) · 20 Dec 2016

| REVIEWER 1 | SUBMITTED MANUSCRIPT / REVIEWER COMMENTS | AUTHOR COMMENTS / REVISED MANUSCRIPT |
|---|---|---|
| Comment 1 / L. 279 | "…SEM images on the left hand side of Figs. 5 and 6 are taken from the…" | Changed to: "…SEM images on the left hand side of Figs. 4 and 5 are taken …" |
| Comment 2 / L. 303 | "…The small changes in MUD values must be attributed to fact that it was impossible …" | Changed to: "…The small changes in MUD values may be attributed to the fact that it was impossible…" |
| Comment 3 / L. 343-348 | "…Thus, as the replacement reaction proceeds, the percolating diagenetic pore fluid is undersaturated with respect to aragonite but is saturated with respect to calcite…."

 If the fluid were saturated with respect to calcite this phase would not nucleate in the first place and would not grow after its nucleation. A certain degree of supersaturation is required for the system to overcome the energy barriers associated to both, heterogeneous nucleation of calcite on aragonite and calcite growth. This could be better explained. | As the reviewer requested, we explain the nucleation of calcite even at a low degree of supersaturation. We include an additional Figure (Fig. 12) and include in the text a new chapter (4.1), see below.

 **4.1 Driving force in comparison to nucleation barrier**
 In sedimentary environments the fate of metastable biogenic aragonite or high-Mg calcite can follow two scenarios: (1) the metastable biogenic matter can be completely dissolved and removed by fluid transport to form molds that are later filled by cement or other neogenic minerals or (2) the metastable minerals may be replaced by stable low-Mg calcite in-situ, by a process which involves dissolution of the metastable phase into a nano- to micro-scale local fluid volume (e.g. a thin fluid film) from which the stable low-Mg calcite precipitates without long-range transport (Brand & Veizer, 1980, 1981; Brand, 1991, 1994; Bathurst, 1994; Maliva 1995, 1998; Maliva et al., 2000; Titschak et al., 2009, Brand et al., 2010).). The latter process may preserve original morphological boundaries and microstructures such as prisms, tablets and fibres in bivalve shells. The replacement reaction from aragonite to stable low-Mg calcite is driven by the higher solubility (free energy) of the the metastable phase compared to the the stable phase. Thus, as the replacement reaction proceeds, the reactive, percolating experimental or diagenetic pore fluid becomes undersaturated with respect to aragonite owing to its relative supersaturation with respect to calcite, the less soluble mineral phase in the system. The maximal |

supersaturation $\Omega_{max}$ with respect to calcite, which can be obtained in a fluid, which draws its calcium and carbonate ions from the dissolution of aragonite, can be described as:

$$\Omega_{max} = \frac{K_{Sp}\,(Aragonite)}{K_{Sp}\,(Calcite)}$$

(1)

, where $K_{sp}$ stands for the ion activity products of the respective phase in the relevant pore fluid. The free energy difference or thermodynamic driving force is given by $\Delta G_{max} = - RT\ln \Omega_{max}$ . To obtain an estimate we used the data of Plummer & Busenberg (1982) and calculated the solubility products for calcite and aragonite for 25 °C, 100 °C, and 175 °C (Fig. 12). The maximal supersaturations $\Omega_{max}$ thus obtained are 1.39 (25 °C), 1.26 (100 °C), and 1.18 (175 °C). The replacement reaction first requires a nucleation step: the formation of the first calcite crystallites larger than the critical size r* (Morse et al, 2007). Empirical nucleation theory relates the activation energy $\Delta G_A(r^*)$ necessary to form a nucleus of critical size to the specific surface energy σ needed to form the interface between the nucleating phase and the matrix phase as

$$\Delta G_A(r^*) \propto \frac{\sigma^3}{(-RT ln\Omega)^2}$$

(2)

Only supercritical nuclei or pre-existing seed crystals of size r > r* of calcite can lower their free energy as their volume free energy gained by growth exceeds the adverse energy contributions of increasing interface area. To obtain a significant number of supercritical nuclei a critical supersaturation needs to be reached (Morse et al., 2007, Gebauer et al., 2008, Nindiyasari et al., 2014, Sun et al. 2015). Reported values for critical supersaturation levels $\Omega_{crit}$ required for calcite nucleation in various conditions range from the order of 3.7 (Lebron & Suarez, 1996, Zeppenfeld, 2003) to the order of 30 (Morse et al., 2007; Gebauer et al., 2008) or even several hundreds e.g. in hydrogel matrices (Nindiyasari et al., 2014). The DFT study of Sun et al.

| | | |
|---|---|---|
| | | (2015) arrives at $\Omega_{crit}$ = 5 for systems free of inhibitors such as Mg, and $\Omega_{crit}$ = 35 for modern sea-water. Accordingly, the supersaturation produced by the dissolution of aragonite is very small compared to supersaturation levels typically required for the nucleation of calcite. Thus we can expect that nucleation is a critical kinetic step in the replacement reaction of aragonite by calcite. |
| Comment 4 / L. 420-428 | "…However, we find that experiments conducted with the Mg-containing burial solution yield larger calcite crystals (black arrows in Fig. 14B) in comparison to the size of the grains obtained from experiments carried out with meteoric water (Fig. 14A). Grains obtained from alteration experiments with meteoric fluid show a significantly higher degree of mean misorientation (up to 10 degrees, black arrows in Fig. 14A), compared to that in grains that grew in burial solution. We attribute this to the nucleation rate: the crystals growing from each nucleus consume the aragonite educt (the precursor, original aragonite) until they abutted each other. Thus, larger crystals in the experiment with burial solution result from a smaller number of calcite nuclei. Again, this supports the idea that Mg2+ inhibits calcite nucleation. …"

I basically agree with the authors explanation. However, I would have liked a discussion of the Mg content of the newly formed calcite. If this is magnesian calcite, is higher solubility compared to that of pure calcite would determine a smaller driving force for the transformation. In other words, both nucleation and growth would occur under lower supersaturation, which would further | We included a paragraph (see below, chapter 4.5), a new Table (Table A1) and a Figure showing the Mg distribution in an altered shell measured by electron microprobe analysis (Fig. A13).

The newly formed calcite contains only small amounts of magnesium (Table A1) in the order of 0.1 wt % (or 0.006 in the formula unit), while the strontium content of the original aragonite in the order of 0.4 wt.% is retained in the calcite (0,005 in the formula unit). At the rim of the sample, where it was in direct contact with the bulk of the experimental fluid, we observe the local formation of Mg-rich carbonates in some places only (Fig. A13B and Table A1), with measured Mg-contents up to 19.7 wt % (0.716 in the formula unit, encountered in scan field 3 at the outer rim of the sample). The averaged composition in scan fields 4 and 9 may indicate dolomite, but like scan field 3, which has a Mg content exceeding that of dolomite, we more likely have magnesite with some calcite present, as judged from the EPMA map (Fig. A13B). |

|  | explain the smaller number of crystals and their larger sizes. |  |
|---|---|---|
| Comment 5 / L. 472 | "…in palaeontology as it a prerequisite to taxonomic,…" | Changed to:
"…in palaeontology as it is a prerequisite to taxonomic, ,…" |
| Comment 6 / L. 489-492 | "…In particular, the resistance of biogenic aragonite to replacement by calcite up to temperature of 175 °C during hydrothermal alteration offers an additional explanation for the preservation of aragonitic shells/skeletons, besides the taphonomic windows envisaged by Cherns et al. (2008). …"

The authors conduct experiments at two temperatures, 100 and 175 ºC. In my opinion, the fact that at 100 ºC the aragonite-calcite transformation does not occur after 28 days does not qualify them to state that there is a resistance of biogenic aragonite to be replaced by calcite at temperatures below 175 ºC. The window temperature between 100 and 175 ºC is too large and 28 days is not such a large time, not for an experiment and more so when compared to geological times. | We have conducted the following alteration experiments: at 100 °C, 125 °C, 150 °C and 175 °C, ranging for time periods between one and 84 days. Details are given in Table 1, Fig. A11 and all experiments are described in the method (2.2.3) as well as in the results (3.2) sections. Thus, we investigated in this study shell samples that were not only altered at 100 °C and 175 °C but also between these two temperatures. In addition we conducted experiments that lasted up to three months, and not only for 28 days. During the time span of a Ph. D. study (3 years) we could not conduct experiments with geologic time scales. |
| Comment 7 / L. 522-524 | "…7. Between two tipping points, one between 50 and 60 °C, the other between 160 and 180 °C, aragonite appears to precipitate from supersaturated aqueous solutions rather than calcite, such that the hydrothermal treatments of aragonite within this temperature bracket do not yield calcite. …"

This is not a conclusion of this work. I understand that it must correspond to the paper by Balthasar and Cusack | Taking results from the literature and from our experiments conducted at additional temperatures, we can state that between the two tipping points, one between 50 and 60 °C (Kitano et al. 1962; Taft, 1967, Ogino et al. 1987, Balthasar and Cusack, 2015), the other between 160 and 180 °C (Perdikouri et al, 2011, 2013, this paper), aragonite appears to precipitate from supersaturated aqueous solutions rather than calcite, such that the hydrothermal treatments of aragonite within this temperature bracket do not yield calcite.…" |

| | (2015), but it is not supported by results in this manuscript. | |
|---|---|---|
| Comment 8 / L. 525-527 | "… 8. The absence of aragonite replacement by calcite at temperatures lower than 175°C contributes to explain why aragonitic or bimineralic shells and skeletons have a good potential of preservation and a complete fossil record. …"

See comment 6. | See reply to comment 6.
During the time span of a Ph. D. study (3 years) we could not conduct experiments with geologic time scales.
Our experiments record the short term answer which may be different from the the possible change through geological times. However, within the applied time and temperature range we find very interesting results that help to understand microstructural and mineralogical findings generated from diagenesis taking place in nature. There is evidence that our results may explain some patterns of the geological record and if this is true for the 100 °C experiments it is also true for the 175 °C ones. There is no reason to expect a different behavior in 28 days in the window between 100 and 175 °C. So our assumptions and conclusions are valid and should be kept. The important point is that they help to explain and discuss several patterns of aragonitic preservation which were left unsolved up to now. Of course these are preliminary data, but they help to identify an experimental procedure to follow to understand diagenetic processes.
We will never be able to experiment with geological time, but we can get observations that help to explain patterns. |
| Comment 9 / L. 1019-1020 | "…shown in Fig. 5. 10 mM NaCl + 10 mM…" | Changed to:
"…shown in Fig. 4. 10 mM…" |
| Comment 10 / L. 1108-1113 | "… **Fig. 14.** Grain area versus mean misorientation within individual grains obtained for newly formed calcite at alteration of *Arctica islandica* aragonite in artificial meteoric (A) and in burial (B) solutions at 175 °C and for 7 and 84 days, respectively. The Mg-containing (burial) alteration fluid induces the formation of large calcite grains that show a low degree of misorientation within | We briefly discussed this point in sub chapter 4.5 (see below).

Grains obtained from alteration experiments with meteoric fluid show a significantly higher degree of mean misorientation (up to 10 degrees, black arrows in Fig. 15A), compared to the grains that grew in burial solution. Large mean misorieantations of >4 ° occur notably in the grains grown in the 7 days treatment in meteoric solution, while the corresponding 84 days |

| | | |
|---|---|---|
| | the grains (B), while with artificial meteoric solution, the solution that is devoid of Mg, significantly smaller grains are obtained. However, the latter occur with a high mean misorientation within the individual, newly formed grains. …"

I miss a discussion of why the mean misorientation within the individual, newly formed grains in contact with burial solutions. I guess that this points to Mg incorporating into the newly formed calcite. | treatment does not show a significant increase in grain area compared to the 7 days treatment. |
| **REVIEWER 2** | | |
| Comment 1 | The main result of this study is that below 175 °C there are no signs of aragonite to calcite trans-formation. At 175 °C the authors found a period of 4 days where the systems seems "dormant". After this period the transformation reaction becomes detectable and runs to completion within a remark-able short time of just 6 days. It would be of great interest for future studies to elucidate the processes that occur during the 4 day dormant period. This seems to be a likely analog to cement hydration reactions, in which an induction period of low heat production is followed by an "acceleration period", during which significant nucleation and growth advances the overall extent of reaction. | We thank for the comment and fully agree with the reviewer. Accompanying experiments are conducted right now. |
| Comment 2 | For the community, it is an interesting question as to whether (micro)organisms simply passively modify local environmental conditions, thus rendering metastable what would otherwise be thermodynamically unstable | The question of the production of metastable aragonite by organisms was not an issue in this study. |

| | | |
|---|---|---|
| | phases, or if, conversely, such organisms actively form unstable phases by pumping electrons or redox-sensitive molecules against gradients [1,2]. In any case, we would expect that the deviation from thermodynamic equilibrium would be relatively small. In other words, one would expect that energetically expensive reactions would also be "expensive" for the organism involved. Casella et al.'s findings seem to nicely support this hypothesis. It takes a significant increase in temperature to convert the aragonite into thermodynamically stable calcite. However, this insight does not answer the important question: why do organisms produce metastable aragonite and not the stable phase calcite? | |
| Comment 3 | Another interesting point is the potential occurrence of spatial patterns in the distribution of replacement reaction rates. The existence of a characteristic porosity distribution within the shell material is able to foster heterogeneous material fluxes. Thus, in future studies we expect direct mapping of the reaction rate in order to identify rate components that form the heterogeneous overall reaction rate. Such rate components provide important input values for the simulation of fluid-solid reactions [3]. | We thank the reviewer for his highly valuable suggestions and will perform future experiments and simulation studies accordingly. |
| **REVIEWER 3** | | |
| Comment 1 | Does the paper address relevant scientific questions within the scope of BG? YES, particularly taphonomy of shells. | ---- |
| Comment 2 | Does the paper present novel concepts, ideas, tools, or data? YES, particularly, the focus is highly innovative. | --- |

**Reply and revision of bg-2016-355 Casella et al.**

| | | |
|---|---|---|
| Comment 3 | Are substantial conclusions reached? YES, it adds substantial knowledge to the particular process studied. | --- |
| Comment 4 | Are the scientific methods and assumptions valid and clearly outlined? YES, in general, although I find that some conceptual aspects should be explained more in length (see detailed comments. | --- |
| Comment 5 | Are the results sufficient to support the interpretations and conclusions? YES, in general (but see detailed comments). | --- |
| Comment 6 | Is the description of experiments and calculations sufficiently complete and precise to allow their reproduction by fellow scientists (traceability of results)? YES, in general, although additional data on the composition of the fluids (Table 1) are needed | Chemical and experimental information on hydrothermal experiments utilised in the present study are given in Table 1. |
| Comment 7 | Do the authors give proper credit to related work and clearly indicate their own new/original contribution? YES, in general (but see detailed comments). | --- |
| Comment 8 | Does the title clearly reflect the contents of the paper? YES. | --- |
| Comment 9 | Does the abstract provide a concise and complete summary? YES. | --- |
| Comment 10 | Is the overall presentation well-structured and clear? YES. | --- |
| Comment 11 | Is the language fluent and precise? YES | --- |
| Comment 12 | Are mathematical formulae, symbols, abbreviations, and units correctly defined and used? Some technical abbreviations (EPMA, TAP, PET, LPET, LLIF: : :) should be defined for non-specialists.
L. 173: "…SEM visualization and EPMA…" | Following abbreviations, which are not common, were defined:
"…SEM visualization and electron Probe Micro Analysis (EPMA)…"
"…TAP (thallium acid pthalate) crystal…"
"…PET (pentaerythritol) crystal…"
"…LPET (large pentaerythritol) crystal…LLIF (large lithium flouride) crystal…." |

| | L. 190: "…TAP crystal…"
L. 191: "…PET crystal…"
L. 192: "…LPET…LLIF…" | |
|---|---|---|
| Comment 13 | Should any parts of the paper (text, formulae, figures, tables) be clarified, reduced, combined, or eliminated? NO. | --- |
| Comment 14 | Are the number and quality of references appropriate? YES. | --- |
| Comment 15 | Is the amount and quality of supplementary material appropriate? YES. | --- |
| Comment 16 / L. 133 | "…create local micro-environments for physiological generate of their composite…"

'for physiological generate', this is an odd expression | Changed to:
"… living organisms create local micro-environments for physiological generation of their composite…" |
| Comment 17 / L. 138 | "…dealt with pre-Neogene *A. islandica* specimens…"

'pre-Neogene A. islandica', certainly not this species. | Changed to:
"…dealt with *A. islandica* specimens…" |
| Comment 18 / L. 157 | "…samples as well as the mineral in reference, geologic,…" | Changed to:
"…samples as well as the mineral part in the reference specimens, i.e. geologic, and non-biological aragonite, …" |
| Comment 19 / L. 159 | "…with glass knifes to…" | Changed to:
"…with glass knives to…" |
| Comment 20 / L. 193 | "…and hematite (Fe), and were used…" | Changed to:
"…and hematite (Fe), were used…" |
| Comment 21 / L. 198-199 | "… Hydrothermal alteration experiments mimicked burial diagenetic (and meteoric) alteration of recent *A. islandica* under controlled laboratory conditions. Chemical and isotopic compositions of experimental fluids are given in Table 1.…" | The used fluid compositions are empirical values and may differ from those which can be found in nature. Nevertheless, we decided to use 10 mM of NaCl for the simulated meteoric fluid and 100 mM of NaCl and 10 mM of $MgCl_2$ for the simulated burial fluid due to the fact that burial waters have more Na compared to meteoric fluids but also some Mg. |

| | | |
|---|---|---|
| | This initial bit needs to be expanded further. More data on the composition of fluids and why their compositions have been chosen are necessary. | Moreover, both simulated diagenetic fluids were spiked with $^{18}$O-depleted oxygen in order to trace fluid-solid exchange reactions. The isotopic aspect of the performed hydrothermal alterations were analysed by other project members (see Ritter et al., 20016, accepted manuscript in Sedimentology) in order to trace fluid-solid exchange reactions. Due to the fact that the present manuscript does neither focus, nor deal with isotopic changes, we deleted the mentioned isotopic fluid compositions. |
| Comment 22 / L. 207 | "… this temperature regime is far beyond natural meteoric diagenetic environments but are typical for the burial realm…."

At least a reference is needed before the period. | References added, see below:
"…Obviously, this temperature regime is far beyond natural meteoric diagenetic environments (Lavoie and Bourque, 1993) but are typical for the burial realm (Heydari, 1997). …" |
| Comment 23 / L. 237-238 | "…. The shell portion facing seawater, indicated with yellow stars in Figs. 1A and 1B, consists of aragonite crystal units in the 5 µm size range (Fig. 2A). This shell portion is highly porous (Fig. 1B) with pore diameters in the range of a few (< 5) micrometres…"

This feature is not visible at the magnification of Fig. 1B. | Changed to:
"… This shell portion is highly porous (see the white dotted features in Fig. 1B), pore diameters range within a few micrometers…"

See Fig.A2 |
| Comment 24 / L. 240 | "… dense and is composed of very small aragonite crystallites with sizes of less than 1 µm (Fig. 2B) and contains very few pores…."

Fig. 2B is heavily etched. Accordingly, any conclusion about the density of pores is doubtful. | Changed to:
"… is dense and is composed of very few small aragonite crystallites with pore sizes of less than 1 µm (Fig. 2B)…." |
| Comment 25 / L. 260 | "… shell of A. islandica is composed of nanoparticles that are a few tens of nanometres in diameter…"

Again, the nanogranules I Fig. 3C are etching and not original features. | Changed to:
"… In order to check the validity of nanoscale structural features observed in pristine Arctica islandica shells, we prepared non-biological aragonite grown from solution in a similar way (microtome cut, polished, etched slightly, only for 180 seconds, critical point dried). As it is well visible in |

**Reply and revision of bg-2016-355 Casella et al.**

| | | Fig. A3 etch pits develop, the presence in aragonite grown from solution and assembly of nanoparticles is not evident…." |
|---|---|---|
| Comment 26 / L. 260-261 | "…the shell of *A. islandica* is composed of nanoparticles that are a few tens of nanometres in diameter (white arrows in Fig. 3C). These are co-aligned to form mesocrystals - here in the 1-5 µm size range…."

'These are co-aligned to form mesocrystals', probably true, but not demonstrated here. I advise deletion. | These are co-aligned to form mesocrystals – probably true but not demonstrated here.
The term mesocrystal was deleted. |
| Comment 27 / L. 284-287 | "…At higher magnification a multitude of tiny holes (indicated with yellow arrows in Figs. 5C, 5D) become readily visible. In the unaltered shell these holes were filled with the network of biopolymer fibrils interconnecting the mineral units (e.g. Fig. 3B)…."

'holes', do the authors refer to the membranes between mineral units? Please, be precise. | "… At higher magnification a multitude of tiny holes (indicated with yellow arrows in Figs. 5C, 5D and enlarged in Figs. A7A and A8B)  become readily visible …"

We changed the text accordingly and included two new Figures in the appendix. |
| Comment 28 / L. 302-303 | "… Hydrothermal treatment of *A. islandica* at 100 °C does not produce a significant change in aragonite co-orientation pattern, texture, grain fabrics, and grain size distributions. The pristine and the hydrothermally treated shell materials appear to be quite similar…."

the coincidence in the orientations of the c-axes of aragonite and calcite, before and after alteration, is remarkable. I would like the authors to comment on this. Could there be some kind of epitaxial growth of calcite on the aragonite? | In the first few days of alteration the pristine aragonite microstructure predetermines to some degree the microstructure of the newly formed product. |
| Comment 29 / L. 303 | "…values must be attributed to fact…" | Changed to:
 "…values must be attributed to the fact…" |

**Reply and revision of bg-2016-355 Casella et al.**

| Comment 30 / L. 304-305 | "…EBSD scan fields on the different samples in exactly corresponding spots with respect to the outer shell margin and to the patterns of growth lines…"

'and to the patterns of growth lines', what is meant with this exactly? please reword. | "…EBSD scan fields on the different samples in exactly corresponding spots with respect to the outer shell margin and to the patterns of annual growth lines…" |
|---|---|---|
| Comment 31 / L. 305-306 | "…The shell is not uniformly textured. In particular, the slight preferred crystallographic orientation of the a*-axes (the (100) plane normal) in Fig. 6C is a singular case, while c-axis preferred orientation is otherwise dominant (Fig. 6C)…."

There is something wrong here. The maximum for a* is more or less coincident with the maximum of c*, and does not conform with the orientations of b*. The authors should review their data to make sure that the a* pole figure is the correct one. | We checked our data shown in Fig. 6 amended the text accordingly to the reviewers comment. |
| Comment 32 / L. 306 | "…while c-axis preferred…." | Changed to:
"…while the c-axis preferred…" |
| Comment 33 / L. 329-330 | "…The MUD values for the newly formed calcite material are high (Figs. 9, 10), but this is related to the fact that within the range of the EBSD scan just a small number of large, newly formed, individual crystals is encountered. …"

The explanation based on the small number of crystals mapped is not enough. Again, as above, is there the possibility of epitaxial growth? | We cannot change the number of newly formed crystals. The investigation of epitaxial growth of the product on the educt was not the major point of investigation in our study. |
| Comment 34 / L. 351 | "…Characteristics of the grains obtained by reaction at 100 °C and 175 °C…" | We changed the title according to the comment of the reviewer. |

|  | Subsection 4.1.- I do not think that the title (particularly 'Characteristics of the grains') actually conforms to the contents. |  |
| --- | --- | --- |
| Comment 35 / L. 372 | "…Calcite nucleation occurs (and replacement reaction proceeds) where the hydrothermal fluid is in contact with the bio-aragonite: at the surfaces of the shell, in pores and along growth lines (Figs. 7, 8, 9). …"

'Figs. 7, 8, 9', I find that only Fig. 9b is relevant in the context. | We changed the text accordingly. |
| Comment 36 / L. 394-395 | "…). A high degree of internal strain is indicated by blue colours, while light green to yellow colours highlight shell portions where local misorientation is low…"

May be I do not understand properly, but in the scale of misorientations, blue colors imply very low values, whereas green, yellow and red are increasing values. Isn't it the other way round? | We agree with the reviewer and the text was changed (see below):

Blue colours indicate the absence of measurable internal strain, while light green to yellow colours highlight areas where local misorientation is larger than experimental resolution. |
| Comment 37 / L. 400-401 | "…all calcite grains contain numerous small calcite crystallites, a clear cause for the occurrence of internal strains…"

'numerous small 400 calcite crystallites, a clear cause for the occurrence of internal strains'. Something is inconsistent here. Crystallite boundaries are defined by misorientations values above 5°, but this is not real, just a convenience. At the positions of 'crystallites' (defined in this way) misorientations values just peak. Therefore, they are not real and cannot be the origin of | We revised the statement in the text. |

| | misorientations. To demonstrate that they are real, other techniques (e.g., TEM) are necessary. | |
|---|---|---|
| Comment 38 / L. 411 | "…curvilinear structures in the cross section (white arrows in Figs. 13A, 13B) and …" | Changed to:
 "…curvilinear structures in the cross section (white arrows in Figs. 13A, 13C) and …" |
| Comment 39 / L. 412 | "…correspond to subgrain boundaries within the newly formed calcite crystals. These boundaries do not appear to heal or to disappear with an increased alteration time, an indication again of the little effect of alteration duration on the fabric and internal structure of calcite grains crystallized from *Arctica islandica* shell bioaragonite…"

The linear structures in 13C at consistent angles argue for crystalline structures ({104} faces of calcite?). | We do not understand the comment of the reviewer and did not change our text. We think that what we state in the text is right. |
| Comment 40 / L. 422-424 | "…Grains obtained from alteration experiments with meteoric fluid show a significantly higher degree of mean misorientation (up to 10 degrees, black arrows in Fig. 14A), compared to that in grains that grew in burial solution.…"

This conclusion does not seem significant in the absence of a statistical analysis, and in view of the low number of data which deviate from the trend in 14B. | We do observe this feature in duplicate samples. |
| Comment 41 / L. 426-428 | "…Thus, larger crystals in the experiment with burial solution result from a smaller number of calcite nuclei. Again, this supports the idea that $Mg^{2+}$ inhibits calcite nucleation.…" | We expanded this statement in subchapter 4.5 (see also below):

We attribute the large calcite grains to the nucleation rate: The crystals growing from each nucleus consume the aragonite educt (the precursor, original aragonite) until they abutted each other. Thus, larger crystals in the experiment with burial solution result from a smaller number of |

| | I cannot follow the statements contained in the two sentences. | calcite nuclei, which may be attributed to the presence of aqueous Mg in the experimental fluid. Note here, that both the reduction of Mg concentration in the reactive fluid, compare to that in the initial burial fluid (see Table 1), as well as speciation calculations, suggest that the formation of Mg-bearing carbonate minerals (magnesite and/or dolomite) is likely possible to occur at the experimental conditions. Indeed, we observe small patches of newly formed Mg-rich carbonates (Fig. A13). The formation of such minerals occurs at lower rates compared to pure Ca-bearing carbonates owing to the slow dehydration of aqueous Mg that is required prior to its incorporation in the crystal (e.g. Mavromatis et al., 2013) even at temperature as high as 200 °C (Saldi et al., 2009; 2012). |
|---|---|---|
| Comment 42 / L. 431 | Subsection 4.3.- In my opinion, the paragraph immediately above could be included within this subsection. | The subsections of the discussion were changed. |
| Comment 43 / L. 451 | "…6C - D illustrate that the (newly formed) calcite product reveals an internal structure that is very reminiscent of the original bioaragonite/biopolymer composite…"

Fig. 6C-D is not relevant within the context. The images supporting this statement have to be referred unequivocally. | We changed the sentence as follows:
"…As the band contrast and orientation maps of Figs. 6A-C illustrate…" |
| Comment 44 / L. 472 | "…as it a …" | Changed to:
"…as it is a …" |
| Comment 45 / L. 475 | '"…skeleton-forming mineral in…" | Changed to:
'"…skeleton-forming minerals in…" |
| Comment 46 / L. 485 | "…allowing molluscs preservation…" | Changed to:
"…allowing mollusc preservation…" |
| Comment 47/ L. 487-493 | "…In this perspective, the laboratory-based hydrothermal alteration experiments performed here | The text has been changed according to the suggestions of the reviewer. |

|  | offer very interesting insights into the fate of the aragonitic or bimineralic hard tissues that escape early dissolution during shallow burial and have the potential to enter the fossil record, a matter relatively neglected so far. In particular, the resistance of biogenic aragonite to replacement by calcite up to temperature of 175 °C during hydrothermal alteration offers an additional explanation for the preservation of aragonitic shells/skeletons, besides the taphonomic windows envisaged by Cherns et al. (2008). The results of our experiments neatly explain the observation that the mollusc fossil record is good and allows restoration of evolutionary patterns…."

Somewhere around here, the authors should state that the change from aragonite to calcite does not always proceed in this
way. For instance, there is also the possibility that total dissolution followed by precipitation by calcite (or other minerals, gypsum, pyrite) takes place. |  |
|---|---|---|
| Comment 48 / L. 522-524 | "…Between two tipping points, one between 50 and 60 °C, the other between 160 and 180 °C, aragonite appears to precipitate from supersaturated aqueous solutions rather than calcite, such that the hydrothermal treatments of aragonite within this temperature bracket do not yield calcite…."

Please, state clearly that the recognition of the first tipping point (50 °C to 60 °C) is not derived from the present study. | The text has been changed accordingly. |

**Reply and revision of bg-2016-355 Casella et al.**

| | | |
|---|---|---|
| Comment 49 / L. 598 | "…Dr. E. M. Harper, Prof. U. Brand…"

Dr. E.M. Harper is also a coauthor; please remove | Request was removed.
"…, Prof. U. Brand…" |
| Comment 50 / L. 654 | "…., and Ramirez,…" | Changed to:
"…Ramírez…" |
| Comment 51 / L. 657 | "…Ramirez-Rico, J., Gonzalez-Segura, and Sanchez-Navas…" | Changed to:
"…Ramirez-Rico, J., González-Segura, and Sánchez-Navas…" |
| Comment 52 / L. 666 | "…Choudens-Sanchez, V., and Gonzales, L. A…." | Changed to:
"…Choudens-Sánchez, V., and Gonzáles, L. A…." |
| Comment 53 / L. 670 | "…genera Glycymeris, Aequipecten and Arctica,…" | Changed to:
"…genera *Glycymeris, Aequipecten* and *Arctica*,…" |
| Comment 54 / L. 677 | "…Perez-Huerta…" | Changed to:
"…Pérez-Huerta…" |
| Comment 55 / L. 718 | "…of bivalvemolluscs…" | Changed to:
"…of bivalve molluscs…" |
| Comment 56 / L. 775 | "…Arctica islandica…" | Changed to:
"…*Arctica islandica*…" |
| Comment 57 / L. 826 | "…Zetterstrom…" | Changed to:
"…Zetterström…" |
| Comment 58 / L. 1023-1024 | "…Readily observable are minute round holes within the mineral units (yellow arrows in B, C, D) that were filled in the pristine shell, prior to alteration, by biopolymer fibrils…."

How do the authors know that the minute holes were filled with biopolymer fibrils. I do not think the statement can be drawn with the techniques used in this study. I will remove it. | "…For further details concerning the interlinkage between mineral units and nanoparticles with organic matrices, see Figs. A7 and A8…." |
| REVIEWER 4 | | |

**Reply and revision of bg-2016-355 Casella et al.**

| | | |
|---|---|---|
| Comment 1 | First paragraph of 'Abstract' may be moved to 'Introduction'. | We do not agree. |
| Comment 2 / L. 52 | ".... Experimental conditions were between 100 °C and 175 °C..."

I understand from the text that experimental conditions were at 100°C and 175°C, and not 'between' these temperatures. | Changed to:
"...Experimental conditions were between 100 °C and 175 °C with the main focus on 100 °C and 175 °C,..." |
| Comment 3 / L. 58 | "... In all experiments below 175 °C there are no signs..."

'. . .in all experiments below 175 °C. . .' This implies that the described observation occurs at, e.g., 174 °C. Be precise. | The text has been changed accordingly (see below):

".... In all experiments up to 174 °C there are no signs of a replacement reaction of shell aragonite to calcite in X-ray diffraction bulk analysis. At 175 °C the replacement reaction started after a dormant time of 4 days ..." |
| Comment 4 / L. 72 | "...The absence of aragonite replacement by calcite at temperatures lower than 175°C contributes to explain why aragonitic or bimineralic shells and skeletons have a good potential of preservation..."

Again, this statement implies that 175 °C is a critical threshold temperature, while it is simply the temperature at which many of the experiments were run. | This finding has been explained in the text in great detail. The alteration temperature of 175 °C corresponds to temperatures that are present at shallow burial diagenesis. |
| Comment 5 / L. 83 | "... They are widespread in the fossil record and are sensitive to changes in seawater composition -which they record with a limited vital effect (e.g. Brand et al., 2003; Parkinson et al., 2005; Schöne & Surge, 2012; Brocas et al., 2013) ..." | We follow the suggestion of the reviewer and changed that sentence accordingly.

"...They are widespread in the fossil record and are sensitive to changes in seawater composition  (e.g. Brand et al., 2003; Parkinson et al., 2005; Schöne & Surge, 2012; Brocas et al., 2013) ..." |

| | | |
|---|---|---|
| | '. . .limited vital effect. . .' This is certainly an over-simplification, as 'vital effects' depend on the complexity of the biomineralisation process of the organism, the element or isotope system considered; and can be negligible or significant. | |
| Comment 6 / L. 91-106 | "… we performed laboratory-based alteration experiments with *Arctica islandica* shells with the aim to obtain time series data sets. The bivalve *A. islandica* has been studied in several scientific disciplines, i.e. biology (Morton, 2011; Oeschger and Storey, 1993; Taylor, 1976; Strahl et al., 2011), ecology (Beal and Kraus, 1989; Kilada et al., 2007; Lewis et al., 2001; Ridgway et al., 2012; Thórarinsdóttir and Einarsson, 1996), gerontology (Abele, 2002; Ridgway and Richardson, 2011; Strahl, 2007), pollution monitoring (Krause-Nehring et al., 2012; Palmer and Rand, 1977; Swaileh, 1996) and shellfisheries management (Adelaja et al., 1998; Harding et al., 2008; Thórarinsdóttir and Jacobson, 2005). *A. islandica* has also gained profound attention in paleoclimatology due to its long lifespan and its use as a high-resolution long-term archive (e. g. Schöne, 2004; Schöne, 2005a, 2005b; Wanamaker et al., 2008; Marchitto et al., 2000, Butler et al., 2009, Wanamaker et al., 2011, Karney et al., 2012 Schöne, 2013, Butler et al., 2013). On the long-term perspective, *A. islandica* plays an important role in palaeontology, not only as a Neogene palaeoecological and palaeoclimatic archive (e.g. Schöne, 2004; Schöne, 2005a, 2005b; Wanamaker et al., 2008; Marchitto et al., 2000, Butler et al., 2009, Wanamaker et al., 2011, Karney et al., 2012 Schöne, 2013, Butler et al., 2013, Crippa et | We shortened the text according to the suggestion of the reviewer. |

| | | |
|---|---|---|
| | al., 2016), but also as a biostratigraphic tool. Formerly considered a marker for the Pliocene-Pleistocene boundary (Raffi, 1986) in the Mediterranean region, its first appearance is now regarded as an indicator of the Gelasian-Calabrian (Early Pleistocene) boundary, around 1.7 Ma (Crippa & Raineri, 2015). …"

Do we need to know that much about *A. islandica* in order to understand the significance of this paper, which is more concerned about general aspects of biological aragonite diagenesis? I suggest shortening this paragraph as it distracts from the main focus of the paper. | |
| Comment 7 / L. 124 | " . . .natural or synthetic aragonite. . ." | Changed to:
" . . .geological or synthetic aragonite. . ." |
| Comment 8 / L. 128-130 | "… Metzger & Banard (1968) and Perdikouri et al. (2011, 2013) investigated aragonite blocks or single crystals and report that temperatures IN EXCESS of 160-170 °C are required to transform the aragonite to calcite within a couple of days, whereas BELOW 160 °C aragonite remains present over many weeks. …"

edit capitalized words | Changed to:
"… Metzger & Banard (1968) and Perdikouri et al. (2011, 2013) investigated aragonite blocks or single crystals and report that temperatures *in excess* of 160-170 °C are required to transform the aragonite to calcite within a couple of days, whereas *below* 160 °C aragonite remains present over many weeks. …" |
| Comment 9 / L. 132-140 | "… During biomineralisation living organisms create local micro-environments for physiological generate of their composite hard tissues. After the death of the organism all tissues become altered by equilibration with the surrounding environment - part of the complex set of processes called diagenesis. Thus, as diagenetic alteration proceeds, the species-specific fingerprint of | We do not agree with the reviewer. |

| | the biogenic structure disappears and is replaced by inorganic features. Despite the fact that the evolutionary line of *A. islandica* dates back to the Jurassic (Casey, 1952) only a limited number of studies have dealt with *A. islandica* specimens due to the thermodynamically unstable nature of their aragonitic shells…."

 These statements should be at the beginning of the 'Introduction'. | |
|---|---|---|
| Comment 10 / L. 199-200 | No isotope data are reported in table 1. | All fluids were spiked with $^{18}$O-depleted oxygen in order to trace fluid-solid exchange reactions and isotopic studies investigated by Ritter et al., 2016. |
| Comment 11 / L. 200 | "… All fluids for this were spiked with $^{18}$O-depleted oxygen in order to trace fluid-solid exchange reactions…."

 Something wrong with this sentence; as no isotope data are reported or discussed, sentence could be omitted. | See comment above |
| Comment 12 / L. 205 | "…oven at temperatures between 100 °C and 175 °C for different periods of time between one day and 84 days (see Table 2)…"

 According to table 2, samples were experimentally altered during 7, 28, and 84 days at temperatures of 100 or 175°C. Be precise. | Changed to:
 "…oven at temperatures between 100 °C and 175 °C for different periods of time ranging between one day and 84 days ( see Table 1, Fig. A11 and Table 2 for experiments focussing on 100 °C and 175 °C)…" |
| Comment 13 / L. 232 | "…Figures 1 to 5 show characteristic ultrastructural features of the shell of modern *A. islandica*. Images of the pristine shell are given in Figs. 1-3, while Figs. 4 and 5 present structural features of the hydrothermally…" | We changed the text accordingly. |

| | | |
|---|---|---|
| | It may be advisable to refer to inner and outer shell layers, consistent with the terminology in Fig. A1. 'Shell portion facing seawater' is somewhat misleading, as only the surface of the shell may be in contact with seawater, and is additionally protected by the periostracum (see Fig. A1). | |
| Comment 14 / L. 273-275 | "…Relative to neighbouring shell increments, the Sr content along the growth lines is always higher. Maximal concentrations (along annual growth lines) in pristine and altered shells vary between 0.4 and 0.6 wt% Sr (Figs. A3, A4, A5)…."

 No explanation is offered for the high Sr-concentrations along growth lines throughout the text. Is Sr part of the organic matrix, or the mineral aragonite? | "… Relative to neighbouring shell increments, the Sr content along the growth lines is always higher (Shirai et al., 2014)…."
 According to Shirai et al. (2014) Sr is contained in the mineral part of the shell. |
| Comment 15 / L. 288 | I suggest reporting the EBSD results in a separate section with its own subheading. | We do not agree with the reviewer, and need to present data from all methods together in one major results section. |
| Comment 16 / L. 315 | "…Using X-ray diffraction (XRD) we obtained an overview of the kinetics of the *A. islandica* biogenic aragonite to calcite transition …"

 The observation of calcite formation is a very fundamental result that should be higher up in the 'Result' section, not at its end, hidden in the description of aragonite microfabrics. | We do not agree with the reviewer, and need to present data from all methods together in one major results section. |
| Comment 17 / L. 368 | "…reaction in our 100 °C treatments is related to inhibition of calcite nucleation (Sun et al., 2015), a mechanism that has rarely been rigorously explored…" | Possibly, this issue was not a major point of investigation in our study. |

| | Is there evidence that supersaturation was reached in the 100°C experiments? The experimental fluids should be initially undersaturated, no aragonite dissolution is reported from the experiments, and the data in Table 1 are not sufficient to estimate saturation states. | |
|---|---|---|
| Comment 18 / L. 370 | "… At 175 °C the replacement reaction of biological aragonite to coarse-grained calcite occurs rapidly; it starts after a dormant period of about 4 days and proceeds rapidly almost to completion after 3 more days (Figs. A3, 13A)…"

Fig. A3 shows alteration at 100°C, and not at 175°C. | Changed to:
 "… At 175 °C the replacement reaction of biological aragonite to coarse-grained calcite occurs rapidly; it starts after a dormant period of about 4 days and proceeds rapidly almost to completion after 3 more days (Figs. A11, 8)…" |
| Comment 19 / L. 375 | **"…4.2 The time lag of aragonite to calcite replacement reaction at 175 °C…** The several-day dormant…. critical size can form…"

Only the first paragraph of this section is about the time lag of calcite replacement, while 4/5 of the section are about orientation of newly formed calcite crystals. The latter discussion is very important, and certainly deserves its own subheading. | We changed the title of the subparagraph 4.2. |
| Comment 20 / L. 445 | "…We observed that the fluids used (artificial meteoric and/or burial fluids) cause only a minor difference in replacement reaction kinetics in our experiments, with the $MgCl_2$-bearing artificial burial fluid reducing the nucleation rate of calcite, thus, leading to the observed significantly larger calcite crystals in the recrystallised product…" | We do not have any data on the partial pressure of $CO_2$. Concerning the further comments of the reviewer, we slightly modified the discussion. |

| | | |
|---|---|---|
| | The pH of the solution and/or CO2 partial pressure must be considered important, as they control the solubility of the original phase. Experiments of the current study were run with fluids with relatively high pH values and presumably very low pCO2 (no data provided). This should be discussed when comparing the present result with older studies that observed more rapid reactions with low-pH solutions (lines 435-436). | |
| Comment 21 / L. 470-493 | **"…4.4. A paleontological perspective of our laboratory-based hydrothermal alteration experiments…", complete part**

I disagree with many statements in this section. Aragonite dissolution is known from many near-surface environments and occurs essentially at surface temperatures, e.g. in the vadose environment. Factors other than temperature, e.g. pH of vadose fluids, anoxic diagenetic environments with sulfate reduction etc., have to be considered. Much has been written about a taphonomic bias related to shell mineralogy as well as about carbonate diagenesis, and these complex issues cannot be adequately addressed in such a short paragraph. It is not strictly relevant to the main topic addressed in this paper and I suggest omitting it. | The focus of this paragraph is to underscore the significance of the experiments for understanding the palaeontological record and not to discuss the factors that control aragonite dissolution, which, as the reviewer says represent, a very complex and debated subject. We think that it is very important to link the results of these experiments on extant bivalves to the possible fate of the main aragonite biomineralisers, which are vulnerable to early dissolution and yet can be found in the fossil record. This paragraph can raise the attention of palaeontologists and fill the gap, which is invariably present, between those who study recent taxa or recent processes and the researchers who study the fossil record, widening the audience of the readers of this paper and providing new ideas for future collaboration. So the message contained in this paragraph is different to what inferred by the reviewer. More than discussing aragonite dissolution, this paragraph seeks to offer an additional explanation for the preservation of aragonitic shells/skeletons and thus their rather complete fossil record. We have slightly changed the text to make our message clearer.

We modified chapter 4.4 (see below and revised manuscript).

The alteration experiments of recent *A. islandica* under controlled laboratory conditions are very important from a palaeontological perspective as they reproduce burial diagenetic conditions. The |

| | | understanding of the diagenetic processes which control organism hard tissue preservation in fact a fundamental prerequisite to taxonomic, taphonomic, palaeoecological, and biostratigraphic studies (e.g. Tucker, 1990). Most organisms have hard tissues composed of calcium carbonate, and its metastable form, aragonite, is one of the first biominerals produced at the Precambrian-Cambrian boundary (Runnegar & Bengtson, 1990), as well as one of the most widely used skeleton-forming minerals in the Phanerozoic record and today; in fact, aragonitic shells/skeletons are produced by hyolithids, cnidarians, algae, and by the widespread and diversified molluscs. |
|---|---|---|
| | | Several studies (Cherns & Wright, 2000; Wright et al., 2003; Wright & Cherns, 2004; James et al., 2005) have underscored that Phanerozoic marine faunas seem to be dominated by calcite-shelled taxa, the labile aragonitic or bimineralic groups being lost during early diagenesis (in the soft sediment, before lithification), potentially causing a serious taphonomic loss. Considering that most molluscs are aragonitic or bimineralic, this loss could be particularly detrimental both for palaeoecological and biostratigraphic studies. However, it has been shown that the mollusc fossil record is not so biased as expected (Harper, 1998; Cherns et al., 2008). This is due to high frequency taphonomic processes (early lithification/hardgrounds, storm plasters, anoxic bottoms, high sedimentation rates) that. throughout the control of organic matter content and residence time in the taphonomically active zone, produce taphonomic windows allowing mollusc preservation (James et al. 2005; Cherns et al., 2008). Even if the factors that control aragonite dissolution are multiple and their interpretation is complex. |
| | | The laboratory-based hydrothermal alteration experiments performed here offer very interesting insights into the fate of the aragonitic or bimineralic hard tissues that escape early dissolution during shallow burial and have the potential to enter the fossil record. In particular, the |

**Reply and revision of bg-2016-355 Casella et al.**

| | | |
|---|---|---|
| | | resistance of biogenic aragonite to replacement by calcite up to temperature of 175 °C during hydrothermal alteration offers an additional explanation for the preservation of aragonitic shells/skeletons once they have escaped early dissolution. The results of our experiments neatly explain the observation that the mollusc fossil record is good and allows restoration of evolutionary patterns. |
| Comment 22 / L. 499 | "… and contains mineral units in the 1-5 µm size range, the inner shell layers (which are closer to the soft tissue of the animal) are characterised by a dense shell structure…"

While it is correct that the inner shell layer is closer to the soft body of the bivalve, this is probably quite irrelevant for the observed differences in shell structure and porosity. Inner and outer shell layers are precipitated from the extrapallial fluid, i.e. at about the same distance from the tissue of the mantle margin where the shell is formed. | Changed to:
Next to the soft tissue of the animal → inner shell layer / portion
Next to the seawater → outer shell layer / portion / section |
| Comment 23 / Table 1 | Please clarify that all data (except Mg) are referring to fluids before the experiment. Did changes of any of the tabulated items occur during the experiment, or was this not analyzed? | All given values are valid for the fluids after the experiments as the stock fluids prior to the experiment were not analysed. |
| Comment 24 | 'hydrothermal': rephrase to 'solution', 'diagenetic fluid', or 'experimental' fluid. 'Hydrothermal' is not synonymous with 'hot water'. | Changed to:
"experimental fluid(s)" |
| Comment 25 | What about the importance of the organic matrix, its decay, and its role of providing pathways for fluids and in changing the chemistry of fluids? The time lag observed before nucleation of calcite crystals could be related to the time required for disintegration of the organic | The disintergration of the organic matrix is not mainly responsible for the time lag before nucleation of new calcite crystals, as the organic matrix disinteragrates very quickly and well below 100 °C. |

| | matrix, formation of local acidic conditions due to the addition of $CO_2$ from the inorganic breakdown of organic molecules, and the permeability created in the process. | |
|---|---|---|

---

## Author Comment (AC2)

[revised manuscript text omitted]
α, Mg-Kα, and Na-Kα were measured on a TAP (thallium acid pthalate) crystal. Ca-Kα, and Ba-Lα were measured on a PET (pentaerythritol) crystal, whereas Kα emission lines of P, and Cl were measured on a LPET (large pentaerythritol) crystal. Lα emission lines of Mn, and Fe were detected with a LLIF

(large lithium fluoride) crystal. A step size in the range of 1-2 µm with a dwell time of 150 ms was chosen for the element mappings. Celestine (Sr), dolomite (Ca, Mg), ilmenite (Mn), apatite (P), albite (Na), benitoite (Ba), vanadinite (Cl), and hematite (Fe) were used as standard materials. Matrix correction was carried out using the

PAP procedure (Pouchou and Pichoir, 1984).

**2.2.3 Alteration experiments**

Hydrothermal alteration experiments mimicked burial diagenetic (and meteoric) alteration of recent *A. islandica* under controlled laboratory conditions. Chemical and experimental information on hydrothermal experiments utilised in the present study are given in Table 1. All fluids were spiked with $^{18}$O-depleted oxygen in order to trace fluid-solid exchange reactions and isotopic studies investigated by Ritter et al., 2016.
Details of the experimental protocol can be found in Riechelmann et al. (2016). Briefly, pieces (2 x 1 cm) of recent *A. islandica* specimens were placed in a PTFE liner together with 25 mL of either the meteoric (10 mM NaCl aqueous solution) or burial fluid (100 mM NaCl + 10 mM MgCl$_2$ aqueous solution) and sealed with a PTFE lid. Each of the PTFE liners was placed in a stainless steel autoclave, sealed and kept in the oven at temperatures of 100 °C, 125 °C, 150 °C and 175 °C for different periods of time ranging between one day and 84 days (see Table 1, Fig. A11 and Table 2 for experiments, main focus was on 100 °C and 175 °C). Obviously, this temperature regime is far beyond natural meteoric diagenetic environments (Lavoie and Bourque, 1993) but are typical for the burial realm (Heydari, 1997). Nevertheless, elevated fluid temperatures were applied to meteoric experiments, too, as reaction rates under surface conditions are too slow for experimental approaches. After the selected time period, an autoclave was removed from the oven, cooled down to room temperature and then opened. The aqueous fluid that had passed through a 0.2 μm cellulose acetate filter was subjected to further chemical and isotopic analyses. Recovered solids were dried at 40 °C overnight.

**2.2.4 X-ray diffraction analysis**

[revised manuscript text omitted]

**4.1 Driving force in comparison to nucleation barrier**

In sedimentary environments the fate of metastable biogenic aragonite or high-Mg calcite can follow two scenarios: (1) the metastable biogenic matter can be completely dissolved and removed by fluid transport to form molds that are later filled by cement or other neogenic minerals or (2) the metastable minerals may be replaced by stable low-Mg calcite in-situ, by a process which involves dissolution of the metastable phase into a nano- to micro-scale local fluid volume (e.g. a thin fluid film) from which the stable low-Mg calcite precipitates without long-range transport (Brand & Veizer, 1980, 1981; Brand, 1991, 1994; Bathurst, 1994; Maliva 1995, 1998; Maliva et al., 2000; Titschak et al., 2009, Brand et al., 2010).). The latter process may preserve original morphological boundaries and microstructures such as prisms, tablets and fibres in bivalve shells. The replacement reaction from aragonite to stable low-Mg calcite is driven by the higher solubility (free energy) of the metastable phase compared to the the stable phase. Thus, as the replacement reaction proceeds, the reactive, percolating experimental or
diagenetic pore fluid becomes undersaturated with respect to aragonite owing to its relative supersaturation with
respect to calcite, the less soluble mineral phase in the system. The maximal supersaturation $\Omega_{max}$ with respect to
calcite, which can be obtained in a fluid, which draws its calcium and carbonate ions from the dissolution of
aragonite, can be described as:

$$\Omega_{max} = \frac{K_{sp}\,(Aragonite)}{K_{sp}\,(Calcite)} \qquad (1)$$

, where $K_{sp}$ stands for the ion activity products of the respective phase in the relevant pore fluid. The free energy
difference or thermodynamic driving force is given by $\Delta G_{max}$ = - RTln $\Omega_{max}$ . To obtain an estimate we used the
data of Plummer & Busenberg (1982) and calculated the solubility products for calcite and aragonite for 25 °C,
100 °C, and 175 °C (Fig. 12). The maximal supersaturations $\Omega_{max}$ thus obtained are 1.39 (25 °C), 1.26 (100 °C),
and 1.18 (175 °C). The replacement reaction first requires a nucleation step: the formation of the first calcite
crystallites larger than the critical size r* (Morse et al, 2007). Empirical nucleation theory relates the activation
energy $\Delta G_A(r^*)$ necessary to form a nucleus of critical size to the specific surface energy $\sigma$ needed to form the
interface between the nucleating phase and the matrix phase as

$$\Delta G_A(r^*) \propto \frac{\sigma^3}{(-RTln\Omega)^2} \qquad (2)$$

Only supercritical nuclei or pre-existing seed crystals of size r > r* of calcite can lower their free energy as their
volume free energy gained by growth exceeds the adverse energy contributions of increasing interface area. To
obtain a significant number of supercritical nuclei a critical supersaturation needs to be reached (Morse et al., 2007,
Gebauer et al., 2008, Nindiyasari et al., 2014, Sun et al. 2015). Reported values for critical supersaturation levels
$\Omega_{crit}$ required for calcite nucleation in various conditions range from the order of 3.7 (Lebron & Suarez, 1996,
Zeppenfeld, 2003) to the order of 30 (Morse et al., 2007; Gebauer et al., 2008) or even several hundreds e.g. in
hydrogel matrices (Nindiyasari et al., 2014). The DFT study of Sun et al. (2015) arrives at $\Omega_{crit}$ = 5 for systems
free of inhibitors such as Mg, and $\Omega_{crit}$ = 35 for modern sea-water. Accordingly, the supersaturation produced by
the dissolution of aragonite is very small compared to supersaturation levels typically required for the nucleation
of calcite. Thus, we can expect that nucleation is a critical kinetic step in the replacement reaction of aragonite by
calcite.

**4.2 Aragonite metastability at 100 °C up to 160 °C**

In our laboratory-based hydrothermal alteration experiments at 100 °C in both meteoric and burial fluids, the
aragonite mineral as well as the characteristic biological microstructure survive the hydrothermal treatment up to
at least 28 days. In experiments at 12 5°C, and 150 °C we did not see any calcite formation from the bioaragonite
either. This is consistent with the findings of Ritter et al. (2016) who analysed the light stable isotope signatures
($\delta^{13}$C, $\delta$ $^{18}$O) of hydrothermally treated samples. In the 100 °C alteration experiments using isotope-doped
experimental fluids, Ritter et al. (2016) found that the carbon and oxygen isotope ratios of the treated shells are
within the same range as those measured in the pristine samples. Furthermore, no obvious patterns emerge from
the comparison of sub-samples exposed to seawater, meteoric, and burial fluids. Most of the extensive literature
on aragonite precipitation from aqueous solutions and aragonite-calcite replacement reactions in aqueous environments, as reviewed in the introduction, makes clear that both temperatures around the boiling point of water and the presence of $Mg^{2+}$ inhibit calcite nucleation. Thus, the inhibition of calcite nucleation favours the growth of aragonite if the solution is supersaturated with respect to the Ca-carbonate phases. If supersaturation is exceedingly high and rapidly generated, vaterite or even amorphous calcium carbonate will precipitate and reduce the supersaturation below the levels required for aragonite or calcite nucleation (Gebauer et al., 2008, 2012; Navrotsky, 2004; Radha et al., 2010). However, it is unlikely that these levels of supersaturation are reached in our case, as aragonite is already present. We, thus, conclude that the absence of an aragonite to calcite replacement reaction in our 100 °C – 150 °C treatments is related to inhibition of calcite nucleation (Sun et al., 2015), a mechanism that has rarely been rigorously explored.

**4.3 Dormant period followed by rapid reaction at 175 °C**

At 175 °C the replacement reaction of biological aragonite to coarse-grained calcite occurs rapidly; it starts after a dormant period of about 4 days and proceeds rapidly almost to completion after 3 more days (Figs. 8, A11). However, even after 84 days about 5 % of residual aragonite is still present. Calcite nucleation occurs (and replacement reaction proceeds) where the experimental fluid is in contact with the bio-aragonite: at the surfaces of the shell, in pores and along growth lines (Figs. 9B, 11, A4-A6).

**4.4 Nucleation and the time lag of the aragonite to calcite replacement reaction at 175 °C**

A certain time lag in the hydrothermal treatment experiments is expected for the initial dissolution of shell aragonite to build-up a sufficiently high ion activity product in the solution to precipitate any calcite. However, the several-day dormant period followed by the rapid growth of calcite indicates that the nucleation of calcite is inhibited, at least initially. We discussed in the previous section that the thermodynamic potential (supersaturation) for the formation of calcite from a fluid, which is able to dissolve aragonite, is smaller than the critical supersaturation required to obtain a discernible nucleation rate for calcite in normal laboratory experiments. The presence of magnesium in the solution further inhibits calcite nucleation and likewise do high temperatures between 70 °C and 160 °C (Kitano et al., 1962; Taft, 1967; Kitano et al., 1972; Katz, 1973; Berner, 1975; Morse et al., 1997; Choudens-Sánchez, 2009; Radha et al. 2010, Balthasar & Cusack, 2015; Sun et al., 2015, Perdikouri et al., 2011, 2013), which is supported by the lack of calcite formation in our experiments between 100°C and 150 °C (Table 1, Fig. A11). A possible scenario explaining the dormant period could be simply that the nucleation rate of calcite is extremely small due to the limited supersaturation, but non-zero. Once a few nuclei formed after a few days, the actual growth process proceeds rapidly from these few nuclei. Another scenario may be the initial, rapid formation of a passivation layer on the surface of the aragonite or on the surface of any calcite nuclei; the dormant period is then the time that is needed to dissolve this passivation layer, at least in some places, where subsequently calcite nuclei of critical size can form. In order to explain this second scenario we can only speculate that after initial dissolution of the biogenic aragonite with excess free energy due to its hybrid nanoscale composite structure an inorganic aragonite precipitates first on the surface of the biogenic aragonite.

**4.5 Grain size and chemistry of the newly formed calcite**

Compared to the nano- to microscale grain fabric of the original aragonite material the newly formed calcite crystals are remarkably large. In meteoric solutions the grain size of the newly formed calcite reaches 200 micrometres (e.g. Figs. 9C) while in the Mg-bearing burial solution newly formed calcite crystals reach sizes in the 1 mm range, in both, the 7- and 84-day treatments (e.g. Figs. 10B, 10C). ). The large calcite grains obtained can very likely be the result of the formation of very few calcite nuclei.

Other explanations for the formation of large calcite grains from the original nano- to microscale grain fabric may be Ostwald-ripening or strain-driven grain growth of the newly formed calcite. The latter could be expected due to the 8.44 % volume increase when the denser aragonite transforms to calcite. To elucidate this possibility we determined the *local misorientation* within the calcite crystals from the EBSD data sets. Maps showing small lattice orientation changes between neighbouring measurement points highlight high dislocation densities and subgrain boundaries, which may have been introduced during the replacement reaction by stresses.

Figure 13 depicts the distribution pattern of local misorientation within five selected EBSD maps (Fig. 13B, 13E, 13H, 13K, 13N). Legends accompany all local misorientation maps (Figs. 13C, 13F, 13I, 13L, 13O). Blue colours indicate the absence of measurable internal misorientation, while light green to yellow colours highlight areas where local misorientation is larger than experimental resolution. Grains in Fig. 1 are defined by a *critical misorientation* selected as 5 ° (i.e. tilts smaller than 5 ° are counted as subgrain boundaries in the mosaic structure of the crystals).

For the better visualization of individual grains we outlined these with white lines. In Figs. 13G, J, M the mosaic structure in the grains is visible in inverse pole figure colouring reflecting lattice orientation. In all five investigated data sets the grain-internal local misorientation reaches up to 2 degrees, thus, neither alteration time, nor the chemical composition of the used alteration solution show any discernible influence on the degree of strain accumulation within the calcite grains. Figure 14 compares the subgrain (mosaic) structure of two large calcite grains obtained in the same experimental fluid at 175 °C, where one grain is from the 7 days treatment, and the other from the 84 days treatment. The grains are marked by stars in Fig. 13K and N, respectively. In these maps of Fig. 14 the colour codes for misorientation relative to a common reference point, rather than for local misorientation. Corresponding legends are given below the grains. The internal misorientation (mosaic spread) for the grain obtained in the 84 days treatment is much higher than that in the grain obtained in the 7 days treatment. We find that the local misorientations are mainly curvilinear structures in the cross section (white arrows in Figs. 14A, 14C) and correspond to subgrain boundaries within the newly formed calcite crystals. These boundaries do not appear to heal or to disappear with an increased alteration time, an indication again of the negligible effect of alteration duration on the fabric and internal structure of calcite grains crystallised from *Arctica islandica* shell bioaragonite.

To further investigate potential grain growth patterns, we took a statistical approach in the analysis of the EBSD measurements shown in Figs. 9 and 10 (alterations experiments carried out for 7 and 84 days at 175 °C in meteoric and burial solution, respectively). Figures 14A and 14B show the statistics of grain area (again, we define a grain by a critical misorientation of 5 °) versus *mean* misorientation within a grain. Based on these statistics, we do not see major evidence for a specific calcite grain growth phenomenon with an increase in alteration time between 7

and 84 days, with the exception of three extremely large grains in the 84 days treatment in burial solution. However, we find that experiments conducted with the Mg-containing burial solution yield larger calcite crystals (black arrows in Fig. 15B) in comparison to the size of the grains obtained from experiments carried out with meteoric water (Fig. 15A). Grains obtained from alteration experiments with meteoric fluid show a significantly higher degree of mean misorientation (up to 10 degrees, black arrows in Fig. 15A), compared to the grains that grew in burial solution. Large mean misorieantations of >4 ° occur notably in the grains grown in the 7 days treatment in meteoric solution, while the corresponding 84 days treatment does not show a significant increase in grain area compared to the 7 days treatment.

In summary, the observations do not support scenarios of Ostwald-ripening or strain-driven anomalous grain growth as the reasons of the large calcite grains. We attribute the large calcite grains to the nucleation rate: The crystals growing from each nucleus consume the aragonite educt (the precursor, original aragonite) until they abutted each other. Thus, larger crystals in the experiment with burial solution result from a smaller number of calcite nuclei, which may be attributed to the presence of aqueous Mg in the experimental fluid. Note here, that both the reduction of Mg concentration in the reactive fluid, compare to that in the initial burial fluid (see Table 1), as well as speciation calculations, suggest that the formation of Mg-bearing carbonate minerals (magnesite and/or dolomite) is likely possible to occur at the experimental conditions. Indeed, we observe small patches of newly formed Mg-rich carbonates (Fig. A13). The formation of such minerals occurs at lower rates compared to pure Ca-bearing carbonates owing to the slow dehydration of aqueous Mg that is required prior to its incorporation in the crystal (e.g. Mavromatis et al., 2013) even at temperature as high as 200 °C (Saldi et al., 2009; 2012). The newly formed calcite contains only small amounts of magnesium (Table A1) in the order of 0.1 wt % (or 0.006 in the formula unit), while the strontium content of the original aragonite in the order of 0.4 wt.% is retained in the calcite (0.005 in the formula unit). The local formation of Mg-rich carbonates occurs at some places at the rim of the sample, where it is in direct contact with the bulk of the experimental fluid (Fig. A13B and Table A1). In these patches measured Mg-contents reach up to 19.7 wt % (0.716 in the formula unit, encountered in scan field 3 at the outer rim of the sample). The averaged composition in scan fields 4 and 9 in Fig. A13B may indicate dolomite, but like scan field 3, which has a Mg content exceeding that of dolomite, we more likely have magnesite with some calcite present, as judged from the EPMA map (Fig. A13B).

**4.6 The calcite to aragonite replacement reaction kinetics**

Inorganic experiments on aragonite to calcite transition at 108 °C in hydrothermal conditions were reported by Bischoff & Fyfe (1968) and by Bischoff (1969). These authors used fine-grained powders as educts (the precursor, original material) and observed a comparatively rapid transition to calcite that was complete within 48 hours, depending on the composition of the fluid. For example, larger $CO_2$ partial pressure (leading to lower pH and thus larger solubility of the carbonates) accelerated, while the presence of Mg-ions retarded the process. This rapid reaction kinetics as reported by Bischoff & Fyfe (1968) and by Bischoff (1969) is discrepant to our observations. We do not see a replacement reaction of the biogenic aragonite to calcite at 100 °C even within 28 days. Hydrothermal experiments by Metzger & Barnard (1968) and by Perdikouri and co-workers (2011, 2013), however, who used aragonite single crystals in their experiments, report reaction kinetics which correspond very well to our observations. They do not observe any evidence of the replacement reaction at 160 °C even within 1

month, but a partial replacement of their aragonite crystals by calcite within 4 weeks at 180 °C. We observed that the fluids used (artificial meteoric and/or burial fluids) cause only a minor difference in replacement reaction kinetics in our experiments, with the $MgCl_2$-bearing artificial burial fluid reducing the nucleation rate of calcite, thus, leading to the observed significantly larger calcite crystals in the product. As compared to the work of

Perdikouri et al. (2011, 2013) on aragonite single crystals, shell-aragonite does not crack during the replacement of the aragonite by calcite. The reason for this difference may be ascribed to the porosity of the bioaragonite, which results from the loss of its organic component. As Figs. 5C - D and the band contrast and orientation maps of Figs.

6A - C illustrate, the (newly formed) calcite product reveals an internal structure that is very reminiscent of the original bioaragonite/biopolymer composite. The structure arises as the solution penetrates along former sites of organic matrix (former aragonite grain boundaries), such that the structural features obtained after alteration still outline the former aragonite grains. Thus, limited grain size of the bioaragonite together with the formerly biopolymer-filled spaces reduce any stresses that may be built up by the specific volume change of the $CaCO_3$

during the replacement reaction. The replacement process preserves original morphological features. Several studies (Putnis & Putnis, 2007, Xia et al., 2009, Putnis and Austrheim, 2010, Kasioptas et al., 2010, Pollok et al.,

2011) experimentally investigated mineral replacement reactions creating pseudomorphs, even reproducing exquisite structures such as the cuttlebone of *Sepia officinalis*. These studies conclude that the essential factor in producing pseudomorphs is the dissolution of the replaced parent material as the rate-limiting step once the replacement reaction proceeds, while the precipitation of the product phase and the transport of solution to the interface must be comparatively fast. The preservation of morphology - even as observed on the nano- to microscale - is ensured if nucleation and growth of the product immediately take place at the surface of the replaced material when the interfacial fluid film between the dissolving and the precipitating phase becomes supersaturated in the product after dissolution of the educt: an interface-coupled dissolution-reprecipitation mechanism (Putnis &

Putnis, 2007). If dissolution of the educt is fast and precipitation of product is slow, more material is dissolved than precipitated, and the solutes can be transported elsewhere. This would create not only an increased pore space which potentially collapses under pressure, but the dissolved material would eventually precipitate elsewhere with its own characteristic (inorganic) morphology rather than reproducing the educt morphology. The fact that some aragonite survives in the dense layers of the shell even after 84 days also points to a slow dissolution rate of aragonite at least in some parts of the shell. New medium-resolution techniques which are capable of mapping the space-dependence of dissolution rates in-situ (Fischer & Lüttge, 2016) may be able to shed some light on the different behaviour of different shell parts in the future.

**4.7 A paleontological perspective of our laboratory-based hydrothermal alteration experiments**

The alteration experiments of recent *A. islandica* under controlled laboratory conditions are very important from a palaeontological perspective as they reproduce burial diagenetic conditions. The understanding of the diagenetic processes which control organism hard tissue preservation is in fact a fundamental prerequisite to taxonomic, taphonomic, palaeoecological, and biostratigraphic studies (e.g. Tucker, 1990). Most organisms have hard tissues composed of calcium carbonate, and its metastable form, aragonite, is one of the first biominerals produced at the

Precambrian-Cambrian boundary (Runnegar & Bengtson, 1990), as well as one of the most widely used skeletonforming minerals in the Phanerozoic record and today; in fact, aragonitic shells/skeletons are produced by hyolithids, cnidarians, algae, and by the widespread and diversified molluscs.

Several studies (Cherns & Wright, 2000; Wright et al., 2003; Wright & Cherns, 2004; James et al., 2005) have underscored that Phanerozoic marine faunas seem to be dominated by calcite-shelled taxa, the labile aragonitic or bimineralic groups being lost during early diagenesis (in the soft sediment, before lithification), potentially causing a serious taphonomic loss. Considering that most molluscs are aragonitic or bimineralic, this loss could be particularly detrimental both for palaeoecological and biostratigraphic studies. However, it has been shown that the mollusc fossil record is not so biased as expected (Harper, 1998; Cherns et al., 2008). This is due to high frequency taphonomic processes (early lithification/hardgrounds, storm plasters, anoxic bottoms, high sedimentation rates) that. throughout the control of organic matter content and residence time in the taphonomically active zone, produce taphonomic windows allowing mollusc preservation (James et al. 2005; Cherns et al., 2008). Even if the factors that control aragonite dissolution are multiple and their interpretation is complex.

The laboratory-based hydrothermal alteration experiments performed here offer very interesting insights into the fate of the aragonitic or bimineralic hard tissues that escape early dissolution during shallow burial and have the potential to enter the fossil record. In particular, the resistance of biogenic aragonite to replacement by calcite up to temperature of 175 °C during hydrothermal alteration offers an additional explanation for the preservation of aragonitic shells/skeletons once they have escaped early dissolution. The results of our experiments neatly explain the observation that the mollusc fossil record is good and allows restoration of evolutionary patterns.

**5 Conclusions**

1. Aragonite crystallite size, porosity, and pore size varies across the cross-section of the valve of modern *Arctica islandica.* While the outer shell layer is highly porous, with pore sizes in the range of a few micrometres, and contains mineral units in the 1-5 μm size range, the inner shell layers are characterised by a dense shell structure with small (1 μm) mineral units and a very low porosity. The innermost section of the shell is penetrated by elongated pores oriented perpendicular to the shell inner surface. At annual growth lines Sr contents are always high, relative to shell increments between the growth lines in both pristine and experimentally altered shell samples. The chemistry of the alteration fluid and the duration of the alteration experiment do not exert a major effect on the concentration of Sr along the growth lines.

2. During hydrothermal alteration at 100 °C for 28 days, most but not the entire biopolymer matrix is destroyed, while shell aragonite and its microstructure are largely preserved.

3. During hydrothermal alteration at 175 °C for 7 days or more, the biopolymer shell fraction is destroyed, such that pathways for fluid penetration are created. At this temperature and time shell aragonite is almost completely transformed to calcite.

4. When meteoric solution is used for alteration, newly formed calcite crystal units reach sizes up to 200 micrometres, while alteration in burial solution induces the formation of calcite crystals that grow up to 1 mm in 7 days. We attribute the latter, larger grains to the Mg-content of the burial solution, which inhibits calcite nucleation. The formation of fewer nuclei leads to the growth of larger calcite crystals.

5. Geochemical results show that calcite nucleates and replacement reaction proceeds where the experimental fluid is in contact with the aragonite: at the two shell surfaces, in pores, and at growth lines, which are thin, formerly organic-filled layers.

6. The replacement reaction of bioaragonite to calcite does not proceed at temperatures much lower than 175 °C. At 175 °C we observe a dormant time of about 4 days during which no XRD-detectable calcite is formed. The replacement reaction then proceeds within 2-3 days to almost completion with small amounts of aragonite still surviving after 84 days in the dense, proximal layer of the shell. The dormant period can be attributed to the low available driving force for calcite nucleation, but further studies dedicated to the nucleation process are necessary.

7. Between two tipping points, one between 50 and 60 °C (Kitano et al. 1962; Taft, 1967, Ogino et al. 1987, Balthasar and Cusack, 2015), the other between 160 and 180 °C (Perdikouri et al, 2011, 2013, this paper), aragonite appears to precipitate from supersaturated aqueous solutions rather than calcite, such that the hydrothermal treatments of aragonite within this temperature bracket do not yield calcite.

8. The tardy kinetics of aragonite replacement by calcite at temperatures lower than 175 °C contributes to explain why aragonitic or bimineralic shells and skeletons have a good potential of preservation and a complete fossil record.

**7 Competing interests**

The authors declare that they have no conflict of interest.

**8 Acknowledgements**

We sincerely thank Dr. F. Nindiyasari for her help with biochemical sample preparation, microtome cutting and microtome polishing and S. He for the preparation of samples for XRD measurements. We very much thank Prof. J. Pasteris, Prof. U. Brand, Prof. L. Fernández Díaz and Prof. C. Putnis for their corrections and fruitful discussions. We thank the German Research Council (DFG) for financial support in the context of the collaborative research initiative CHARON (DFG Forschergruppe 1644, Grant Agreement Number SCHM 930/11-1).

[revised manuscript text omitted]

**Fig. 12.** Solubility products (SP) of aragonite and calcite calculated from the data of Plummer & Busenberg (1982). The labels at the ordinate give the powers of ten, the numbers in the plot give the mantissa of the SP. $\Omega_{max}$ is the difference between the value for aragonite (red) and calcite (green), respectively, and it is the upper bound of the supersaturation available to drive calcite precipitation from aragonite dissolution (thermodynamic driving force $\Delta_{max} = RT \ln \Omega_{max}$). To drive dissolution of aragonite and precipitation of calcite at non-zero rates, the pore fluid needs to be undersaturated with respect to aragonite and supersaturated with respect to calcite.

**Fig. 13.** Calcite grain structure (A, D, G, J, M, IPF colours as indicated in the insert in C) and maps of grain-internal local misorientation distribution (B, E, H, K, N, scales and probability distributions given in C, F, I, L, O) for experimentally altered shells of *A. islandica* carried out in simulated meteoric solution at 175 °C for 7 (A to C) and 84 days (D, E, F), and in burial solution at 175 °C for 7 (G to L) and 84 days (M to O), respectively. Grains are defined by using a critical misorientation of 5 °. Local misorientation reaches up to 2-3 degrees (see legends in C, F, I, L, O), irrespective of alteration duration and solution. The white star in K marks stress-free shell portions, while the yellow star in N indicates the location of an increased stress accumulation.

**Fig. 14.** Colour-coded visualisation (A, C) and degree of internal misorientation (B, D) within two large, mm-sized grains that grew in simulated burial solution at 175 °C for 7 (A) and 84 (C) days. The grain shown in A contains some stress-free portions within its centre (indicated by blue colours and the white star in A), while internal misorientation in the grain shown in C is highly increased and occurs everywhere within the grain (D). The yellow star in C points to the region where, in this grain, stress accumulation is highest.

**Fig. 15.** Grain area versus mean misorientation within individual grains obtained for newly formed calcite at alteration of *Arctica islandica* aragonite in artificial meteoric (A) and in burial (B) solutions at 175 °C and for 7 and 84 days, respectively. The Mg-containing (burial) alteration fluid induces the formation of large calcite grains that show a low degree of misorientation within the grains (B), while with artificial meteoric solution, the solution that is devoid of Mg, significantly smaller grains are obtained. However, the latter occur with a high mean misorientation within the individual, newly formed grains.

**Appendix Fig. A1.** Morphological characteristics of the shell of the bivalve *Arctica islandica*. A detailed description is given in Schöne (2013).

**Appendix Fig. A2.** Accumulation of pores (whitish circular features) within the outer shell portions (A). Yellow stars in B point to the location of two, a few nanometre-sized pores.

**Appendix Fig. A3.** FE-SEM image of microtome cut, polished, etched and critical-point-dried surface of non-biologic aragonite grown from solution.

**Appendix Fig. A4.** $Sr^{2+}$, $Na^+$, and $Cl^-$ concentrations along annual growth lines in a hydrothermally altered shell portion of *Arctica islandica*. The alteration fluid is NaCl-rich, simulating meteoric waters. The degree of fluid infiltration into and through the shell is well traceable with Na+ and Cl-concentrations. Infiltration occurs, in addition through pores, along growth lines that act as conduits for fluid circulation.

**Appendix Fig. A5.** $Sr^{2+}$ concentrations along annual growth lines in pristine (A, B) and hydrothermally altered (C, D) *Arctica islandica* shell portions. White stars indicate regions of the outer shell layer, while yellow stars point to the inner shell parts. Fluids enter the shell at its two surfaces (see enrichment in $Sr^{2+}$ in Fig. A5D) and, especially along growth lines. Neither the degree of hydrothermal alteration, nor the chemistry of the alteration fluid changes significantly the $Sr^{2+}$ contents along the growth lines. Maximal values for both, pristine and altered samples, range between 0.4 and 0.6 wt% Sr.

**Appendix Fig. A6.** $Sr^{2+}$ concentrations along annual growth lines in hydrothermally altered *Arctica islandica* shell portions. Alteration temperature was 175 °C; meteoric water was used as alteration fluid; the alteration experiments lasted for 7 and 84 days. $Sr^{2+}$ concentration scatters for both alteration times around 0.4 wt% $Sr^{2+}$ and is similar to the value measured in the pristine *Arctica islandica* reference samples (see Figs. A5A, A5B).

**Appendix Fig. A7.** Hydrothermally altered *Arctica islandica* shell portions. Burial fluid was used for alteration at 100 °C and for 28 days. A: As the organic membranes and fibrils are destroyed by alteration, large gaps appear between and numerous minute holes within the mineral units. B: Even though, at an alteration temperature of 100°C and alteration times of 28 days biological aragonite of *Arctica islandica* retains its nanoparticulate appearance.

**Appendix Fig. A8.** Pristine (A) and hydrothermally altered (B) shell portion of *Arctica islandica*. Alteration occurred in burial fluid at 175 °C and lasted for 7 days. Well visible in A is the network of biopolymer fibrils between and within pristine aragonite nanoparticles and mineral units.This is destroyed at alteration and umerous voids (B) become visible within the mineral units.

**Appendix Fig. A9.** EBSD band contrast images taken along a cross section from different parts of the shell of pristine *Arctica islandica*. (A) Outer shell layer, (B) central shell section, and (C) inner shell. Well visible is the difference in crystallite size. In contrast to the outer shell layer (A), the innermost shell section is highly dense and consists of minute aragonite crystals.

**Appendix Fig. A10.** Pole figures obtained from EBSD measurements shown in Figure A9. Measurements are done on pristine *Arctica islandica*. SEM images on the left hand side indicate the location of EBSD maps; (A) outer shell layer, (B) central shell portion, (C) inner shell part. The pole figures and MUD values indicate clearly that aragonite co-orientaion increases significantly towards innermost shell sections.

**Appendix Fig. A11.** XRD measurements of experimentally altered *Arctica islandica* samples subjected to alteration temperatures between 125 °C and 175 °C for various lengths of time (1, 2, 3, 4 and 14 days). Calcite formation starts at 175 °C and an alteration time of four days. Red Miller indices (Cc): calcite and black Miller indices: aragonite.

**Appendix Fig. A12.** Representative Rietveld plot for the product of the alteration experiment at 175 °C for 6 days (A) and 84 days (B) in artificial burial solution measured with $MoK_{\alpha 1}$ in transmission and with $CuK_{\alpha 1}$ in reflection, respectively. The diffuse amorphous signal peaking near 12.5° 2θ is due to the Lindemann glass capillary (Ø 0.3 mm) containing the sample.

**Appendix Fig. A13**. BSE image (A) and Mg concentrations (B) of hydrothermally altered *Arctica islandica* shell portion. Alteration occurred in burial solution at 175°C for 84 days. The yellow rectangle in A indicates the shell portion that is shown in B and that was scanned with EPMA. White rectangles in B highlight the extent of shell portions that was used for the determination of mean Mg concentrations given in yellow within each rectangle. Note the formation of magnesium-rich carbonates (see Table A1) along the outer rim of the sample.

**Tables**

**Table 1.** Detailed conditions used in hydrothermal alteration experiments of modern *Arctica islandica*.
Major and minor element chemical data of pristine *Arctica islandica* aragonite and the calcite obtained
after treatment are given in table A1.

| Sample name | Fluid type | NaCl content [mM] | MgCl₂ content [mM] | Temperature [°C] | Experimental time | Alkalinity [mM] | pH | Mg-content of fluid after experiment [mg/L] |
|---|---|---|---|---|---|---|---|---|
| CHA-M-040 AI21 B2 | meteoric | 10 | - | 100 | 28 days | 1.69 | 7.91 | 3 |
| CHA-M-042 AI 23 B2 | meteoric | 10 | - | 175 | 7 days | 7.72 | - | 0 |
| CHA-M-046 AI27 B1 | meteoric | 10 | - | 175 | 84 days | 10.75 | 7.78 | 1 |
| CHA-M-043 AI24 B2 | burial | 100 | 10 | 100 | 28 days | 2.02 | 8.39 | 112 |
| CHA-M-041 AI22 B2 | burial | 100 | 10 | 175 | 7 days | 9.96 | - | 84 |
| CHA-M-046 AI 27 B2 | burial | 100 | 10 | 175 | 84 days | 6.99 | 7.51 | 165 |
| CHA-M-044 AI29 L1 | burial | 100 | 10 | 125 | 1 day | | | |
| CHA-M-044 AI29 L2 | burial | 100 | 10 | 125 | 14 days | | | |
| CHA-M-044 AI29 L3 | burial | 100 | 10 | 150 | 2 days | | | |
| CHA-M-044 AI26 L1 | burial | 100 | 10 | 175 | 1 day | | | |
| CHA-M-044 AI20 L3 | burial | 100 | 10 | 175 | 3 days | | | |
| CHA-M-044 AI28 L2 | burial | 100 | 10 | 175 | 4 days | | | |
| CHA-M-044 AI28 L1 | burial | 100 | 10 | 175 | 4 ¼ days | | | |
| CHA-M-044 AI28 L2 | burial | 100 | 10 | 175 | 4 ¾ days | | | |
| CHA-M-044 AI20 L1 | burial | 100 | 10 | 175 | 5 days | | | |
| CHA-M-044 AI20 L2 | burial | 100 | 10 | 175 | 6 days | | | |

**Table 2.** Crystal co-orientation (texture) strength expressed as multiple of uniform (random) distribution
(MUD) of modern and experimentally altered *Arctica islandica* shells. Ar: aragonite, Cc: calcite.

| Sample name | Fluid type | Temperature [°C] | Experimental time | MUD value of the outermost shell part | MUD value of the central shell part | MUD value of the innermost shell part |
|---|---|---|---|---|---|---|
| modern reference | - | - | - | 12 Ar/32 Ar | 58 Ar | 88 Ar |
| altered specimen *CHA-M-040 AI21 B2* | meteoric | 100 | 28 days | 7 Ar | 27 Ar | 94 Ar |
| altered specimen *CHA-M-043 AI24 B2* | burial | 100 | 28 days | 4 Ar | - | 99 Ar |
| altered specimen *CHA-M-042 AI23 B2* | meteoric | 175 | 7 days | 18 Cc | 15 Cc | - |
| altered specimen *CHA-M-046 AI27 B1* | meteoric | 175 | 84 days | 25 Cc | 32 Cc | - |
| altered specimen *CHA-M-041 AI22 B2* | burial | 175 | 7 days | 36 Cc | 90 Cc | 80/81 Cc |
| altered specimen *CHA-M-046 AI27 B2* | burial | 175 | 84 days | 64 Cc | 62 Cc | - |

**Table A1.** Electron microprobe analyses (CAMECA SX100 system and procedures described in Goetz
et al., 2014) of the original pristine *Arctica islandica* aragonite and of the treated sample CHA-M046
AI27 B2 near the outer rim of the specimen. The analysed regions are shown in Fig. A13B. The [CO$_3$]-
content is nominal.

| Analysed Region | | Mg | Ca | Mn | Na | P | Sr | Fe(II) | C | O | Σ Cations (except P and C) |
|---|---|---|---|---|---|---|---|---|---|---|---|
| 1 | wt% | 8.91 | 25.53 | 0.1 | 0.06 | 0.02 | 0.3 | 0.15 | 51.58 | 13.29 | |
| | Formula | 0.3425 | 0.596 | 0.0015 | 0.0025 | 0.0005 | 0.003 | 0.0025 | 3.018 | 1.034 | 0.9480 |
| 2 | wt% | 8.91 | 2.53 | 0.1 | 0.06 | 0.02 | 0.3 | 0.14 | 51.33 | 13.29 | |
| | Formula | 0.385 | 0.584 | 0.0015 | 0.002 | 0.0005 | 0.003 | 0.0025 | 3.007 | 1.014 | 0.9780 |
| 3 | wt% | 19.74 | 11.08 | 0.07 | 0.28 | 0.05 | 0.25 | 0.17 | 54.46 | 13.82 | |
| | Formula | 0.716 | 0.2445 | 0.001 | 0.011 | 0.0015 | 0.0025 | 0.003 | 3.007 | 1.015 | 0.9775 |
| 4 | wt% | 14.31 | 18.62 | 0.09 | 0.16 | 0.04 | 0.28 | 0.15 | 52.84 | 13.44 | |
| | Formula | 0.5305 | 0.4285 | 0.0015 | 0.006 | 0.001 | 0.003 | 0.0025 | 3.010 | 1.018 | 0.9720 |
| 5 | wt% | 9.46 | 25.49 | 0.1 | 0.06 | 0.02 | 0.29 | 0.16 | 51.29 | 13.05 | |
| | Formula | 0.365 | 0.5965 | 0.002 | 0.0025 | 0.0005 | 0.003 | 0.0025 | 3.01 | 1.019 | 0.9715 |
| 6 | wt% | 0.1 | 38.19 | 0.11 | 0.13 | 0.02 | 0.43 | 0.15 | 48.43 | 12.36 | |
| | Formula | 0.004 | 0.948 | 0.002 | 0.0055 | 0.0005 | 0.005 | 0.0025 | 3.011 | 1.022 | 0.9670 |
| 7 | wt% | 2.48 | 31.32 | 0.1 | 0.12 | 0.03 | 0.36 | 0.14 | 51.44 | 13.94 | |
| | Formula | 0.095 | 0.751 | 0.0015 | 0.005 | 0.001 | 0.004 | 0.0025 | 3.047 | 1.094 | 0.8590 |
| 8 | wt% | 0.15 | 38.26 | 0.11 | 0.12 | 0.02 | 0.43 | 0.15 | 48.37 | 12.32 | |
| | Formula | 0.006 | 0.949 | 0.002 | 0.005 | 0.0005 | 0.005 | 0.0025 | 3.010 | 1.020 | 0.9695 |
| 9 | wt% | 14.4 | 18.03 | 0.09 | 0.17 | 0.03 | 0.28 | 0.15 | 53.15 | 13.62 | |
| | Formula | 0.534 | 0.411 | 0.0015 | 0.0065 | 0.001 | 0.003 | 0.0025 | 3.013 | 1.027 | 0.9585 |
| Original | wt% | 0.07 | 39.24 | 0.11 | 0.46 | 0.02 | 0.43 | 0.15 | 47.44 | 11.76 | 0.07 |
| Aragonite | Formula | 0.003 | 0.988 | 0.002 | 0.02 | 0.0005 | 0.005 | 0.0025 | 2.989 | 0.987 | 1.02 |